# Beyond Minimax Rates in Group Distributionally Robust Optimization via a Novel Notion of Sparsity

Quan Nguyen [1]   Nishant A. Mehta [1]   Cristóbal Guzmán [2]

## Abstract

The minimax sample complexity of group distributionally robust optimization (GDRO) has been determined up to a $\log(K)$ factor, where $K$ is the number of groups. In this work, we venture beyond the minimax perspective via a novel notion of sparsity that we call $(\lambda, \beta)$-sparsity. In short, this condition means that at any parameter $\theta$, there is a set of at most $\beta$ groups whose risks at $\theta$ are all at least $\lambda$ larger than the risks of the other groups. To find an $\epsilon$-optimal $\theta$, we show via a novel algorithm and analysis that the $\epsilon$-dependent term in the sample complexity can swap a linear dependence on $K$ for a linear dependence on the potentially much smaller $\beta$. This improvement leverages recent progress in sleeping bandits, showing a fundamental connection between the two-player zero-sum game optimization framework for GDRO and per-action regret bounds in sleeping bandits. We next show an adaptive algorithm which, up to logarithmic factors, obtains a sample complexity bound that adapts to the best $(\lambda, \beta)$-sparsity condition that holds. We also show how to obtain a dimension-free semi-adaptive sample complexity bound with a computationally efficient method. Finally, we demonstrate the practicality of the $(\lambda, \beta)$-sparsity condition and the improved sample efficiency of our algorithms on both synthetic and real-life datasets.

## 1. Introduction

Performing well across different data subpopulations and being robust to distribution-shift in testing are two of the most important goals in building machine learning models (Ben-Tal et al., 2013; Williamson & Menon, 2019; Sagawa et al., 2020). These goals are especially important for models making decisions that could have societal and safety impacts. A recently proposed framework for achieving these goals is the group distributionally robust optimization (GDRO) framework, in which a learner aims to find a single hypothesis that minimizes the maximum risk over a finite number of data distributions. This minimax objective is often considered in the context of fairness (Rawls, 1971; Williamson & Menon, 2019; Abernethy et al., 2022) when the distributions represent different demographic groups, or as a means to promote robustness when they represent possible shifts in the data distribution (Mohri et al., 2019; Sagawa et al., 2020).

More formally, given an $n$-dimensional hypothesis set $\Theta$ and a group of $K$ distributions $\mathcal{P}_i$, the learner aims to solve the optimization $\min_{\theta \in \Theta} \max_{i \in \{1,...,K\}} R_i(\theta)$, where $R_i(\theta)$ is the risk of the learner with respect to $\mathcal{P}_i$. Intuitively, this objective encourages the learner to find a model with good balance in performance with respect to a finite number of distributions of data, and avoid models that might perform extremely well on one distribution but have significantly worse performance on others. The GDRO framework assumes that the learner has access to a sampling oracle, which returns an i.i.d sample from $\mathcal{P}_i$ upon receiving a request $i \in [K]$. The sample complexity of the learner is the number of samples needed to find an $\epsilon$-optimal hypothesis $\bar{\theta}$ such that the optimality gap $\max_i R_i(\bar{\theta}) - \max_i R_i(\theta^*)$ is smaller than a target value $\epsilon$, where $\theta^*$ is an optimal hypothesis.

Throughout the paper, the $\tilde{O}$ notation hides logarithmic factors. Existing works (Soma et al., 2022; Zhang et al., 2023) have shown a sample complexity lower bound of order $\Omega(\frac{G^2 D^2 + K}{\epsilon^2})$ and a near-matching $\tilde{O}\left(\frac{G^2 D^2 + K}{\epsilon^2}\right)$ worst-case upper bound, where $D$ is the $\ell_2$ diameter of $\Theta$ and $G$ is the Lipschitz constant of the loss function. While these existing results are useful for understanding worst-case scenarios, practical problems may have additional structure that allows for significantly lower sample complexity. In particular, the $\Omega(\frac{G^2 D^2 + K}{\epsilon^2})$ lower bound construction in (Soma et al., 2022) relies on having arbitrarily small gaps (i.e., difference in risks) between groups for all $\theta \in \Theta$. This property rarely holds in practice, where most hypotheses

[1] Department of Computer Science, University of Victoria, Canada [2] Institute for Mathematical and Computational Engineering, Faculty of Mathematics and School of Engineering, Pontificia Universidad Católica de Chile. Correspondence to: Quan Nguyen <manhquan233@gmail.com>.

*Proceedings of the 42nd International Conference on Machine Learning*, Vancouver, Canada. PMLR 267, 2025. Copyright 2025 by the author(s).

can have significant gaps between groups. For example, in car manufacturing, each car model often has noticeably different effects on different surfaces and road conditions.

## 1.1. Contributions and Techniques

We transcend the established minimax bounds by considering problem instances with additional structure. We formally define such a structure called $(\lambda, \beta)$-sparsity in Section 2.1. The main idea of $(\lambda, \beta)$-sparsity is that for all $\theta$, the groups can be divided into two sets: one contains groups with large risks and the other contains groups with small risks. The parameter $\lambda$ specifies the risk-difference between these two sets of groups, while $\beta$ specifies the number of groups with large risks. Let $\beta_\lambda$ denote the smallest $\beta$ for which $(\lambda, \beta)$-sparsity holds. For problem with $(\lambda, \beta)$-sparsity, we show that the dependence on $K$ in the leading term (here and throughout, the term for which $1/\epsilon$ is of the highest order) can be reduced from $O(K \ln K)$ to $O(\beta_\lambda \ln K)$. Table 1 summarizes our main results, which consist of three high-probability upper bounds and a lower bound. The leading terms in the upper bounds grow with $\tilde{O}\left(\frac{D^2 G^2 + \beta_\lambda}{\epsilon^2}\right)$ instead of $\tilde{O}\left(\frac{D^2 G^2 + K}{\epsilon^2}\right)$, where $\beta_\lambda$ could be much smaller than $K$.[1] To the best of our knowledge, these are the first bounds that go beyond the established minimax bound in (Soma et al., 2022; Zhang et al., 2023). The near-matching lower bound is of order $\Omega\left(\frac{D^2 G^2 + \beta}{\epsilon^2}\right)$, generalizing the minimax lower bound in (Soma et al., 2022).

Technically, our results are based on improving the sample complexity of the two-player zero-sum game framework for GDRO (Nemirovski et al., 2009; Zhang et al., 2023). In this framework, a game is played repeatedly as follows: in round $t$, the first player (the min-player) plays a hypothesis $\theta_t \in \Theta$ and the second player (the max-player) plays a group index $i_t \in [K]$ and draws a sample from distribution $\mathcal{P}_{i_t}$. For the max-player, choosing one of $K$ groups and getting an i.i.d sample from that group is similar to pulling one of $K$ arms and getting feedback from that arm in a multi-armed bandit problem. While existing works (Soma et al., 2022; Zhang et al., 2023) use a fixed set of $K$ arms in every round for the max-player to choose from, the $(\lambda, \beta)$-sparsity condition allows us to use a smaller, time-varying subset of active arms of size at most $\beta$. To handle this time-varying action set, we use the sleeping bandits framework (Kleinberg et al., 2010) to model the learning process of the max-player. Critically, recent progress (Nguyen & Mehta, 2024) in bounding the per-action regret in sleeping bandits (details in Section 3.2) enables us to reduce the max player's regret bound and improve the dependency on the number of groups from $K \ln K$ to $\beta \ln K$ in the leading term of the sample complexity.

For the dimension-dependent bounds, the computation of the time-varying subsets of arms for the max-player is based on a uniform convergence bound for $\Theta$ that uses $\tilde{O}\left(\frac{Kn}{\lambda^2}\right)$ samples. The first bound is obtained by an algorithm called `SB-GDRO` that takes $\lambda$ as input and outputs an $\epsilon$-optimal hypothesis using $\tilde{O}\left(\frac{Kn}{\lambda^2} + \frac{G^2 D^2 + \beta_\lambda}{\epsilon^2}\right)$ samples. Letting $\lambda^*$ be the $\lambda$ that minimizes the sample complexity bound of `SB-GDRO`, a natural question is whether it is possible to nearly obtain this minimum sample complexity *without* knowing $\lambda^*$. Surprisingly, in Section 4 we show that such adaptivity is possible. A disadvantage of the fully-adaptive approach is that it is computationally expensive due to the explicit computation of covers of the potentially high-dimensional set $\Theta$. In Section 4.2, we partially resolve this by proposing a computationally efficient semi-adaptive algorithm achieving another dimension-independent $\tilde{O}(\frac{D^2 G^2 + \max(\ln(K), \beta_{\lambda^*})}{\epsilon^2})$ bound in high-precision settings where $\epsilon \ll \lambda^*$.

For the dimension-free bound, we use the Lipschitzness of the loss function and the stability properties in the strategy of the min-player to compute the time-varying active arms using $\tilde{O}\left(\frac{DKG\sqrt{D^2 G^2 + \beta} \ln\left(\frac{KDG}{\epsilon\lambda}\right)}{\lambda^3 \epsilon}\right)$ samples. This leads to a dimension-free bound that is potentially much smaller than the dimension-dependent bounds when $n$ is large.

In Section 6, we present experimental results showing that not only this $(\lambda, \beta)$-sparsity condition holds for high-dimensional practical setting *around* the optimal hypothesis $\theta^*$, but also our algorithms can efficiently (in both sample and computational complexity) compute an estimate of $\lambda^*$ and leverage it to find $\epsilon$-optimal hypotheses with significantly fewer samples compared to baseline methods.

## 1.2. Related Works

We consider the GDRO problem where the loss function is real-valued in $[0, 1]$ and the hypothesis space $\Theta$ is convex and compact. In this setting, (Nemirovski et al., 2009)[2] converts GDRO to a stochastic saddle point problem and uses stochastic mirror descent methods with $O(\frac{K(G^2 D^2 + \ln(K))}{\epsilon^2})$ sample complexity guarantee. (Sagawa et al., 2020) adopts the two-player convex-concave game framework from the deterministic min-max optimization literature (Freund & Schapire, 1999; Cesa-Bianchi & Lugosi, 2006) to obtain $O\left(\frac{K^2(G^2 D^2 + \ln(K))}{\epsilon^2}\right)$ sample complexity bound, which was improved to $\tilde{O}(\frac{D^2 G^2 + K \ln(K)}{\epsilon^2})$ by (Zhang et al., 2023) by refining the approach. An $\Omega\left(\frac{G^2 D^2 + K}{\epsilon^2}\right)$ information-theoretic lower bound was shown in (Soma et al., 2022).

A related, more constrained setting is the class of multi-

---

[1]See Table 1 for full results.

[2] (Nemirovski et al., 2009) do not impose the bounded loss assumption, although (Zhang et al., 2023) do adopt this assumption.

*Table 1.* Summary of main results. $\lambda$-adapt indicates if the bound is adaptive to the best $\lambda^*$ possible. Dimension-free indicates whether the bound depends on the dimension of $\Theta$. $\delta$ is the failure probability.

| Upper and Lower Bounds | $\lambda$-adapt? | Dimension-free? |
|---|---|---|
| $O\left(\frac{Kn\ln\left(\frac{GDK}{\delta}\right)}{\lambda^2} + \frac{(G^2D^2+\beta_\lambda)\ln\left(\frac{K}{\delta}\right)}{\epsilon^2}\right)$ (Theorem 3.4) | $\times$ | $\times$ |
| $O\left(\left(\frac{Kn\ln\left(\frac{GDK}{\delta}\right)}{(\lambda^*)^2} + \frac{(G^2D^2+\beta_{\lambda^*})\ln\left(\frac{K}{\delta}\right)}{\epsilon^2}\right)\ln\left(\frac{1}{\epsilon}\right)\right)$ (Theorem 4.1) | $\checkmark$ | $\times$ |
| $O\left(\frac{DKG\sqrt{(D^2G^2+\beta)\ln(K/\delta)}\ln\left(\frac{KDG}{\epsilon\lambda\delta}\right)}{\lambda^3\epsilon} + \frac{(D^2G^2+\beta)\ln(K/\delta)}{\epsilon^2}\right)$ (Theorem 5.1) | $\times$ | $\checkmark$ |
| $O\left(\frac{(D^2G^2+\max(\ln(K),\beta_{\lambda^*}))\ln(K/\delta)}{\epsilon^2}\right)$ (Theorem 4.2) | $\checkmark$ | $\checkmark$ |
| $\Omega\left(\frac{D^2G^2+\beta}{\epsilon^2}\right)$ (Theorem 3.5) | - | - |

distribution binary classification problems in which $\Theta$ has finite VC-dimension $d$ (Haghtalab et al., 2022; Awasthi et al., 2023). Recent works have established tight minimax sample complexity bounds of order $\frac{d+K}{\epsilon^2}$ for this setting (Zhang et al., 2024; Peng, 2024). Multi-distribution learning with multi-label prediction with offline data was recently explored in (Jang et al., 2024). We refer interested readers to (Haghtalab et al., 2022; Zhang et al., 2024) for a more comprehensive discussion of related works in min-max fairness and federated learning settings.

## 2. Problem Setup

Let $\Theta \subset \mathbb{R}^n$ be a compact convex set of hypotheses, $\mathcal{Z}$ be a sample space and $\ell : \Theta \times \mathcal{Z} \to [0, 1]$ be a loss function measuring the performance of a hypothesis on a data point. Similar to previous works in GDRO (Sagawa et al., 2020; Haghtalab et al., 2022), we use the following assumption.

**Assumption 2.1.** The diameter of $\Theta$ is bounded as $\|\theta\|_2 \leq D$ for all $\theta \in \Theta$. The loss function $\ell$ is convex and $G$-Lipschitz in the first argument, i.e., $|\ell(\theta, \cdot) - \ell(\theta', \cdot)| \leq G\|\theta - \theta'\|_2$ for all $\theta, \theta' \in \Theta$.

There are $K$ groups, each associated with a distribution $(\mathcal{P}_i)_{i,i=1,\ldots,K}$ over $\mathcal{Z}$. Let $[K] = \{1, 2, \ldots, K\}$. Let $R_i(\theta) = \mathbb{E}_{z\sim\mathcal{P}_i}[\ell(\theta, z)]$ be the risk of $\theta$ with respect to group $i$. The worst-case risk of a hypothesis $\theta$ is measured by its maximum risk over these distributions:

$$\mathcal{L}(\theta) = \max_{i\in[K]} R_i(\theta).$$

The objective is find a hypothesis $\theta^*$ with minimum $\mathcal{L}(\theta)$:

$$\theta^* = \underset{\theta\in\Theta}{\arg\min}\, \mathcal{L}(\theta) = \underset{\theta\in\Theta}{\arg\min}\, \max_{i\in[K]} R_i(\theta). \quad (1)$$

The optimality gap of $\bar{\theta} \in \Theta$ is $\text{err}(\bar{\theta}) = \mathcal{L}(\bar{\theta}) - \mathcal{L}(\theta^*)$. Similar to previous works, we assume that the learner has access to a sampling oracle that, for every query $i \in [K]$,

returns an i.i.d sample $z \sim \mathcal{P}_i$. Given a target optimality $\epsilon$, the sample complexity of a learner is the number of samples to find an $\epsilon$-optimal hypothesis $\bar{\theta}$ such that $\text{err}(\bar{\theta}) \leq \epsilon$.

### 2.1. $(\lambda, \beta)$-Sparsity Structure

First, we formally define the notion of a $\lambda$-dominant set.

**Definition 2.2.** For any $\lambda \in [0, 1]$ and $\theta \in \Theta$, a non-empty set of groups $S \subseteq [K]$ is $\lambda$-dominant at $\theta$ if for all $j \notin S$,

$$\min_{i\in S} R_i(\theta) \geq R_j(\theta) + \lambda. \quad (2)$$

Note that $S = [K]$ is a dominant set, as there is no $j$ in the empty set $[K] \setminus S$ such that $R_j(\theta) + \lambda > \min_{i\in[K]} R_i(\theta)$. Next, we introduce $(\lambda, \beta)$-sparsity, our novel condition for GDRO problems.

**Definition 2.3.** For $\lambda \geq 0$ and $\beta \in [1, K]$, a GDRO problem is $(\lambda, \beta)$-sparse if for all $\theta \in \Theta$, there exists a $\lambda$-dominant set whose size is at most $\beta$. If $\lambda > 0$ and $\beta < K$, we say that $(\lambda, \beta)$ is nontrivial.

By definition, a GDRO instance can be $(\lambda, \beta)$-sparse for multiple $(\lambda, \beta)$. For example, a $(0.2, 10)$-sparse problem with $K = 20$ is also $(0.2, 11)$ and $(0.1, 10)$-sparse. Similarly, there can be multiple $\lambda$-dominant sets at each $\theta$. Let $\mathbb{S}_{\lambda,\theta}$ be the collection of all $\lambda$-dominant sets at $\theta$. Since $[K]$ is always a $\lambda$-dominant set, this collection always contains $[K]$. Let $\beta_{\lambda,\theta} = \min_{S\in\mathbb{S}_{\lambda,\theta}} |S|$ be the size of the smallest $\lambda$-dominant set at $\theta \in \Theta$. Then, we have $\beta_\lambda = \max_{\theta\in\Theta} \beta_{\lambda,\theta}$ is the smallest value of $\beta$ such that $(\lambda, \beta)$-sparsity holds. Moreover, all GDRO instances are trivially $(0, 1)$-sparse, in which case the $0$-dominant set contains one of the groups with maximum expected loss. If $(\lambda, \beta)$-sparsity holds for nontrivial $(\lambda, \beta)$, then for every model, there is a prominent gap in the outcome (i.e., risks) of applying that model across different groups.

## 2.2. $(\lambda, \beta)$-sparsity for linear regression with linear Gaussian model

Figure 1 (Right) illustrates the mathematical plausibility of nontrivial $(\lambda, \beta)$-sparsity in the continuous domain via a simple example with $\Theta = [0, 1]$. In the following, we show that a non-trivial $(\lambda, \beta)$-sparsity holds for the problem of linear regression with linear Gaussian model where the number of groups $K$ and the dimensionality $n$ are arbitrarily large. In this problem, each group $i \in [K]$ is characterized by two vectors $\mu_i, \theta_i^* \in \mathbb{R}^n$. A sample $z = (X, y)$ is generated from group $i$ by $X \sim \mathcal{N}(\mu_j, I)$ and $y \sim \mathcal{N}(\langle \theta_j^*, X \rangle, 1)$.

Next, taking $K = 3$ and $n = 1$, we show a choice of the $\mu_j$'s and $\theta_j^*$'s for which $(\lambda = 0.5, \beta = 2)$-sparsity holds. We set the parameter space $\Theta = [-1, 1]$ and use squared loss $\ell(\theta, z) = (X\theta - y)^2$. Next, we take $\mu_j = 0$ for all $j$ and set $\theta_1^* = -1$, $\theta_2^* = 1$, and $\theta_3^* = 0$. The model is well-specified since all $\theta_j^*$ are in $\Theta$. In Appendix A, we show that for any $\theta \in \Theta$, its risks on groups 1, 2, and 3 are $R_1(\theta) = (\theta + 1)^2 + 1$, $R_2(\theta) = (\theta - 1)^2 + 1$, and $R_3(\theta) = \theta^2 + 1$, respectively. Moreover, $\max\{R_1(\theta), R_2(\theta)\} - R_3(\theta) \geq 1$ holds for all $\theta \in \Theta$. This immediately implies that this GDRO instance is $(0.5, 2)$-sparse, and the $0.5$-dominant set is either $\{1\}$, $\{2\}$, or $\{1, 2\}$.

Note that everything extends to higher dimensions $n > 1$ (as higher dimensions only increase the gaps between the groups). To extend to larger $K$, the idea for $j \geq 4$ is to take $\mu_j = 0$ and let the parameter $\theta_j^*$ be slightly perturbed from $\theta_3^*$ so that all these groups' risks are always dominated by (the maximum of) groups 1 and 2 with a margin of 1. As a result, it still is the case that $(0.5, 2)$-sparsity holds.

Next, in Section 3, we begin by presenting an algorithm which, for any input $\lambda \in (0, 1]$, returns an $\epsilon$-optimal hypothesis with sample complexity $\tilde{O}\left(\frac{Kn}{\lambda^2} + \frac{D^2 G^2 + \beta_\lambda}{\epsilon^2}\right)$. For *any* such $\lambda$, including trivial choices for which $\beta_\lambda = K$, this algorithm (with high probability) provides a valid sample complexity guarantee, but the guarantee is most useful for the unknown, optimal $\lambda$ — call it $\lambda^*$ — that minimizes the sample complexity. The focus of Section 4 is adaptive algorithms that obtain, without any knowledge of $\lambda^*$, sample complexity whose order is only larger than that of our previous algorithm (were it given $\lambda^*$) by a logarithmic factor.

## 3. Two-Player Zero-Sum Game Approach

In this section, we present a new algorithm SB-GDRO that, for a given input $\lambda \in (0, 1]$, obtains an $O\left(\frac{Kn \ln(GDK/\delta)}{\lambda^2} + \frac{(G^2 D^2 + \beta_\lambda) \ln(K/\delta)}{\epsilon^2}\right)$ sample complexity. Let $\Delta_K$ be the $K$-dimensional probability simplex. For any $q \in \Delta_K$, let $\phi(\theta, q) = \sum_{i=1}^{K} q_i R_i(\theta)$ be the weighted sum of the risks of $\theta$ over $K$ groups. Following (Nemirovski

et al., 2009), we write the objective function in (1) as

$$\min_{\theta \in \Theta} \mathcal{L}(\theta) = \min_{\theta \in \Theta} \max_{q \in \Delta_K} \phi(\theta, q).$$

The duality gap of $\bar{\theta} \in \Theta$ and $\bar{q} \in \Delta_K$ is defined as

$$\text{err}(\bar{\theta}, \bar{q}) = \max_{q \in \Delta_K} \phi(\bar{\theta}, q) - \min_{\theta \in \Theta} \phi(\theta, \bar{q}).$$

Since $\mathcal{L}(\theta) \geq \phi(\theta, \bar{q})$ for all $\theta$, we have $\text{err}(\bar{\theta}) \leq \text{err}(\bar{\theta}, \bar{q})$. To minimize $\text{err}(\bar{\theta}, \bar{q})$, similar to existing works (Nemirovski et al., 2009; Soma et al., 2022), we employ the following two-player zero-sum game approach: a game is run in $T$ rounds, where in each round, there are two players $\mathcal{A}_\theta$ and $\mathcal{A}_q$ corresponding to the min and max operators in the objective function (1). In round $t$, the min-player $\mathcal{A}_\theta$ first plays a hypothesis $\theta_t$, and then the max-player $\mathcal{A}_q$ plays a vector $q_t \in \Delta_K$. Then, a random group $i_t \sim q_t$ is drawn, and the sampling oracle returns a sample $z_{i_t, t} \sim \mathcal{P}_{i_t}$. The two players compute $\theta_{t+1}$ and $q_{t+1}$ for the next round based on $i_t$ and $z_{i_t, t}$. The min-player's goal is to minimize its regret with respect to the best hypothesis in hindsight:

$$R_{\mathcal{A}_\theta} = \sum_{t=1}^{T} \phi(\theta_t, q_t) - \min_{\theta \in \Theta} \sum_{t=1}^{T} \phi(\theta, q_t). \qquad (3)$$

The max-player's goal is to minimize its regret with respect to the best weight vector in hindsight:

$$R_{\mathcal{A}_q} = \max_{q \in \Delta_K} \sum_{t=1}^{T} \phi(\theta_t, q) - \sum_{t=1}^{T} \phi(\theta_t, q_t). \qquad (4)$$

The SB-GDRO algorithm is illustrated in Algorithm 1. Before the game starts, SB-GDRO draws a set $V_i$ of $m$ samples from each group $i \in [K]$, where $m$ is defined in Lemma 3.1. Let $V = \{V_1, \ldots, V_K\}$ be the collection of these sets. The strategies of the two players are as follows:

- The min-player $\mathcal{A}_\theta$ follows the stochastic mirror descent framework similar to (Zhang et al., 2023). Specifically, given a sample $z_{i_t, t} \sim \mathcal{P}_{i_t}$ and an existing $\theta_t$, $\mathcal{A}_\theta$ computes $\theta_{t+1}$ by

$$\theta_{t+1} = \arg\min_{\theta \in \Theta} \left\{ \eta_{w,t} \langle \tilde{g}_t, \theta - \theta_t \rangle + \frac{1}{2} \|\theta - \theta_t\|_2^2 \right\} \quad (5)$$

where $\eta_{w,t} = \frac{D}{G\sqrt{t}}$ is a time-varying learning rate and $\tilde{g}_t = \nabla \ell(\theta_t, z_{i_t, t})$ is a stochastic gradient of $R_{i_t}(\theta_t)$. Note that $\theta_1 = \arg\min_{\theta \in \Theta} \|\theta\|_2$. We refer to the strategy of the min-player as MinP, whose formal procedure is given in Algorithm 4 in Appendix B.

- The max-player $\mathcal{A}_q$ uses $\theta_t$ and $V$ to compute a set of "active" groups $\hat{S}_{\theta_t}$. A group $i$ is *active* if the empirical risk of $\theta$ with respect to $V_i$ is sufficiently large. Then, a

---

**Algorithm 1** `SB-GDRO` with a known $\lambda$

---

**Input:** Constants $K, D, G, \lambda, \epsilon$, hypothesis set $\Theta \subset \mathbb{R}^n$
Draw $m$ (defined in Lemma 3.1) samples from each group into set $V$
Initialize $\theta_1 = \arg\min_{\theta \in \Theta} \|\theta\|_2$
**for** each round $t = 1, \ldots, T$ **do**
$\quad \hat{S}_{\theta_t} = \texttt{DominantSet}(\theta_t, V, 0.7\lambda)$ // $0.4\lambda$-dominant set at $\theta_t$
$\quad q_t = \texttt{MaxP}(t, \hat{S}_{\theta_t})$ // Action of max-player
$\quad$ Draw $i_t \sim q_t$ and $z_{i_t, t} \sim \mathcal{P}_{i_t}$
$\quad \theta_{t+1} = \texttt{MinP}(\theta_t, z_{i_t, t})$ // Action of min-player
**Return:** $\bar{\theta} = \frac{1}{T}\sum_{t=1}^{T} \theta_t$

---

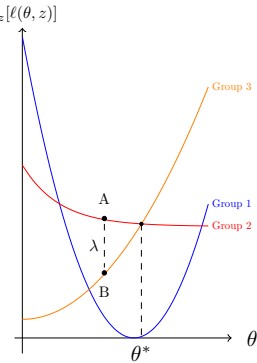

*Figure 1.* (Left) SB-GDRO with known $\lambda$. (Right) A $(\lambda, \beta)$-sparse example with $K = 3, \beta = 2$.

---

**Algorithm 2** `MaxP`: the sleeping bandits max-player $\mathbb{A}_q$

---

**Input:** Time step $t > 0$, a dominant set $\hat{S}_{\theta_t}$
**if** $t = 1$ **then** Initialize $\tilde{q}_{i,t} = 1$ for $i \in [K]$
**else**
$\quad$ Let $h_{i,s} = 1 - \ell(\theta_s, z_{i,s})$ for $s = 1, 2, \ldots, t-1$
$\quad$ Compute $\tilde{q}_{i,t}$ by Equation (7)
**Return:** $q_t$ where $q_{i,t} = \frac{\mathbb{1}\{i \in \hat{S}_{\theta_t}\}\tilde{q}_{i,t}}{\sum_{j=1}^{K} \mathbb{1}\{j \in \hat{S}_{\theta_t}\}\tilde{q}_{j,t}}$

---

**Algorithm 3** `DominantSet`: compute a dominant set $\hat{S}_{\theta_t}$

---

**Input:** $\theta_t \in \Theta$, collection of samples $V$, threshold $\tau > 0$
Compute $\hat{R}_i(\theta_t) = \frac{1}{m}\sum_{j=1}^{m} \ell(\theta_t, V_{i,j})$ for $i \in [K]$
Sort $\hat{R}_i(\theta_t)$ in *decreasing* order and let $\text{ord}(i)$ be the sorted order of group $i$
Compute $\text{nxt}(i)$ by $\text{ord}(\text{nxt}(i)) = \text{ord}(i) + 1$
Let $\hat{i}$ be the first group in $\text{ord}$ such that $\hat{R}_i(\theta_t) \geq \hat{R}_{\text{nxt}(i)}(\theta_t) + \tau$, or $-1$ if no such groups exist.
**Return:** $\hat{S}_{\theta_t} = \{i \in [K] : \text{ord}(i) \leq \text{ord}(\hat{i})\}$ if $\hat{i} \neq -1$, otherwise $\hat{S}_{\theta_t} = [K]$.

---

sleeping bandits algorithm called SB-EXP3 is used to compute a group-sampling probability vector $q_t \in \Delta_K$, where $q_{i,t} > 0$ for $i \in \hat{S}_{\theta_t}$ and $q_{i,t} = 0$ for $i \notin \hat{S}_{\theta_t}$. We refer to the strategy of the max-player as `MaxP`, whose details are given in Algorithm 2.

Compared to existing works (Soma et al., 2022; Zhang et al., 2023; Haghtalab et al., 2022), our two-player zero-sum game procedure has two additional steps: the construction of the collection $V$ and the computation of the set $\hat{S}_{\theta_t}$. At the end of round $T$, the hypothesis $\bar{\theta} = \frac{1}{T}\sum_{t=1}^{T} \theta_t$ is returned. As shown in (Soma et al., 2022), $\text{err}(\bar{\theta}, \bar{q})$ is bounded by $\frac{1}{T}(R_{\mathcal{A}_\theta} + R_{\mathcal{A}_q})$. The min-player $\mathcal{A}_\theta$ uses a variant of the stochastic online mirror descent algorithm in (Zhang et al., 2023) that uses time-varying learning rates instead of fixed learning rates, and obtains the same high-probability $O(DG\sqrt{T\ln(1/\delta)})$ regret bound. Our focus is to obtain an improved bound for the max-player $\mathcal{A}_q$ with a modified strategy.

Next, in Section 3.1, we compute the size of $V$ needed to construct the $\lambda$-dominant sets in each round. Section 3.2 presents the strategy of using $V$ to improve the regret of $\mathcal{A}_q$.

### 3.1. Computing the Dominant Sets

Before the game starts, a set of $m$ samples is drawn from each of $K$ groups. Let $V_{i,j} \in V_i$ be the $j$-th sample collected from group $i$. Let $\hat{R}_i(\theta) = \frac{1}{m}\sum_{j=1}^{m} \ell(\theta, V_{i,j})$ be the

empirical risk of $\theta$ with respect to $V_i$. To compute a $0.4\lambda$-dominant set at $\theta_t$, we use the algorithm `DominantSet` (Algorithm 3) which traverses the groups in order of decreasing $\hat{R}_i(\theta_t)$ and returns a set $\hat{S}_{\theta_t}$ of groups up to (and including) the first group whose empirical risk exceeds the next group's empirical risk by at least $\tau = 0.7\lambda$. The following lemma shows that if $m$ is sufficiently large, then the set $\hat{S}_{\theta_t}$ returned by Algorithm 3 is a $0.4\lambda$-dominant set at $\theta_t$ whose size does not exceed $\beta_\lambda$. This implies that the max-player only needs to sample the groups in $\hat{S}_{\theta_t}$ in order to maximize the cumulative risks over $T$ rounds.

**Lemma 3.1.** *Let* $m = \frac{384n\ln\left(\frac{741GDK}{\delta}\right)}{0.01\lambda^2}$. *With probability at least* $1 - \delta/2$, *for any* $t \in [T]$, `DominantSet` *returns a* $0.4\lambda$*-dominant set* $\hat{S}_{\theta_t}$ *at* $\theta_t$ *satisfying* $\left|\hat{S}_{\theta_t}\right| \leq \beta_\lambda$.

### 3.2. Non-Oblivious Sleeping Bandits

In this section, we discuss the sleeping bandits problem (Kleinberg et al., 2010). Sleeping bandits is a variant of the adversarial multi-armed bandit problem with $K$ arms, where arms can be non-active in each round. Formally, in round $t = 1, 2, \ldots, T$, an adaptive adversary gives the learner a set $\mathbb{A}_t \subseteq [K]$ of active arms. For each arm $i \in \mathbb{A}_t$, the adversary also selects a (hidden) loss value $h_{i,t} \in [0, 1]$. The learner pulls one active arm $i_t \in \mathbb{A}_t$, observes and in-

curs the loss $h_{i_t,t}$. Let $I_{i,t} = \mathbb{1}\{i \in \mathbb{A}_t\}$. For any $a \in [K]$, the per-action regret of the learner with respect to arm $a$ is the difference in the cumulative loss of the learner and that of arm $a$ over the rounds in which $a$ is active:

$$\text{Regret}(a) = \sum_{t=1}^{T} I_{a,t}(h_{i_t,t} - h_{a,t}). \quad (6)$$

**Modified EXP3-IX for sleeping bandits.** We use an algorithm called SB-EXP3 (Nguyen & Mehta, 2024) for sleeping bandits. SB-EXP3 uses the standard IX-loss estimate (Neu, 2015) as the loss estimate $\tilde{h}_{i,t}$ in round $t$, i.e., $\tilde{h}_{i,t} = \frac{h_{i,t}\mathbb{1}\{i_t=i\}}{q_{i,t}+\gamma_t}$, where $\gamma_t > 0$ is the exploration factor in round $t$. For each arm $i$, over $T$ rounds SB-EXP3 maintains a weight vector $\tilde{q}_t \in \mathbb{R}_+^K$ defined as

$$\tilde{q}_{i,t} = \exp\left(\eta_{q,t} \sum_{s=1}^{t-1} I_{i,s}(h_{i_s,s} - \tilde{h}_{i,s} - \gamma_s \sum_{j \in \hat{S}_{\theta_s}} \tilde{h}_{j,s})\right), \quad (7)$$

where $\eta_{q,s} > 0$ is the learning rate and $\tilde{h}_{i,s}$ is the loss estimate of arm $i$ in round $s$. Initially $\tilde{q}_{i,1} = 1$ for $i \in [K]$. The sampling probability $q_t$ is computed by a filtering step, where inactive arms have $q_{i,t} = 0$ and the weights of active arms are normalized as $q_{i,t} = \frac{I_{i,t}\tilde{q}_{i,t}}{\sum_{j=1}^{K} I_{j,t}\tilde{q}_{j,t}}$. The following theorem bounds the per-action regret of SB-EXP3.

**Theorem 3.2.** *With* $\eta_{q,t} = 2\gamma_t = \sqrt{\frac{\ln(3K/\delta)}{\sum_{s=1}^{t}|\mathbb{A}_s|}}$, *SB-EXP3 guarantees that with probability* $1 - \delta$,

$$\max_{a \in [K]} \text{Regret}(a) \leq O\left(\sqrt{\ln(K/\delta) \sum_{t=1}^{T} |\mathbb{A}_t|}\right).$$

Our Theorem 3.2 is a relatively straightforward but important extension of Nguyen & Mehta (2024, Theorem 3). While the latter requires knowing $\max(|\mathbb{A}_t|)_t$ for tuning $\eta_{q,t}$ and $\gamma_t$, we obtain the same bound using adaptive learning rates without knowing anything about future active sets.

### 3.3. Sample Complexity of `SB-GDRO`

In `SB-GDRO`, the max-player uses SB-EXP3 to compute the group-sampling probability $q_t$. For the max-player, the set $\hat{S}_{\theta_t}$ in Algorithm 2 is similar to the set $\mathbb{A}_t$ in sleeping bandits as the set of "active groups" in round $t$ depends on $\theta_t$, which is decided by a non-oblivious adversary (i.e., the min-player). Furthermore, choosing a group $i_t \sim q_t$ and then drawing $z_{i_t,t} \sim \mathcal{P}_{i_t}$ is mathematically equivalent to having $K$ samples $\{z_{i,t} \sim \mathcal{P}_i \mid i \in [K]\}$ (one from each group) but observing only $z_{i_t,t}$. The hidden stochastic loss of group $i$ in round $t$ is $\ell(\theta_t, z_{i,t})$. Note that SB-EXP3 is formulated in terms of minimizing losses rather than maximizing gains, so

similar to (Zhang et al., 2023), we set $h_{i,t} = 1 - \ell(\theta_t, z_{i,t})$ to be the (hidden) stochastic losses of arms $i$ for SB-EXP3. A fundamental connection between the two-player zero-sum game approach in GDRO and sleeping bandits is shown in the following lemma, which states that the regret of the max-player $R_{\mathcal{A}_q}$ is bounded by the per-action regret with $\hat{S}_{\theta_t}$ being the set of active groups at round $t$.

**Lemma 3.3.** *With probability at least* $1 - \delta/2$, *the regret of the max-player is bounded by*

$$R_{\mathcal{A}_q} \leq \max_{i \in [K]} \sum_{t=1}^{T} \mathbb{1}\{i \in \hat{S}_{\theta_t}\}\left(R_i(\theta_t) - \phi(\theta_t, q_t)\right).$$

Theorem 3.2 and Lemma 3.3 imply the following sample complexity bound for `SB-GDRO`.

**Theorem 3.4.** *For any* $\epsilon > 0, \delta \in (0,1)$, *with probability* $1 - \delta$, *Algorithm 1 has sample complexity*

$$O\left(\frac{Kn\ln(GDK/\delta)}{\lambda^2} + \frac{(D^2G^2 + \beta_\lambda)\ln(K/\delta)}{\epsilon^2}\right). \quad (8)$$

In Theorem 3.4, because $\lambda$ is a fixed problem-dependent quantity while the required optimality gap $\epsilon$ can be arbitrarily small, the dependency on $K$ in Theorem 3.4 is dominated by $O\left(\frac{\beta_\lambda \ln(K/\delta)}{\epsilon^2}\right)$. The following lower bound shows that the upper bound in Theorem 3.4 is essentially near-optimal.

**Theorem 3.5.** *For any algorithm $\mathcal{A}$ and any* $\lambda \geq 0.5, \beta \geq 3$, *there exists a* $(\lambda, \beta)$-*sparse GDRO instance with* $\beta_\lambda = \beta$ *so that the sample complexity of $\mathcal{A}$ is at least* $\Omega\left(\frac{G^2D^2 + \beta}{\epsilon^2}\right)$.

## 4. $\lambda^*$-Adaptive Sample Complexity

Theorem 3.4 suggests that a desirable $\lambda$ must be significantly larger than $\epsilon$ (so that $\frac{K}{\lambda^2} \ll \frac{K}{\epsilon^2}$) but also small enough that $\beta_\lambda \ll K$. In this section, we define the notion of an optimal $\lambda^*$ and present a sample-efficient approach for adapting to this unknown $\lambda^*$. First, we write the sample complexity in Theorem 3.4 in the form

$$\ln(K/\delta)\left(\frac{C}{\lambda^2} + \frac{\beta_\lambda}{\epsilon^2}\right) + \frac{D^2G^2\ln(K/\delta)}{\epsilon^2}, \quad (9)$$

where $C = \frac{Kn\ln\left(\frac{GDK}{\delta}\right)}{\ln(K/\delta)}$. By definition, $\beta'_\lambda \leq \beta_\lambda$ for any $\lambda' \leq \lambda$, and thus $\lambda \mapsto \beta_\lambda$ is non-decreasing. Let $\lambda^*$ be the $\lambda$ that minimizes (9). Our goal is to develop a sample-efficient method to find $\lambda^*$.

To describe our approach for finding $\lambda^*$, it will be useful to frame the idea of an optimal $\lambda$ more generically. Consider any $C > K \geq 1$ (not necessarily taking the value above), $\epsilon \in (0,1)$, and $\delta$ in $(0,1)$. Let $g: [0,1] \to [1, K]$ be a

nondecreasing function which is unknown. Now, let $\lambda^*_{C,g}$ be the minimizer, among all $\lambda \in [0, 1]$, of

$$\text{Cost}^{(\text{GDRO})}_{C,g}(\lambda) := \frac{C}{\lambda^2} + \frac{g(\lambda)}{\epsilon^2}. \qquad (10)$$

Clearly, if $C = \frac{Kn\ln\left(\frac{GDK}{\delta}\right)}{\ln(K/\delta)}$ and $g(\lambda) = \beta_\lambda$, then $\lambda^*_{C,g} = \lambda^*$. In general, $g$ (e.g., $\lambda \mapsto \beta_\lambda$) is unknown. However, $g$ can be evaluated at any $\lambda \in (0, 1]$ at a cost of $\text{Cost}^{(\text{Query})}_{C,g}(\lambda) = O(C\ln(K/\delta)/\lambda^2)$ samples. The problem $\text{OPT}(C, g)$ is to find $\lambda^*_{C,g}$ using as few samples as possible.

Now, at a high level (our actual approach in Section 4.1 slightly differs), by solving $\text{OPT}(C, g)$ for $C$ as above and $g(\lambda) = \beta_\lambda$, we obtain a fully adaptive algorithm for GDRO that adapts to $\lambda^*$ and has total (including the cost of finding $\lambda^*$) sample complexity whose rate (in big-$O$) is equal to the product of $\ln(1/\epsilon)$ and (9) with $\lambda$ replaced by $\lambda^*$; here, $\ln(1/\epsilon)$ is the price paid for adaptivity. We present this algorithm in Section 4.1. However, this algorithm is computationally intractable for large $n$, and so Section 4.2 introduces a computationally efficient semi-adaptive algorithm with total sample complexity that, in high-precision settings where $\epsilon \ll \lambda^*$, swaps the $\beta_{\lambda^*}$ in the fully adaptive algorithm's sample complexity with $\max\{\ln K, \beta_{\lambda^*}\}$; moreover it entirely avoids the dimension-dependent term $\frac{C}{(\lambda^*)^2}$, making it dimension-free.

### 4.1. $\lambda^*$-Adaptive Sample Complexity for GDRO

We present an algorithm called SB-GDRO-A, shown in full in Algorithm 8 in Appendix C.2. The idea of this algorithm is to *(i)* construct a non-decreasing function $\hat{g}$ so that $\text{Cost}^{(\text{GDRO})}_{C,\hat{g}}(\lambda^*_{C,\hat{g}})$ is sufficiently close to $\text{Cost}^{(\text{GDRO})}_{C,\beta_{(\cdot)}}(\lambda^*_{C,\beta})$ with high probability; *(ii)* solve $\text{OPT}(C, \hat{g})$ to get $\lambda^*_{C,\hat{g}}$; *(iii)* input $\lambda^*_{C,\hat{g}}$ into SB-GDRO. Our approach uses at most $O(\text{Cost}^{(\text{GDRO})}_{C,\beta_{(\cdot)}}(\lambda^*_{C,\beta})\ln\left(\frac{1}{\epsilon}\right))$ samples for steps *(i)* and *(ii)*, which, together with Theorem 3.4, gives us the following theorem (proved in Appendix C.2).

**Theorem 4.1.** *For any $\epsilon > 0, \delta \in (0, 1)$, with probability at least $1 - \delta$, SB-GDRO-A (Algorithm 8) with $\eta_{w,t}, \eta_{q,t}$ and $\gamma_t$ defined in Theorem 3.4 has sample complexity*

$$O\left(\left(\frac{Kn\ln\left(\frac{GDK}{\delta}\right)}{(\lambda^*)^2} + \frac{(D^2G^2 + \beta_{\lambda^*})\ln\left(\frac{K}{\delta}\right)}{\epsilon^2}\right)\ln\left(\frac{1}{\epsilon}\right)\right).$$

Compared to Theorem 3.4, the sample complexity bound in Theorem 4.1 contains an additional multiplicative factor of $O(\ln(1/\epsilon))$, which we consider a small price for not knowing $\lambda^*$ beforehand. Next, we briefly describe the two main steps above, with the full details in Appendix C.

**First Step: Constructing $\hat{g}$**  We first describe a method that, given $\lambda \in [0, 1]$, returns an estimate $\hat{\beta}_\lambda$ for $\beta_\lambda$ using at most $O(\frac{C\ln(K/\delta)}{\lambda^2})$ samples. This method constructs a $\frac{0.1\lambda}{G}$-cover for $\Theta$, uses Algorithm 3 to compute a $0.4\lambda$-dominant set at each element of the cover, and then returns as its estimate $\hat{\beta}_\lambda$ the maximum cardinality among these dominant sets. Now, the function $\hat{g}$ is defined by setting $\hat{g}(\lambda)$ equal to 1 for $\lambda \le \frac{\epsilon}{2}$, setting it to $\hat{\beta}_\lambda$ for $\lambda$ in the geometric sequence $(1, \frac{1}{5}, \frac{1}{5^2}, \ldots)$, and then interpolating at other $\lambda$ to form a non-decreasing step function. In Appendix C.2, we prove that with high probability, $\beta_{0.2\lambda} \le \hat{\beta}_\lambda \le \beta_\lambda$ and $\hat{g}$ is non-decreasing, leading to $\lambda^*_{C,\hat{g}}$ being close to $\lambda^*_{C,\beta}$.

**Second Step: Solving for $\lambda^*_{C,\hat{g}}$**  Our method for solving $\text{OPT}(C, g)$ is called SolveOpt. It outputs $\hat{\lambda}$ such that $\text{Cost}^{(\text{GDRO})}(\hat{\lambda}) = O(\text{Cost}^{(\text{GDRO})}(\lambda^*))$ while using $O(\text{Cost}^{(\text{GDRO})}(\lambda^*)\ln(1/\epsilon))$ samples; note that we drop the subscripts $C$ and $g$. The main idea of SolveOpt is to maintain two variables $U$ and $L$ which specify an interval $[L, U]$ that always contains a good estimate of $\lambda^*$. We iteratively evaluate $g(\lambda)$ for $\lambda \in [L, U]$ and shrink this interval, i.e., $U$ monotonically decreases while $L$ monotonically increases. The shrinking process is based on comparing $\text{Cost}^{(\text{GDRO})}(\lambda)$ and $\text{Cost}^{(\text{GDRO})}(U)$: if $\text{Cost}^{(\text{GDRO})}(\lambda) < \text{Cost}^{(\text{GDRO})}(U)$, then $U$ is set to $\lambda$ and $L$ is increased accordingly. The process stops when $\lambda < L$, at which point the algorithm return the last value of $U$ as its estimate of $\lambda^*$. The value of $\lambda$ is taken from a geometric sequence; this ensures that at most $\ln(1/\epsilon)$ values of $g(\lambda)$ are evaluated, leading to the $\ln(1/\epsilon)$ multiplicative factor in the final bound.

### 4.2. A Semi-Adaptive Bound in High-Precision Settings

While SB-GDRO-A is fully adaptive to $\lambda^*$, it relies on building covers for $\Theta$, which is computationally intensive when $n$ is large. We now propose a semi-adaptive, computationally efficient algorithm called SB-GDRO-SA that avoids covers. The main idea is to merge the $\lambda^*$-estimation process into the two-player zero-sum game: starting with $\lambda = 1$, if the dominant sets $S_{\lambda,\theta_t}$ computed in round $t$ of the game is bigger than a threshold (e.g. $\ln(K)$), then similar to SolveOpt, we decrease $\lambda$ exponentially (e.g. $\lambda \leftarrow \lambda/2$). To avoid a too small $\lambda$, we also set a lower threshold $L$ so that $\lambda$ stops decreasing once $\lambda \le L$. These two thresholds, one for $|S_{\lambda,\theta_t}|$ and one for $\lambda$, determine the trade-off between adaptivity and sample complexity. In SB-GDRO-SA, we use $\ln(K)$ and $L = \tilde{O}(\epsilon\sqrt{Kn})$ as the two thresholds. Let $\lambda_{\ln(K)}$ be the largest $\lambda$ such that $\beta_\lambda = \ln(K)$. In high-precision settings where $\epsilon \ll \lambda^*$, the following theorem states that Algorithm 9 is adaptive to $\max(\lambda_{\ln(K)}, \lambda^*)$.

**Theorem 4.2.** *If $\epsilon\sqrt{\frac{C}{\ln(K)}} < \lambda^*$, then with probability at least $1 - \delta$, SB-GDRO-SA (Algorithm 9 in Appendix C.3)*

*has sample complexity*

$$O\left(\frac{(D^2G^2 + \max(\ln(K), \beta_{\lambda^*}))}{\epsilon^2}\ln(K/\delta)\ln\frac{1}{\epsilon}\right)$$

We emphasize that Theorem 4.2 holds without knowing $\lambda^*$. This bound guarantees that in high-precision settings, Algorithm 9 enjoys (on average) dominant sets of small sizes that never exceed $\max(\beta_{\lambda^*}, \ln(K))$. Remarkably, this bound is also dominantly dimension-free, although the algorithm still uses the dimension $n$.

## 5. A Dimension-Independent Approach

In this section, we present `SB-GDRO-DF`, a modified version of Algorithm 1 that uses $O\left(\frac{KDG\sqrt{(D^2G^2+\beta)\ln(K/\delta)}}{\lambda^3\epsilon}\right)$ samples for computing the dominant sets over $T$ rounds of the two-player zero-sum game. This bound avoids the dependency on $n$, the dimension of $\Theta$, which might be preferable in high-dimensional settings. The trade-off for getting rid of $n$ is an additional $\frac{1}{\lambda\epsilon}$ multiplicative factor in the non-leading term of the sample complexity bound.

We assume that a pair $(\lambda, \beta)$ is known such that the problem instance is $(\lambda, \beta)$-sparse. Unlike `SB-GDRO`, `SB-GDRO-DF` does not use a fixed set of samples $V$ for computing the dominant sets of all $\theta \in \Theta$. Instead, `SB-GDRO-DF` computes the dominant sets only for the hypotheses $\theta_t$ that the learner encounters during the game. In particular, the $T$ rounds are divided into $\frac{T}{\sigma}$ episodes, in which each episode has $\sigma$ consecutive rounds that use the same dominant set. By the stability property of the regularized update (5) and the Lipschitzness of the loss function $\ell$, if $\sigma$ is sufficiently small then the differences between the risks of the hypotheses within each episode is small. This implies that a dominant set for $\theta_t$ will remain a dominant set (possibly with smaller gaps) and therefore can be reused for the hypotheses $\theta_{t+1}, \theta_{t+2}, \ldots, \theta_{t+\sigma}$. The full procedure is given in Algorithm 10 in Appendix D, and its sample complexity is stated in the following theorem.

**Theorem 5.1.** *For any $\epsilon > 0, \delta \in (0, 1)$, with probability $1 - \delta$, `SB-GDRO-DF` with $\eta_{w,t} = \frac{2D}{G\sqrt{T}}$, $\eta_{q,t}$ and $\gamma_t$ defined in Theorem 3.4 returns an $\epsilon$-optimal hypothesis with sample complexity*

$$O\left(\begin{array}{c}\frac{DKG\sqrt{(D^2G^2+\beta)\ln(K/\delta)}\ln\left(\frac{KDG}{\epsilon\lambda\delta}\right)}{\lambda^3\epsilon} \\ +\frac{(D^2G^2+\beta)\ln(K/\delta)}{\epsilon^2}\end{array}\right).$$

## 6. Experimental Results

We support our theoretical findings with empirical results in two different GDRO instances: one with the lower bound environment constructed in Theorem 3.5, and another with the Adult dataset (Becker & Kohavi, 1996). On the lower bound environment, we set $\epsilon = 0.005, K = 10, \lambda^* = 0.2$ and $\beta_{\lambda^*} = 2$ so that the maximum risks can only be attained by the first two groups for any $\theta$. On the Adult dataset, we use the same setup as (Soma et al., 2022) and divide $48\,842$ samples into groups based on `race × gender` with the goal of finding a linear classifier that determines whether the annual outcome of a person exceeds USD $50\,000$ based on $n = 5$ features: age, years of education, capital gain, capital loss, and number of working hours. Similar to (Soma et al., 2022), $\mathcal{P}_i$ is the empirical distribution over samples in group $i$. One difference from (Soma et al., 2022) is we have $K = 10$ groups from 5 races and 2 genders instead of 6 groups, so that the difference between $\ln(K)$ and $K$ is amplified. With $\epsilon = 0.001$, we use hinge loss and normalize the features so that the losses are in $[0, 1]$. We set $T = 10^6$ and $\delta = 0.01$ on both GDRO instances. The results are aggregated from five independent runs with random seeds $\{0, 1, 2, 3, 4\}$. To compute $\theta^*$, we run the two-player zero-sum game with *ideal players* who have access to the underlying distributions $\mathcal{P}_i$. More experimental details are in Appendix G.

On both GDRO instances, we compare `SB-GDRO-SA` (Algorithm 9) to the Stochastic Mirror Descent for GDRO algorithm (`SMD-GDRO`) proposed by (Zhang et al., 2023). To the best of our knowledge, `SMD-GDRO` is the only suitable baseline with a near-optimal high-probability guarantee in the minimax regime.

### 6.1. Discovering non-trivial $(\lambda, \beta)$-sparsity

Figure 2 (Left) shows the sizes $\left|\hat{S}_{\theta_t}\right|$ and the average $\frac{1}{t}\sum_{h=1}^t \left|\hat{S}_{\theta_h}\right|$ computed by `SB-GDRO-SA` in the first $10\,000$ rounds. On GDRO with Adult dataset, it indicates that `SB-GDRO-SA` quickly discovers dominant sets of sizes smaller than $\lceil\ln(K)\rceil$ within the first 3000 rounds. This shows that a non-trivial $(\lambda, \ln(K))$-sparsity condition indeed holds for hypotheses *around* $\theta^*$ in practical settings. Further inspection reveals this $(\lambda, \ln(K))$-sparsity is discovered early in the game without using too many samples: on the lower bound environment the final $\lambda$ is $0.125 \approx 0.5\lambda^*$ using roughly 3000 samples, while on the Adult dataset the final $\lambda$ is $\frac{1}{2^9} \approx \epsilon\sqrt{\frac{C}{\ln(K)}}$ using roughly $36\,000$ samples. Both of these values are much smaller than $T$, and as $T$ is scaled with $\frac{1}{\epsilon^2}$, this empirically supports the insight in Theorem 4.2 that the sample complexity is dominated by the number of rounds needed in the two-player zero-sum game.

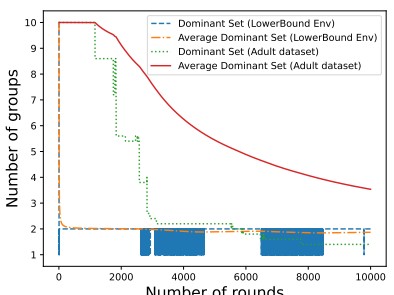 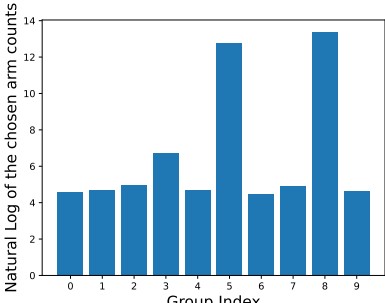

*Figure 2.* (Left) Sizes of the dominant sets in the first 10000 rounds computed by `SB-GDRO-SA`. (Right) The number of times a group is selected by the max-player, displayed in natural log. The highest group (group 8) is female Amer-Indian-Eskimo people.

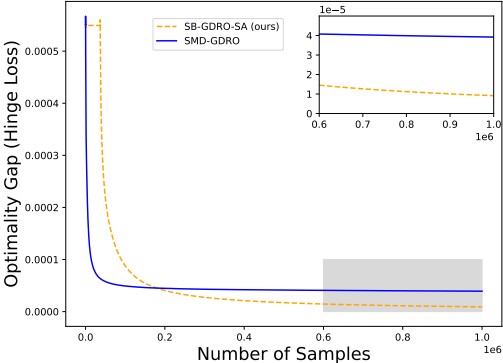

*Figure 3.* The optimality gap of `SB-GDRO-SA` and `SMD-GDRO` on GDRO with the Adult dataset. Lower is better.

### 6.2. Convergence Properties of `SB-GDRO-SA`

Next, we show results indicating that `SB-GDRO-SA` finds a $\epsilon$-optimal hypothesis using fewer samples than `SMD-GDRO`. Figure 3 shows the optimality gap err of $\bar{\theta}_t$ of `SB-GDRO-SA` and `SMD-GDRO` as a function of the number of drawn samples on the Adult dataset. Initially, `SB-GDRO-SA` uses more samples than `SMD-GDRO` because `SB-GDRO-SA` needs to estimate $\lambda^*$. However, as $\theta_t$ gets closer to $\theta^*$, the optimality gap of `SB-GDRO-SA` decreases much quicker since it only collects samples from the two groups with the largest risks. While `SMD-GDRO` struggles to get an optimality gap under $4 \times 10^{-5}$ even after nearly $T = 10^6$ samples, `SB-GDRO-SA` manages to do so well below $4 \times 10^5$ samples. Figure 2 (Right) shows an interesting observation that more than $60\%$ of the samples drawn by `SB-GDRO-SA` are from the *female Amer-Indian-Eskimo* group. This is in stark contrast to the fact that this group constitutes only $0.3\%$ of the dataset (186 out of $48\,842$ samples). This underlines the *robustness* aspect of GDRO, which is different compared to the traditional empirical risk minimization regime where samples from the largest groups contribute more to the optimization process.

## 7. Conclusion and Future Work

We introduced a new structure called $(\lambda, \beta)$-sparsity into the GDRO problem. We showed a fundamental connection between the per-action regret in sleeping bandits and the optimality gap of the two-player zero-sum game approach for the GDRO problem, and then improved the dependency from $O(K \ln(K))$ to $O(\beta \ln(K))$ in the leading term of the sample complexity of $(\lambda, \beta)$-sparse problems, even when the optimal $\lambda$ is unknown. We also showed a near-matching lower bound, which both extends and generalizes the lower bound construction in minimax settings to the $(\lambda, \beta)$-sparse settings.

**Global versus local sparsity.** One interesting future direction is relax the $(\lambda, \beta)$-sparsity to hold only within some neighborhood of $\theta^*$. This *local sparsity* condition is a more practical version of Definition 2.3. Our experiments show that our algorithms, which are developed for the global version of $(\lambda, \beta)$-sparsity, still work well for a real-world dataset where only the local version holds. To rigorously solve the local sparsity version, it seems that developing *high-probability last iterate convergence guarantee* for the sequence of $\theta_t$'s in stochastic games is needed. To our knowledge, this is a still an fundamental open problem for non-strongly convex losses.

## Impact Statement

We study the GDRO problem generally; yet, because the GDRO problem setting can capture societally relevant applications like min-max fairness, we acknowledge that our theoretical work could later be applied in settings where min-max fairness is the goal. Whether or not min-max fairness is appropriate can depend on the particular application domain being considered and especially on the particular meaning of the loss function adopted in that domain.

## Acknowledgments

QN and NM were supported by NSERC Discovery Grant RGPIN-2018-03942. CG was partially supported by INRIA Associate Teams project, ANID FONDECYT 1210362 grant, ANID Anillo ACT210005 grant, and National Center for Artificial Intelligence CENIA FB210017, Basal ANID.

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

# A. Proof for the Global Sparsity in Linear Regression in Section 2.2

## A.1. Expected Loss of Linear Regression

Let $\mu \in \mathbb{R}, \theta^* \in [-1, 1], X \sim \mathcal{N}(\mu, 1)$ and $Y(X) \sim \mathcal{N}(X\theta^*, \sigma^2)$. Consider the linear regression problem where the goal is to predict $\theta^*$. Let $\theta \in \mathbb{R}$ be a prediction for $\theta^*$. The loss of $\theta$ on an example $(X, Y)$ is the squared loss

$$\ell(\theta, (X, Y) = (X\theta - Y)^2. \tag{11}$$

The expected loss of $\theta$ is

$$\mathbb{E}[(X\theta - Y)^2] = \mathbb{E}[X^2\theta^2 + Y^2 - 2YX\theta] \tag{12}$$
$$= \theta^2\mathbb{E}[X^2] + \mathbb{E}[Y^2] - 2\theta\mathbb{E}[YX]. \tag{13}$$

We have $\mathbb{E}[X^2] = (\mathbb{E}[X])^2 + \mathrm{Var}(X) = \mu^2 + 1$. Moreover,

$$\begin{aligned} \mathbb{E}[Y^2] &= \mathbb{E}_X[\mathbb{E}_{Y|X}[Y^2 \mid X]] \\ &= \mathbb{E}_X[(\mathbb{E}_{Y|X}[Y \mid X])^2 + \mathrm{Var}_{Y|X}(Y)] \\ &= \mathbb{E}_X[(X\theta^*)^2 + \sigma^2] \\ &= (\theta^*)^2\mathbb{E}[X^2] + \sigma^2 \\ &= (\theta^*)^2(\mu^2 + 1) + \sigma^2. \end{aligned} \tag{14}$$

In addition,

$$\begin{aligned} \mathbb{E}[YX] &= \mathbb{E}_X[X\mathbb{E}_{Y|X}[Y \mid X]] \\ &= \mathbb{E}_X[X^2\theta^*] \\ &= \theta^*\mathbb{E}_X[X^2] \\ &= \theta^*(\mu^2 + 1). \end{aligned} \tag{15}$$

Overall, we have

$$\begin{aligned} \mathbb{E}[(X\theta - Y)^2] &= \theta^2(\mu^2 + 1) + (\theta^*)^2(\mu^2 + 1) + \sigma^2 - 2\theta\theta^*(\mu^2 + 1) \\ &= (\mu^2 + 1)(\theta^2 + (\theta^*)^2 - 2\theta\theta^*) + \sigma^2 \\ &= (\mu^2 + 1)(\theta - \theta^*)^2 + \sigma^2. \end{aligned} \tag{16}$$

The expected loss of $\theta$ is $R(\theta) = (\mu^2 + 1)(\theta - \theta^*)^2 + \sigma^2$.

## A.2. GDRO with $K = 3$

Let $K = 3$. Consider the GDRO problem with $K$ groups, $\mu_1 = \mu_2 = \mu_3 = 0, \sigma_1 = \sigma_2 = \sigma_3 = 1$ and $\theta_1^* = -1, \theta_2^* = 1, \theta_3^* = 0$. For any $\theta \in \mathbb{R}$, from the previous section, we have the expected loss of $\theta$ on these three groups is

$$\begin{aligned} R_1(\theta) &= (\mu_1^2 + 1)(\theta - \theta_1^*)^2 + \sigma_1^2 = (\theta + 1)^2 + 1, \\ R_2(\theta) &= (\mu_2^2 + 1)(\theta - \theta_2^*)^2 + \sigma_2^2 = (\theta - 1)^2 + 1, \\ R_3(\theta) &= (\mu_3^2 + 1)(\theta - \theta_3^*)^2 + \sigma_3^2 = \theta^2 + 1. \end{aligned} \tag{17}$$

Next, we will show that for all $\theta \in \mathbb{R}$,

$$\max\{R_1(\theta), R_2(\theta)\} - R_3(\theta) \geq 1. \tag{18}$$

Indeed, we have

$$\begin{aligned} \max\{R_1(\theta), R_2(\theta)\} - R_3(\theta) &= \max\{\theta^2 + 2\theta + 2, \theta^2 - 2\theta + 2\} - (2\theta^2 + 1) \\ &= \max\{2\theta, -2\theta\} + 2 \geq 1. \end{aligned}$$

Obviously, the result holds for $\theta \in [-1, 1]$ since it holds for $\theta \in \mathbb{R}$. We conclude that that this GDRO instance is $(0.5, 2)$-sparse.

## B. Proofs for Section 3

For a pseudo-metric space $(\mathcal{F}, \|.\|)$, for any $\nu > 0$, let $\mathcal{N}(\mathcal{F}, \nu, \|.\|)$ be the $\nu$-covering number of $\mathcal{F}$; that is $\mathcal{N}(\mathcal{F}, \nu, \|.\|)$ is the minimal number of balls of radius $\nu$ needed to cover $\mathcal{F}$.

First, we prove the following lemma on a uniform convergence bound that holds for a sufficiently large value of $m$.

**Lemma B.1.** *Let* $m = \frac{384 n \ln\left(\frac{741 G D K}{\delta}\right)}{0.01 \lambda^2}$. *With probability at least* $1 - \delta/2$, *the event*

$$\mathcal{E}_{i,\theta} = \{|\hat{R}_i(\theta) - R_i(\theta)| \leq 0.15\lambda\} \tag{19}$$

*holds simultaneously for all* $i \in [K]$ *and* $\theta \in \Theta$.

### B.1. Proof of Lemma B.1

Our proof for the uniform convergence bound in Lemma B.1 is based on the Rademacher complexity bound of the class of functions $L_\Theta$ defined as follows:

$$L_\Theta = \{\ell(\theta, .) : \mathcal{Z} \to [0, 1], \theta \in \Theta\}, \tag{20}$$

which is the set of all possible functions $\ell(\theta, .)$ for $\theta \in \Theta$. First, we state the following bound for the empirical Rademacher complexity based on the chaining argument (Dudley, 1967; Liao, 2020).

**Lemma B.2.** *(Dudley's Entropy Integral Bound (Dudley, 1967; Liao, 2020)) Let* $\mathcal{F} = \{f : \mathcal{Z} \to \mathbb{R}\}$ *be a class of real-valued functions,* $S = \{z_1, z_2, \ldots, z_m\}$ *be a set of* $m$ *random i.i.d samples. For a function* $f \in \mathcal{F}$, *let*

$$\|f\|_{2,S} = \sqrt{\frac{1}{m} \sum_{j=1}^{m} (f(z_j))^2} \tag{21}$$

*be an* $S$-*dependent seminorm of* $f$. *Assuming*

$$\sup_{f \in \mathcal{F}} \|f\|_{2,S} \leq c,$$

*where* $c$ *is a positive constant, we have*

$$\mathrm{Rad}(\mathcal{F}, S) \leq \inf_{\epsilon \in [0, \frac{c}{2}]} \left(4\epsilon + \frac{12}{\sqrt{m}} \int_\epsilon^{\frac{c}{2}} \sqrt{\ln\left(\mathcal{N}(\mathcal{F}, \nu, \|.\|_{2,S})\right)} d\nu\right), \tag{22}$$

*where* $\mathrm{Rad}(\mathcal{F}, S) = \frac{1}{m} \mathbb{E}_{\sigma \in \{\pm 1\}^m} \left[\sup_{f \in \mathcal{F}} \sum_{j=1}^{m} \sigma_j f(z_j)\right]$ *is the empirical Rademacher complexity of* $\mathcal{F}$ *and* $\mathcal{N}(\mathcal{F}, \nu, \|.\|_{2,S})$ *is the size of a* $\nu$-*cover of* $\mathcal{F}$.

A proof of this lemma can be found in (Liao, 2020). We now prove Lemma B.1.

*Proof (of Lemma B.1).* For $i \in [K]$, Theorem 26.5 in (Shalev-Shwartz & Ben-David, 2014) states that with probability at least $1 - \frac{\delta}{4K}$ over the set $V_i$ of size $m$, for all $\theta \in \Theta$,

$$\left|\frac{1}{m} \sum_{j=1}^{m} \ell(\theta, V_{i,j}) - R_i(\theta)\right| \leq 2\mathrm{Rad}(L_\Theta, V_i) + \sqrt{\frac{32 \ln(4K/\delta)}{m}}.$$

Our proof is based on the fact that the covering number of the compact set $\Theta \subset \mathbb{R}^n$ is finite, and hence the empirical Rademacher complexity $\mathrm{Rad}(L_\Theta, V_i)$ is bounded for all $i \in [K]$. Because the values of the loss function $\ell$ is in $[0, 1]$, we

have $\|f\|_{2,V_i} \leq 1$ for all $f \in L_\Theta$. Moreover, the diameter of $L_\Theta$ measured in $\|.\|_{2,V_i}$ is

$$\max_{\theta,\theta' \in \Theta} \sqrt{\frac{1}{m} \sum_{j=1}^m \left(\ell(\theta, V_{i,j}) - \ell(\theta', V_{i,j})\right)^2} \leq \max_{\theta,\theta' \in \Theta} \sqrt{\frac{1}{m} \sum_{j=1}^m G^2 \|\theta - \theta'\|_2^2}$$

$$\leq \sqrt{\frac{1}{m} \sum_{j=1}^m G^2 D^2}$$

$$= GD,$$

where the first inequality is due to the Lipschitzness of the loss function $\ell$ and the second inequality is due to $D$ being the diameter of $\Theta$ measured in $\ell_2$-norm. Applying Lemma B.2 with $c = 1$ and $\epsilon = 0$, we have

$$\text{Rad}(L_\Theta, V_i) \leq \frac{12}{\sqrt{m}} \int_0^{\frac{1}{2}} \sqrt{\ln\left(\mathcal{N}(L_\Theta, \nu, \|.\|_{2,V_i})\right)} d\nu$$

$$\leq \frac{12G}{\sqrt{m}} \int_0^{\frac{1}{2}} \sqrt{\ln\left(\frac{4GD}{\nu}\right)^n} d\nu$$

$$= \frac{12\sqrt{n}}{\sqrt{m}} \int_0^{\frac{1}{2}} \sqrt{\ln\left(\frac{4GD}{\nu}\right)} d\nu$$

where the second inequality is due to a result that the size of the smallest $\nu$-cover on a set $\mathcal{F}$ with diameter $d$ is bounded by $(\frac{4d}{\nu})^n$ (see e.g. Carl & Stephani, 1990, Equation 1.1.10). To compute this integral, let $u = \sqrt{\ln(4GD/\nu)}$. We then have $\nu = 4GDe^{-u^2}$, and $d\nu = 4GDd(e^{-u^2})$. As $\nu \to 0, u \to \infty$. As $\nu \to \frac{1}{2}, u \to \sqrt{\ln(8GD)}$. Hence,

$$\int_0^{\frac{1}{2}} \sqrt{\ln\left(\frac{4GD}{\nu}\right)} d\nu = 4GD \int_\infty^{\sqrt{\ln(8GD)}} u d(e^{-u^2})$$

$$= 4GD \left( ue^{-u^2} \Big|_\infty^{\sqrt{\ln(8GD)}} - \int_\infty^{\sqrt{\ln(8GD)}} e^{-u^2} du \right)$$

$$= 4GD \left( \frac{\sqrt{\ln(8GD)}}{8GD} + \int_{\sqrt{\ln(8GD)}}^\infty e^{-u^2} du \right)$$

$$\leq 4GD \left( \frac{\sqrt{\ln(8GD)}}{8GD} + \frac{\sqrt{\pi}}{2} e^{-\ln(8GD)} \right)$$

$$= \frac{2\sqrt{\ln(8GD)} + \sqrt{\pi}}{4},$$

where the second equality is integration by parts and the inequality is by a Chernoff-type bound on the Gaussian error function $\frac{2}{\sqrt{\pi}} \int_x^\infty e^{-t^2} dt \leq e^{-x^2}$ (Chang et al., 2011). Overall, we have

$$\text{Rad}(L_\Theta, V_i) \leq \frac{3\sqrt{n}}{\sqrt{m}} \left(2\sqrt{\ln(8GD)} + \sqrt{\pi}\right).$$

We conclude that the uniform convergence bound is

$$\left| \frac{1}{m} \sum_{j=1}^m \ell(\theta, V_{i,j}) - R_i(\theta) \right| \leq \frac{3\sqrt{n}}{\sqrt{m}} \left(2\sqrt{\ln(8GD)} + \sqrt{\pi}\right) + \sqrt{\frac{32 \ln(4K/\delta)}{m}}.$$

By setting the right-hand side to $0.15\lambda$, solving for $m$ and simplifying, we obtain the following sufficient condition on $m$:

$$m \geq \frac{384n \ln\left(\frac{741GDK}{\delta}\right)}{0.01\lambda^2} \tag{23}$$

so that with probability at least $1 - \frac{\delta}{4K}$, we have $\left| \frac{1}{m} \sum_{j=1}^m \ell(\theta, V_{i,j}) - R_i(\theta) \right| \leq 0.15\lambda$ for all $\theta \in \Theta$. Taking a union bound over all $K$ groups leads to the desired statement. $\qquad\square$

---

**Algorithm 4** `MinP`: the stochastic-OMD min-player $\mathbb{A}_\theta$

---

**Input:** $\theta_t \in \Theta$, sample $z_{i_t,t}$

Compute $\tilde{g}_t = \nabla \ell(\theta_t, z_{i_t,t})$

Compute $\theta_{t+1} = \arg \min_{\theta \in \Theta} \{\eta_{w,t} \langle \tilde{g}_t, \theta - \theta_t \rangle + \frac{1}{2}\|\theta - \theta_t\|_2^2\}$ by Equation (5)

**Return:** $\theta_{t+1}$

---

## B.2. Proof of Lemma 3.1

*Proof.* From Lemma B.1, we immediately have the event $\mathcal{E}_{i,\theta}$ holds simultaneously for all $i \in [K]$ and $\theta \in \Theta$ with probability at least $1 - \frac{\delta}{2}$. Thus, it suffices to prove the desired statement assuming that all $\mathcal{E}_{i,\theta}$ hold. Let $R_{i,t} = R_i(\theta_t)$ and $\hat{R}_{i,t} = \hat{R}_i(\theta_t)$ be the risk and empirical risk of $\theta_t$ with respect to group $i$, respectively. We consider two cases: $\beta_\lambda < K$ and $\beta_\lambda = K$.

**When $\beta_\lambda < K$:**

In this case, there exists a non-empty set $\lambda$-dominant set $S_{\lambda,\theta_t}$ whose size is smaller than $\beta_\lambda < K$. This implies that the set $[K] \setminus S_{\lambda,\theta_t}$ is also non-empty. For any $i \in S_{\lambda,\theta_t}$ and $k \in [K] \setminus S_{\lambda,\theta_t}$, due to $\mathcal{E}_{i,\theta_t}, \mathcal{E}_{k,\theta_t}$ and by Definition 2.2, we have

$$
\begin{aligned}
\hat{R}_{i,t} - \hat{R}_{k,t} &\geq (R_{i,t} - 0.15\lambda) - (R_{k,t} + 0.15\lambda) \\
&= R_{i,t} - R_{k,t} - 0.3\lambda \\
&\geq \lambda - 0.3\lambda \\
&= \tau > 0.
\end{aligned}
$$

Thus, at any time $t$, the sorted sequence of groups can be divided into two non-empty parts: the first contains all groups in $S_{\lambda,\theta_t}$ and the second contains the rest. Since $|S_{\lambda,\theta_t}| \leq \beta_\lambda$, the size of the first part is at most $\beta_\lambda$. Let $i^* = \arg \max_{j \in S_{\lambda,\theta_t}} \{\mathrm{ord}(j)\}$ be the last group in the first part. Since $\mathrm{nxt}(i^*) \in [K] \setminus S_{\lambda,\theta_t}$, we have $\hat{R}_{i^*,t} \geq \hat{R}_{\mathrm{nxt}(i^*),t} + \tau$. This satisfies the condition in Algorithm 3, therefore the resulting set $\hat{S}_{\theta_t}$ is non-empty and its size does not exceed $\beta_\lambda$. To show that $\hat{S}_{\theta_t}$ is a $0.4\lambda$-dominant set, for any $i' \in \hat{S}_{\theta_t}$ and $k' \in [K] \setminus \hat{S}_{\theta_t}$, we have

$$
\begin{aligned}
R_{i',t} - R_{k',t} &\geq (\hat{R}_{i',t} - 0.15\lambda) - (\hat{R}_{k',t} + 0.15\lambda) \\
&= \hat{R}_{i',t} - \hat{R}_{k',t} - 0.3\lambda \\
&\geq \hat{R}_{\hat{i},t} - \hat{R}_{\mathrm{nxt}(\hat{i}),t} - 0.3\lambda \\
&\geq \tau - 0.3\lambda = 0.4\lambda,
\end{aligned}
\tag{24}
$$

where the second inequality is from the definition of $\hat{i}$ and $\hat{S}_{\theta_t} = \{i \in [K] : \mathrm{ord}(i) \leq \mathrm{ord}(\hat{i})\}$, we have $\mathrm{ord}(i') \leq \mathrm{ord}(\hat{i}), \mathrm{ord}(\mathrm{nxt}(\hat{i})) \geq \mathrm{ord}(k')$ and the empirical risks are sorted in decreasing order.

**When $\beta_\lambda = K$:**

In this case, the inequality $\left|\hat{S}_{\theta_t}\right| \leq \beta_\lambda$ holds trivially. To show that $\hat{S}_{\theta_t}$ is a $0.4\lambda$-dominant set, we further consider two sub-cases: $\hat{i} \neq -1$ and $\hat{i} = -1$.

- $\hat{i} \neq -1$: in this case, the set $\hat{S}_{\theta_t} = \{i \in [K] : \mathrm{ord}(i) \leq \mathrm{ord}(\hat{i})\}$ has size at most $K - 1$ because the group with the largest empirical risk is excluded. Therefore, by the same argument as in (24), the set $\hat{S}_{\theta_t}$ is a $0.4\lambda$-dominant set.

- $\hat{i} = -1$: in this case, we have $\hat{S}_{\theta_t} = [K]$ is trivially a $0.4\lambda$-dominant set by Definition 2.2.

We conclude that the set $\hat{S}_{\theta_t}$ is a $0.4\lambda$-dominant set at $\theta_t$ and $\left|\hat{S}_{\theta_t}\right| \leq \beta_\lambda$. $\qquad \square$

---

**Algorithm 5** `FTARLShannon`: Follow the regularized and active leader with Shannon entropy regularizer and time-varying learning rates for sleeping bandits

---

**Input:** $K \geq 2$
Initialize $\tilde{L}_{i,0} = 0$ for all arms $i \in [K]$.
**for** each round $t = 1, \ldots,$ **do**
    The non-oblivious adversary selects and reveals $\mathbb{A}_t$
    Compute $q_{i,t} = \frac{\exp\left(-\eta_t \tilde{L}_{i,t}\right)}{\sum_{j=1}^{K} \exp\left(-\eta_t \tilde{L}_{j,t}\right)}$
    Compute $p_{i,t} = \frac{I_{i,t} q_{i,t}}{\sum_{j=1}^{K} I_{j,t} q_{j,t}}$ by Equation (26)
    Draw arm $i_t \sim p_t$ and observe $\hat{\ell}_t = \ell_{i_t, t}$
    **for** each arm $i \in [K]$ **do**
        If $I_{i,t} = 1$, compute $\tilde{\ell}_{i,t} = \frac{\mathbb{1}\{i_t=i\}\hat{\ell}_t}{p_{i,t}+\gamma_t}$ by Equation (27)
        If $I_{i,t} = 0$, compute $\tilde{\ell}_{i,t} = \hat{\ell}_t - \gamma_t \sum_{j \in \mathbb{A}_t} \tilde{\ell}_{j,t}$ by Equation (28)
        Update $\tilde{L}_{i,t} = \tilde{L}_{i,t-1} + \tilde{\ell}_{i,t}$

---

### B.3. Proof of Theorem 3.2

Let $A_t = |\mathbb{A}_t|$ be the number of active arms in round $t$. Throughout this section, we write $\eta_t = \eta_{q,t}$ for the learning rate of the SB-EXP3 algorithm used by the max-player.

The $O\left(\sqrt{\ln(K/\delta)\sum_{t=1}^{T} A_t}\right)$ high-probability per-action regret bound of the SB-EXP3 algorithm in (Nguyen & Mehta, 2024) was established for a fixed learning rate $\eta_t = \eta$ and a fixed exploration factor $\gamma_t = \gamma$. In this section, we generalize their result to algorithms with time-varying learning rates and exploration factors defined as follows:

$$\eta_t = 2\gamma_t = \sqrt{\frac{\ln(3K/\delta)}{\sum_{s=1}^{t} A_s}}. \tag{25}$$

Note that $\eta_t$ and $\gamma_t$ are chosen *after* the set of active arms $\mathbb{A}_t$ is revealed. As pointed out in Nguyen & Mehta (2024, Appendix G), the SB-EXP3 algorithm is equivalent to their Follow-the-Regularized-and-Active-Leader (FTARL) algorithm with the Shannon entropy regularizer. Therefore, a high-probability regret bound of FTARL with Shannon entropy regularizer and $\eta_t$ and $\gamma_t$ defined in (25) would imply Theorem 3.2. For completeness, we provide the full procedure of FTARL with Shannon entropy regularizer in Algorithm 5. For each arm $i \in [K]$ and round $t \in [T]$, this algorithm maintains an estimated cumulative loss $\tilde{L}_{i,t}$ defined as

$$\tilde{L}_{i,t} = \sum_{s=1}^{t} \tilde{\ell}_{i,t},$$

and computes the weight of arm $i$ in round $t$ by

$$q_{i,t} = \frac{\exp\left(-\eta_t \tilde{L}_{i,t-1}\right)}{\sum_{j=1}^{K} \exp\left(-\eta_t \tilde{L}_{j,t-1}\right)},$$

where $\eta_t$ is the learning rate in round $t$. Initially, $\tilde{L}_{i,0} = 0$ for all arms $i \in [K]$. Upon receiving the set $\mathbb{A}_t$ of active arms, the sampling probability $p_t$ is computed by normalizing $I_{i,t} q_{i,t}$ as follows:

$$p_{i,t} = \frac{I_{i,t} q_{i,t}}{\sum_{j=1}^{K} I_{j,t} q_{j,t}}. \tag{26}$$

Note that $I_{i,t} = \mathbb{1}\{i \in \mathbb{A}_t\}$, hence $p_{i,t}$ is non-zero only for active arms. An arm $i_t \sim p_t$ is drawn according to $p_t$ and its loss $\hat{\ell}_t = \ell_{i_t, t}$ is observed. For an active arm $i \in \mathbb{A}_t$, its loss estimate is the IX-loss estimator (Neu, 2015):

$$\tilde{\ell}_{i,t} = \frac{\mathbb{1}\{i_t = i\}\hat{\ell}_t}{p_{i,t} + \gamma_t}, \tag{27}$$

where $\gamma_t$ is the exploration factor in round $t$. For a non-active arm $i \notin \mathbb{A}_t$, its loss estimate is defined as the difference between the observed loss $\hat{\ell}_t$ and the weighted sum of estimated losses of active arms (Nguyen & Mehta, 2024):

$$\tilde{\ell}_{i,t} = \hat{\ell}_t - \gamma_t \sum_{j \in \mathbb{A}_t} \tilde{\ell}_{j,t}. \tag{28}$$

The following theorem states the per-action regret bound of Algorithm 5.

**Theorem B.3.** *Let $(\eta_t)_{t=1,\dots}$ and $(\gamma_t)_{t=1,\dots}$ be two sequences of non-increasing learning rates and exploration factors such that $\eta_t \leq 2\gamma_t$. With probability at least $1 - \delta$, FTARLShannon (Algorithm 5) guarantees that*

$$\max_{a \in [K]} \text{Regret}(a) \leq \frac{\ln(K)}{\eta_T} + \frac{\ln(3K/\delta)}{2\gamma_T} + \ln(3/\delta) + \sum_{t=1}^{T} \left(\frac{\eta_t}{2} + \gamma_t\right) A_t. \tag{29}$$

The proof of this theorem is in Appendix E. We are now ready to prove Theorem 3.2

*Proof (of Theorem 3.2).* By plugging (25) into the bound in Theorem B.3, we obtain

$$
\begin{aligned}
\max_{a \in [K]} \text{Regret}(a) &\leq \frac{\ln(K)}{\eta_T} + \frac{\ln(3K/\delta)}{\eta_T} + \ln\left(\frac{3}{\delta}\right) + \sum_{t=1}^{T} \eta_t A_t \\
&\leq \frac{2\ln(3K/\delta)}{\eta_T} + \ln\left(\frac{3}{\delta}\right) + \sum_{t=1}^{T} \eta_t A_t \\
&= \frac{2\ln(3K/\delta)}{\eta_T} + \ln\left(\frac{3}{\delta}\right) + \sqrt{\ln(3K/\delta)} \sum_{t=1}^{T} \frac{A_t}{\sqrt{\sum_{s=1}^{t} A_s}} \\
&= 2\sqrt{\ln(3K/\delta) \sum_{t=1}^{T} A_t} + \ln\left(\frac{3}{\delta}\right) + \sqrt{\ln(3K/\delta)} \sum_{t=1}^{T} \frac{A_t}{\sqrt{\sum_{s=1}^{t} A_s}}.
\end{aligned}
$$

We bound $\sum_{t=1}^{T} \frac{A_t}{\sqrt{\sum_{s=1}^{t} A_s}}$ as follows: let $C_t = \sum_{s=1}^{t} A_s$ and $C_0 = 0$. Then,

$$
\begin{aligned}
\sum_{t=1}^{T} \frac{A_t}{\sqrt{\sum_{s=1}^{t} A_s}} &= \sum_{t=1}^{T} \frac{C_t - C_{t-1}}{\sqrt{C_t}} \\
&= \sum_{t=1}^{T} \int_{C_{t-1}}^{C_t} \frac{dx}{\sqrt{C_t}} \\
&\leq \sum_{t=1}^{T} \int_{C_{t-1}}^{C_t} \frac{dx}{\sqrt{x}} \\
&= \int_{C_0}^{C_T} \frac{dx}{\sqrt{x}} \\
&= 2\sqrt{C_T},
\end{aligned}
$$

where the inequality holds because $\frac{1}{\sqrt{x}} \geq \frac{1}{\sqrt{C_t}}$ for all $C_{t-1} \leq x \leq C_t$. This implies that

$$
\begin{aligned}
\max_{a \in [K]} \text{Regret}(a) &\leq 2\sqrt{\ln(3K/\delta) \sum_{t=1}^{T} A_t} + \ln\left(\frac{2}{\delta}\right) + 2\sqrt{\ln(3K/\delta) \sum_{t=1}^{T} A_t} \\
&= O\left(\sqrt{\ln(K/\delta) \sum_{t=1}^{T} A_t}\right).
\end{aligned}
$$

$\square$

## B.4. Proof of Lemma 3.3

*Proof.* Since $\Delta_K$ is convex, we can write

$$\max_{q\in\Delta_K} \sum_{t=1}^{T} \phi(\theta_t, q) = \max_{q\in\Delta_K} \sum_{t=1}^{T} \sum_{i=1}^{K} q_i R_i(\theta_t)$$

$$= \max_{q\in\Delta_K} \sum_{i=1}^{K} q_i \sum_{t=1}^{T} R_{i,t}$$

$$= \max_{i\in[K]} \sum_{t=1}^{T} R_{i,t}.$$

Thus,

$$R_{\mathcal{A}_q} = \max_{i\in[K]} \sum_{t=1}^{T} R_{i,t} - \sum_{t=1}^{T} \phi(\theta_t, q_t).$$

If a group $i$ is not included in $\hat{S}_{\theta_t}$ at time $t$, then by Lemma 3.1, for any $k \in \hat{S}_{\theta_t}$ we have

$$R_{i,t} < R_{i,t} + 0.4\lambda \le R_{k,t}.$$

By construction, the probability vector $q_t$ contains non-zero elements only for groups in $\hat{S}_{\theta_t}$, hence for any $i \notin \hat{S}_{\theta_t}$, we have

$$R_{i,t} - \phi(\theta_t, q_t) = \sum_{k\in\hat{S}_{\theta_t}} q_{k,t}(R_{i,t} - R_{k,t}) \le 0.$$

We conclude that for any $i \in [K]$,

$$\sum_{t=1}^{T} R_{i,t} - \phi(\theta_t, q_t) \le \sum_{t=1}^{T} \mathbb{1}\{i \in \hat{S}_{\theta_t}\}(R_{i,t} - \phi(\theta_t, q_t)),$$

hence

$$R_{\mathcal{A}_q} = \max_{i\in[K]} \sum_{t=1}^{T} R_{i,t} - \sum_{t=1}^{T} \phi(\theta_t, q_t)$$

$$\le \max_{i\in[K]} \sum_{t=1}^{T} \mathbb{1}\{i \in \hat{S}_{\theta_t}\} (R_i(\theta_t) - \phi(\theta_t, q_t)).$$

$\square$

## B.5. Proof of Theorem 3.4

Let $\beta_t = \left|\hat{S}_{\theta_t}\right|$ be the size of $\hat{S}_{\theta_t}$. Let $\bar{\beta}_T = \frac{1}{T}\sum_{t=1}^{T}\beta_t$ be the average number of active groups over $T$ rounds. We first state the following bound for the regret of the max-player as a function of $\beta_t$, which is obtained directly by combining Theorem 3.2 and Lemma 3.3.

**Lemma B.4.** *With probability at least $1 - \delta/4$, the regret of the max-player in* SB-GDRO *(Algorithm 1) is bounded by*

$$R_{\mathcal{A}_q} \le O\left(\sqrt{\sum_{t=1}^{T} \beta_t \ln(K/\delta)}\right).$$

*Proof.* The max-player in Algorithm 1 uses the sleeping bandits algorithm SB-EXP3 with the stochastic loss of arm $i$ at round $t$ is

$$h_{i,t} = 1 - \ell(\theta_t, z_{i,t}).$$

Let $H_{i,t} = \mathbb{E}_{z_{i,t} \sim \mathcal{P}_i}[h_{i,t}]$ be the expected value of $h_{i,t}$. We have $H_{i,t} = 1 - R_i(\theta_t)$. Note that both $h_{i,t}$ and $H_{i,t}$ are in $[0,1]$. Fix a group $a \in [K]$ and let $I_{a,t} = \mathbb{1}\{a \in \hat{S}_{\theta_t}\}$. The per-action regret of group $a$ is

$$
\begin{aligned}
\text{GroupRegret}(a) &= \sum_{t=1}^{T} I_{a,t}(R_a(\theta_t) - \phi(\theta_t, q_t)) \\
&= \sum_{t=1}^{T} I_{a,t}\left(R_a(\theta_t) - \sum_{i=1}^{K} q_{i,t} R_i(\theta_t)\right) \\
&= \sum_{t=1}^{T} I_{a,t}\left(\sum_{i=1}^{K} q_{i,t} H_{i,t} - H_{a,t}\right) \\
&= \sum_{t=1}^{T} I_{a,t}\left(\sum_{i=1}^{K} q_{i,t} H_{i,t} - h_{i_t,t} + h_{i_t,t} - h_{a,t} + h_{a,t} - H_{a,t}\right) \\
&= \underbrace{\sum_{t=1}^{T} I_{a,t}\left(\sum_{i=1}^{K} q_{i,t} H_{i,t} - h_{i_t,t}\right)}_{(A)} + \underbrace{\sum_{t=1}^{T} I_{a,t}(h_{a,t} - H_{a,t})}_{(B)} + \underbrace{\sum_{t=1}^{T} I_{a,t}(h_{i_t,t} - h_{a,t})}_{(C)}.
\end{aligned}
$$

The term $C$ is exactly the per-action regret of arm $a$ in SB-GDRO-SA defined in Equation (6) which, by Theorem 3.2, is bounded by $O\left(\sqrt{\ln(K/\delta) \sum_{t=1}^{T} \beta_t}\right)$ with probability at least $1 - \frac{\delta}{12}$ simultaneously for all $a \in [K]$. Next, we bound the terms $A$ and $B$. Since

$$\mathbb{E}_{i_t \sim q_t}[\mathbb{E}_{z_{i_t,t} \sim \mathcal{P}_{i_t}}[h_{i_t,t}]] = \mathbb{E}_{i_t \sim q_t}[H_{i_t,t}]$$
$$= \sum_{i=1}^{K} q_{i,t} H_{i,t}$$

and

$$
\begin{aligned}
\left|\sum_{i=1}^{K} q_{i,t} H_{i,t} - h_{i_t,t}\right| &= \left|\sum_{i=1}^{K} q_{i,t}(H_{i,t} - h_{i_t,t})\right| \\
&\leq \sum_{i=1}^{K} q_{i,t}|H_{i,t} - h_{i_t,t}| \\
&\leq 1,
\end{aligned}
$$

$A$ is a sum of a martingale difference sequence in which the absolute values of its elements are bounded by 1. By Azuma-Hoeffding inequality, with probability at least $1 - \frac{\delta}{12K}$, we have

$$A \leq \sqrt{2T \ln(12K/\delta)}. \tag{30}$$

For term $B$, we also have $\mathbb{E}_{z_{a,t} \sim \mathcal{P}_a}[h_{a,t}] = H_{a,t}$, therefore $B$ is also a sum of a martingale difference sequence with elements' absolute values bounded by 1. We then have $B \leq \sqrt{2T \ln(12K/\delta)}$ with probability at least $1 - \frac{\delta}{12K}$. By taking a union bound twice: once over $A$ and $B$ for each action $a$ and once all $K$ actions, we obtain with probability at least $1 - \frac{\delta}{6}$,

$$A + B \leq 2\sqrt{2T \ln(12K/\delta)}$$

simultaneously for all $a \in [K]$. Furthermore, since

$$T = \sum_{t=1}^{T} 1 \leq \sum_{t=1}^{T} \beta_t$$

due to $1 \leq \beta_t$, we obtain that with probability at least $1 - \frac{\delta}{4}$,

$$\max_{a \in [K]} \text{GroupRegret(a)} \leq O\left(\sqrt{\ln(K/\delta) \sum_{t=1}^{T} \beta_t}\right). \tag{31}$$

By Lemma 3.3, when $\mathcal{E}_{i,\theta}$ holds simultaneously for all $i \in [K]$ and $\theta \in \Theta$, we have

$$R_{A_q} \leq \max_{a \in [K]} \text{GroupRegret(a)}$$

$$\leq O\left(\sqrt{\ln(K/\delta) \sum_{t=1}^{T} \beta_t}\right).$$

$\qquad\square$

**Corollary B.5.** *For any $T \geq 1$, SB-GDRO (Algorithm 1) guarantees that with probability at least $1 - \frac{\delta}{2}$,*

$$\text{err}(\bar{\theta}, \bar{q}) \leq O\left(\frac{(DG + \sqrt{\bar{\beta}_t})\sqrt{\ln(K/\delta)}}{\sqrt{T}}\right) \tag{32}$$

*Proof.* By (Zhang et al., 2023), the duality gap is bounded by the average regret of the two players:

$$\text{err}(\bar{\theta}, \bar{q}) \leq \frac{1}{T}\left(R_{\mathcal{A}_\theta} + R_{\mathcal{A}_q}\right). \tag{33}$$

In Appendix F, we prove that with probability $1 - \delta/4$, the regret of the min-player is bounded by

$$R_{\mathcal{A}_\theta} \leq O\left(DG\sqrt{T \ln(1/\delta)}\right). \tag{34}$$

For the max-player, Lemma B.4 implies that with probability $1 - \delta/4$,

$$R_{\mathcal{A}_q} \leq O\left(\sqrt{\sum_{t=1}^{T} \beta_t \ln(K/\delta)}\right)$$

$$= O\left(\sqrt{T\bar{\beta}_T \ln(K/\delta)}\right) \tag{35}$$

where the equality is from the definition of $\bar{\beta}_T = \frac{\sum_{t=1}^{T} \beta_t}{T}$. Plugging (34) and (35) into (33) and taking a union bound, we obtain that with probability at least $1 - \delta/2$

$$\text{err}(\bar{\theta}, \bar{q}) \leq O\left(\frac{DG\sqrt{\ln(1/\delta)} + \sqrt{\bar{\beta}_T \ln(K/\delta)}}{\sqrt{T}}\right)$$

$$\leq O\left(\frac{(DG + \sqrt{\bar{\beta}_T})\sqrt{\ln(K/\delta)}}{\sqrt{T}}\right).$$

$\qquad\square$

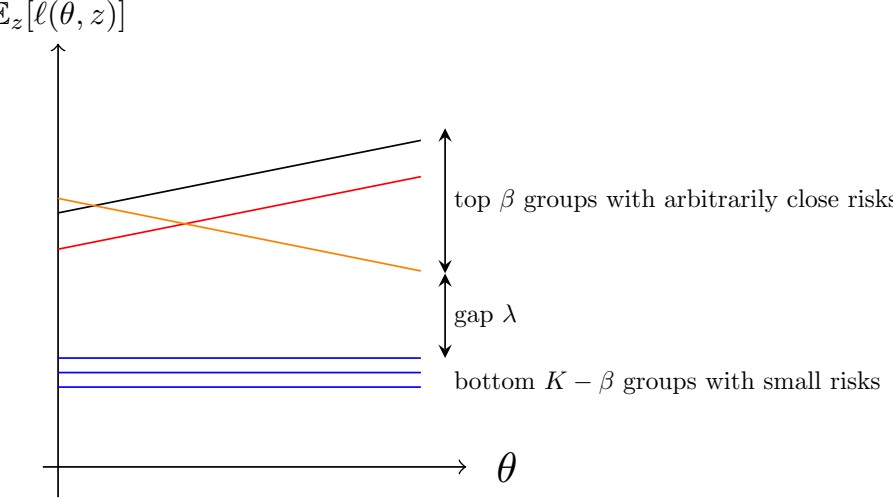

*Figure 4.* The construction for the $\Omega\left(\frac{G^2 D^2 + \beta}{\epsilon^2}\right)$ lower bound.

Corollary B.5 implies $T = O\left(\frac{(D^2 G^2 + \bar{\beta}_T)\ln(K/\delta)}{\epsilon^2}\right)$ is sufficient for a target optimality gap $\epsilon$. This is a self-bounding condition on $T$ since the quantity $\bar{\beta}_T$ is dependent on (and changes with) $T$. Nevertheless, it represents a valid stopping condition because $\bar{\beta}_T$ is fully observable and bounded above by a constant $K$. We are now ready to prove Theorem 3.4.

*Proof (of Theorem 3.4).* In Corollary B.5, by setting the right-hand side to $\epsilon$ and solving for $T$, we obtain that with probability at least $1 - \frac{\delta}{2}$, the number of samples collected during the game for having $\text{err}(\theta, \bar{q}) \leq \epsilon$ is

$$O\left(\frac{(D^2 G^2 + \bar{\beta}_T)\ln(K/\delta)}{\epsilon^2}\right).$$

By Lemma B.1, we collect

$$O\left(\frac{n\ln(GDK/\delta)}{\lambda^2}\right)$$

samples from each group before the game starts so that with probability at least $1 - \frac{\delta}{2}$, we have $\beta_t \leq \beta_\lambda$ simultaneously for all $t \in [T]$. This implies that $\bar{\beta}_T \leq \beta_\lambda$ and thus the bound can be written as

$$O\left(\frac{(D^2 G^2 + \beta_\lambda)\ln(K/\delta)}{\epsilon^2}\right).$$

By taking a union bound, we obtain that with probability at least $1 - \delta$, the total sample complexity is

$$O\left(\frac{Kn(\ln(GDK/\delta))}{\lambda^2} + \frac{(D^2 G^2 + \beta_\lambda)\ln(K/\delta)}{\epsilon^2}\right).$$

$\square$

### B.6. Proof of Theorem 3.5

*Proof.* Our lower bound construction directly extends that of (Soma et al., 2022). In particular, let $\mathcal{Z} = [0,1]^3$ be the set of samples and $\Theta = [0,1]$ be the hypothesis set. The loss of a hypothesis $\theta$ on a sample $z = \begin{bmatrix} z_1 \\ z_2 \\ z_3 \end{bmatrix}$ is

$$\ell(\theta, z) = \delta(z_1\theta + z_2(1-\theta)) + z_3,$$

where $\delta \in (0, 1)$ is a constant defined later.

The distributions of the first $\beta$ groups are similar to that of (Soma et al., 2022), where

- The first $\beta - 1$ distributions are

$$P_i = \begin{cases} z_1 = 0 & \text{almost surely} \\ z_2 = 1 & \text{almost surely} \\ z_3 \sim \text{Bernoulli}(\mu_i), \end{cases}$$

where $\mu_i = \frac{1}{2}$ for $i = 1, 2, \ldots, \beta - 1$.

- The $\beta^{th}$ distribution is

$$P_\beta = \begin{cases} z_1 = 1 & \text{almost surely} \\ z_2 = 0 & \text{almost surely} \\ z_3 \sim \text{Bernoulli}(\mu_\beta), \end{cases}$$

where $\mu_\beta = \frac{1}{2}$.

The last $K - \beta$ distributions are

$$P_i = \begin{cases} z_1 = 0 & \text{almost surely} \\ z_2 = 0 & \text{almost surely} \\ z_3 = \frac{1}{2} - \lambda & \text{almost surely} \end{cases}$$

for $i = \beta + 1, \beta + 2, \ldots, K$. Figure 4 illustrates this construction. The risks of the groups are

$$R_i(\theta) = \mathbb{E}_{z \sim P_i}[\ell(\theta, z)] = \begin{cases} \Delta(1 - \theta) + \mu_i & (i = 1, 2, \ldots, \beta - 1) \\ \Delta\theta + \mu_\beta & (i = \beta) \\ \frac{1}{2} - \lambda & (i = \beta + 1, \beta + 2, \ldots, K). \end{cases}$$

Since $\Delta \geq 0, \theta \in (0, 1)$ and $\mu_i = \frac{1}{2}$ for $i = 1, \ldots, \beta$, we have $R_i(\theta) - R_j(\theta) \geq \lambda$ for any $1 \leq i \leq \beta$ and $\beta + 1 \leq j \leq K$. It follows that the set $[\beta] = \{1, 2, \ldots, \beta\}$ is a $\lambda$-dominant set, and this GDRO instance is $(\lambda, \beta)$-sparse. Because the risk differences between the top $\beta$ groups are upper bounded by

$$|R_1(\theta) - R_\beta(\theta)| = |\Delta(1 - 2\theta)|,$$

which is arbitrarily smaller than $\lambda$, there can be no $\lambda$-dominant sets of size smaller than $\beta$. Thus, we have $\beta_\lambda = \beta$. Moreover, for any $\theta$, its maximal risk is attained on a group within the set $[\beta]$ only. Therefore, the sample complexity of algorithm $\mathcal{A}$ is lower bounded by the total samples drawn from the first $\beta$ groups.

On the other hand, by setting

$$\Delta = O\left(\sqrt{\frac{\beta}{T}}\right),$$

where $T$ is the expected total number of samples drawn by $\mathcal{A}$, the first $\beta$ groups are identical to the groups that give rise to the minimax lower bound in (Soma et al., 2022). It follows that for any algorithm $\mathcal{A}$, there exists a GDRO instance which requires at least

$$\Omega\left(\frac{G^2 D^2 + \beta}{\epsilon^2}\right)$$

samples to find a $\epsilon$-optimal hypothesis. □

---

**Algorithm 6** `SolveOpt`: algorithm for solving $\mathrm{OPT}(C, g)$

---

 **Input:** $\epsilon \in (0, 1), K \geq 3, C > K$, function $g$

 Evaluate $g(1)$

 Initialize $U = 1, L = \sqrt{\frac{C}{C + \frac{g(1)-1}{\epsilon^2}}}, \lambda = 1;$

 **while** $\lambda \geq L$ **do**

  Evaluate $g(\lambda)$

  **if** $f(\lambda) < f(U)$ **then**

   Assign $U \leftarrow \lambda, L \leftarrow \sqrt{\frac{C}{\frac{C}{\lambda^2} + \frac{g(\lambda)-1}{\epsilon^2}}}$

  Update $\lambda = \lambda/5$

 **Return:** $\hat{\lambda} = U$.

---

# C. Proofs for Section 4

*Remark* C.1. For notational simplicity, we use a short-hand notation $f_{C,g}$ for $\mathrm{Cost}_{C,g}^{(GDRO)}$. In other words, we will write

$$f_{C,g}(\lambda) = \frac{C}{\lambda^2} + \frac{g(\lambda)}{\epsilon^2}. \tag{36}$$

We will also drop $C, g$ when it is clear from the context and simply write $f(\lambda)$.

*Remark* C.2. Throughout the proofs for Section 4, some of our bounds contain a $\ln\left(\ln\left(\frac{1}{\epsilon}\right)\right)$ factor. While we will always present this term explicitly the first time they appear in the bounds, for ease of exposition we generally are not pedantic about this term and will treat it as a constant. For example, we will write

$$\ln\left(\frac{K \ln\left(\frac{1}{\epsilon}\right)}{\delta}\right) = \ln\left(\frac{K}{\delta}\right) + \ln\left(\ln\left(\frac{1}{\epsilon}\right)\right)$$

$$= O\left(\ln\left(\frac{K}{\delta}\right)\right),$$

assuming that in practice, the number of arms $K > 1$ is not too small and the failure probability $\delta < 1$ is not too large so that $\frac{K}{\delta} > \ln\left(\frac{1}{\epsilon}\right)$.

## C.1. A Sample-Efficient Approach for Estimating $\lambda_{C,g}^*$

We present an algorithm called `SolveOpt` for solving $\mathrm{OPT}(C, \epsilon, g)$. `SolveOpt` outputs a $\hat{\lambda}$ such that $f(\hat{\lambda}) = O(f(\lambda^*))$ while using at most $O(f(\lambda^*) \ln(K/\delta) \ln(1/\epsilon))$ samples. The significance of this result in the context of GDRO is as follows: by using $\tilde{O}(f(\lambda^*))$ samples to obtain an estimate $\hat{\lambda}$ and then using $\hat{\lambda}$ for GDRO, we guarantee that the total sample complexity is of order $\tilde{O}(f(\lambda^*))$. This implies that without knowing $\lambda^*$, we can achieve a bound with only a logarithmic factor overhead than the bound obtained when $\lambda^*$ is known. Our results and techniques are applicable to other trade-off problems similar to (10), and thus they could be of independent interest.

As mentioned in the main text, `SolveOpt` maintains two variables $U$ and $L$ which specify an interval $[L, U]$ that always contains a good estimate for $\lambda^*$. This $[L, U]$ shrinks over time based on how large $f(\lambda)$ is in comparison to $f(U)$: $U$ is set to $\lambda$ and $L$ is increased accordingly if $f(\lambda) < f(U)$ holds. A crucial element of this process is choosing the geometric sequence $(1, \frac{1}{5}, \frac{1}{25}, \dots)$ of common ratio $\frac{1}{5}$ as the sequence of $\lambda$ at which $g(\lambda)$ is evaluated. The process stops when $\lambda < L$, at which point the algorithm return the last value of $U$ as an estimate for $\lambda^*$. The first key technical insight of this process is that $L$ and $U$ can be computed using only readily known quantities such as $C, K$ and evaluated $g(\lambda)$. The second key technical insight is after some finite number of steps, it is guaranteed that *any* value in the interval $[L, U]$ is a good estimate for $\lambda^*$. The full procedure is given in Algorithm 6. The following lemma states the sample complexity of this approach.

**Theorem C.3.** *For any* $\mathrm{OPT}(C, g)$ *problem defined in* (10)*,* `SolveOpt` *(Algorithm 6) returns a* $\hat{\lambda}$ *such that* $f(\hat{\lambda}) \leq 50 f(\lambda^*)$ *while using at most* $O\left(f(\lambda^*) \ln(K/\delta) \ln(1/\epsilon)\right)$ *samples.*

Before proving Theorem C.3, we note that `SolveOpt` (Algorithm 6) maintains a range of values $[L, U]$ that always contains at least one good estimate for $\lambda^*$, and evaluates $g(\lambda)$ at elements of the geometric series $(U, \frac{U}{5}, \frac{U}{25}, \dots, \frac{U}{5^{\lfloor \log_5\left(\frac{U}{L}\right) \rfloor}})$ to

compute this estimate. Note that all elements of this series are in $[L, U]$. Whenever $f(\lambda)$ is strictly smaller than $f(U)$ for some $\lambda$, we shrink the range $[L, U]$ by setting $U = \lambda$ and $L = \sqrt{\frac{C}{\frac{C}{\lambda^2} + \frac{g(\lambda) - 1}{\epsilon^2}}}$. We first prove the following lemma which shows that $L$ is always smaller than or equal to $\lambda^*$, thus at least one $g(\lambda)$ for $\lambda \leq \lambda^*$ will be evaluated while running `SolveOpt`.

**Lemma C.4.** *For any $U \in (0, 1]$, let*

$$L = \sqrt{\frac{C}{\frac{C}{U^2} + \frac{g(U) - 1}{\epsilon^2}}}.$$

*Then, $L \leq \min\{\lambda^*, U\}$.*

*Proof.* Since $\beta_U \geq 1$, we have $L \leq U$. By definition of $\lambda^*$, we have

$$f(\lambda^*) = \frac{C}{(\lambda^*)^2} + \frac{g(\lambda^*)}{\epsilon^2} \leq f(U) = \frac{C}{U^2} + \frac{g(U)}{\epsilon^2}.$$

Since $g(\lambda^*) \geq 1$, this implies

$$\frac{C}{(\lambda^*)^2} + \frac{1}{\epsilon^2} \leq \frac{C}{U^2} + \frac{g(U)}{\epsilon^2}.$$

Subtracting $\frac{1}{\epsilon^2}$ and dividing $C$ on both sides, we obtain

$$(\lambda^*)^2 \geq \frac{C}{\frac{C}{U^2} + \frac{g(U) - 1}{\epsilon^2}}$$
$$= L^2.$$

We conclude that $L \leq \min\{\lambda^*, U\}$. $\qquad\square$

The next lemma shows that if $\lambda$ falls into the range $[\frac{\lambda^*}{5}, \lambda^*]$ when $f(U)$ is much larger than $f(\lambda^*)$, then the inequality $f(\lambda) < f(U)$ holds.

**Lemma C.5.** *For any $U \in (0, 1]$, if*

$$f(U) > \frac{25C}{(\lambda^*)^2} + \frac{g(\lambda^*)}{\epsilon^2},$$

*then for any $\lambda \in [\frac{\lambda^*}{5}, \lambda^*]$, we have*

$$f(\lambda) < f(U).$$

*Proof.* For any $\lambda \in [\frac{\lambda^*}{5}, \lambda^*]$, we have $\frac{C}{\lambda^2} \leq \frac{25C}{(\lambda^*)^2}$ and $g(\lambda) \leq g(\lambda^*)$. Hence,

$$f(\lambda) = \frac{C}{\lambda^2} + \frac{g(\lambda)}{\epsilon^2}$$
$$\leq \frac{25C}{(\lambda^*)^2} + \frac{g(\lambda^*)}{\epsilon^2}$$
$$< f(U).$$

$\qquad\square$

We need one last lemma, showing that once $U$ is sufficiently close to $\lambda^*$ such that $f(U) = O(f(\lambda^*))$, then any values between $[L, U]$ can be used as an estimate for $\lambda^*$.

**Lemma C.6.** *For any $U \in (0, 1]$, if*

$$f(U) \leq \frac{25C}{(\lambda^*)^2} + \frac{g(\lambda^*)}{\epsilon^2},$$

*then with $L = \sqrt{\frac{C}{\frac{C}{U^2} + \frac{g(U)-1}{\epsilon^2}}}$, we have for any $\lambda \in [L, U]$,*

$$f(\lambda) \leq 50 f(\lambda^*).$$

*Proof.* For any $\lambda \in [L, U]$, we have $\frac{C}{\lambda^2} \leq \frac{C}{L^2}$ and $g(\lambda) \leq g(U)$. Hence,

$$
\begin{aligned}
f(\lambda) &= \frac{C}{\lambda^2} + \frac{g(\lambda)}{\epsilon^2} \\
&\leq \frac{C}{L^2} + \frac{g(U)}{\epsilon^2} \\
&= \frac{C}{U^2} + \frac{g(U)-1}{\epsilon^2} + \frac{g(U)}{\epsilon^2} \qquad \text{since } L = \sqrt{\frac{C}{\frac{C}{U^2} + \frac{g(U)-1}{\epsilon^2}}} \\
&\leq \frac{C}{U^2} + \frac{2g(U)}{\epsilon^2} \\
&\leq 2f(U) \\
&\leq 50 \left( \frac{C}{(\lambda^*)^2} + \frac{g(\lambda^*)}{\epsilon^2} \right) \\
&= 50 f(\lambda^*).
\end{aligned}
$$

$\square$

*Proof (of Theorem C.3).* First, we prove that `SolveOpt` (Algorithm 6) always terminates after a finite number of steps. Observe that during the **while** loop, the sequence of values of $\lambda$ is $(1, \frac{1}{5}, \frac{1}{25}, \dots)$, which is monotonically decreasing. On the other hand, $L$ is non-decreasing from the initial value of $\sqrt{\frac{C}{C + \frac{g(1)-1}{\epsilon^2}}}$. This is because whenever $f(\lambda) \geq f(U)$, the value of $L$ is

$$L = \sqrt{\frac{C}{\frac{C}{U^2} + \frac{g(U)-1}{\epsilon^2}}} = \sqrt{\frac{C}{f(U) - \frac{1}{\epsilon^2}}}.$$

Once the inequality $f(\lambda) < f(U)$ holds, $U$ is assigned to $\lambda$ and $L$ is assigned to a new value $L'$, where

$$L' = \sqrt{\frac{C}{f(\lambda) - \frac{1}{\epsilon^2}}} > \sqrt{\frac{C}{f(U) - \frac{1}{\epsilon^2}}} = L.$$

It follows that the condition $\lambda \geq L$ of the **while** loop must be false after a finite number of steps.

Next, we consider two cases: $f(1) \leq \frac{25C}{(\lambda^*)^2} + \frac{g(\lambda^*)}{\epsilon^2}$ and $f(1) > \frac{25C}{(\lambda^*)^2} + \frac{g(\lambda^*)}{\epsilon^2}$.

**Case 1:** $f(1) \leq \frac{25C}{(\lambda^*)^2} + \frac{g(\lambda^*)}{\epsilon^2}$

In this case, by Lemma C.6, any value in the range $[L, 1]$ where $L = \sqrt{\frac{C}{C + \frac{g(1)-1}{\epsilon^2}}}$ is a good estimate for $\lambda^*$. Because these values are at least $L$ and there are at most $\log_5 \left( \frac{1}{L} \right)$ evaluations, the maximum number of samples needed for testing all

values of $\lambda$ in the sequence $(1, \frac{1}{5}, \frac{1}{5^2}, \ldots, \frac{1}{5^{\lfloor \log_5(1/L) \rfloor}})$ is bounded by

$$
\begin{aligned}
O\left(\frac{C\ln(K/\delta)\log_5\left(\frac{1}{L}\right)}{L^2}\right) &= O\left(\ln(K/\delta)\left(C + \frac{g(1)-1}{\epsilon^2}\right)\log_5\left(\frac{1}{L}\right)\right) \\
&= O\left(\ln(K/\delta)\left(C + \frac{g(1)-1}{\epsilon^2}\right)\log_5\left(\sqrt{1 + \frac{g(1)-1}{C\epsilon^2}}\right)\right) \\
&\leq O\left(\ln(K/\delta)\left(C + \frac{g(1)}{\epsilon^2}\right)\ln\left(1 + \frac{g(1)-1}{C\epsilon^2}\right)\right) \\
&\leq O\left(\ln(K/\delta)\left(C + \frac{g(1)}{\epsilon^2}\right)\ln\left(1 + \frac{1}{\epsilon^2}\right)\right) \\
&\leq O\left(\ln(K/\delta)f(\lambda^*)\ln(1/\epsilon)\right),
\end{aligned}
$$

where the first inequality is from $\log_5(\sqrt{x}) = \frac{\ln(x)}{2\ln(5)} \leq \ln(x)$, the second inequality is due to $g(1) - 1 < K < C$, and the third inequality is from $C + \frac{g(1)}{\epsilon^2} = f(1) \leq 25f(\lambda^*)$ and $\ln\left(1 + \frac{1}{\epsilon^2}\right) \leq \ln\left(\frac{2}{\epsilon^2}\right) = 2\ln\left(\frac{\sqrt{2}}{\epsilon}\right)$.

**Case 2:** $f(1) > \frac{25C}{(\lambda^*)^2} + \frac{g(\lambda^*)}{\epsilon^2}$

In this case, since $f(1) > \frac{24C}{(\lambda^*)^2} + f(\lambda^*) > f(\lambda^*)$, initially, we have $U = 1 > \lambda^*$. By Lemma C.4, we have $\lambda^*$ belongs to the range $[L, 1]$, where $L = \sqrt{\frac{C}{C + \frac{g(1)-1}{\epsilon^2}}}$. As $\lambda$ repeatedly shrinks by $\frac{1}{5}$ from 1, after at most $\log_5(\frac{5}{\lambda^*})$ iterations of the **while** loop, it must fall into the range $[\frac{\lambda^*}{5}, \lambda^*]$. Each of these iterations makes one evaluation $g(\lambda)$ for some $\lambda \geq \lambda^*/5$. Hence, the total number of samples to evaluate $g(\lambda)$ for $\lambda$ from 1 to $\frac{1}{5^{\lfloor \log_5(\frac{5}{\lambda^*}) \rfloor}}$ (i.e., the first element of the geometric series that lies inside the range $[\lambda^*/5, \lambda^*]$), is at most

$$
O\left(\frac{C\ln(K/\delta)\log_5(\frac{5}{\lambda^*})}{(\lambda^*)^2}\right). \tag{37}
$$

Let $U_*$ be the largest value of $\lambda$ being tested for which $f(\lambda) \leq \frac{25C}{\lambda^*} + \frac{g(\lambda^*)}{\epsilon^2}$. By Lemma C.5, $U_* \geq \lambda^*/5$. Note that $U_*$ might be larger than $\lambda^*$. Because the algorithm starts with $f(1) > \frac{25C}{\lambda^*} + \frac{g(\lambda^*)}{\epsilon^2}$, the inequality $f(\lambda) < f(U)$ must be true at $\lambda = U_*$. It follows that $U$ is set to a $U_*$, and $f(U_*) \leq 50f(\lambda^*)$ by Lemma C.6.

Let $L_* = \sqrt{\frac{C}{\frac{C}{U_*^2} + \frac{g(U_*)-1}{\epsilon^2}}}$ be the corresponding value of $L$ after $U$ is assigned to $U^*$. In each of the subsequent iterations, since $L$ is non-decreasing and $U$ is non-increasing, the returned value $\hat{\lambda}$ must be in this range $[L_*, U_*]$. The number of

iterations needed until termination starting from $U_*$ is at most

$$
\begin{aligned}
\log_5\left(\frac{U_*}{L_*}\right) &= \log_5\left(\frac{U_*\sqrt{\frac{C}{U_*^2} + \frac{g(U_*)-1}{\epsilon^2}}}{\sqrt{C}}\right) \\
&\le \log_5\left(\frac{U_*\sqrt{\frac{C}{U_*^2} + \frac{g(U_*)}{\epsilon^2}}}{\sqrt{C}}\right) \\
&= \log_5\left(\sqrt{1 + \frac{g(U_*)U_*^2}{C\epsilon^2}}\right) \\
&\le \log_5\left(\sqrt{1 + \frac{1}{\epsilon^2}}\right) \\
&\le \ln\left(1 + \frac{1}{\epsilon^2}\right) \\
&\le \ln\left(\frac{2}{\epsilon^2}\right),
\end{aligned}
\tag{38}
$$

where the second inequality is from $g(U_*) \le K < C$ and $U_* \le 1$, the third inequality is due to $\log_5(\sqrt{x}) = \frac{1}{2}\log_5(x) = \frac{1}{2}\frac{\ln(x)}{\ln(5)} \le \ln(x)$ for $x > 1$ and the last inequality is $1 + \frac{1}{\epsilon^2} \le \frac{2}{\epsilon^2}$ for $\epsilon \le 1$. In each of these iterations, $\texttt{SolveOpt}$ evaluates $g(\lambda)$ once for $\lambda \ge L_*$. In total, the number of samples in these iterations is at most

$$
\begin{aligned}
O\left(\frac{C\ln(K/\delta)\log_5\left(\frac{U_*}{L_*}\right)}{L_*^2}\right) &= O\left(\ln(K/\delta)\left(\frac{C}{U_*^2} + \frac{g(U_*)-1}{\epsilon^2}\right)\log_5\left(\frac{U_*}{L_*}\right)\right) \\
&\le O\left(\ln(K/\delta)f(U_*)\ln\left(1/\epsilon^2\right)\right) \\
&= O\left(\ln(K/\delta)f(\lambda^*)\ln(1/\epsilon)\right)
\end{aligned}
\tag{39}
$$

where the first inequality is due to (38) and the second inequality is due to $f(\lambda^*) \le f(U_*) \le 50f(\lambda^*)$ and $\ln\left(1/\epsilon^2\right) = 2\ln(1/\epsilon)$. The total number of samples used by Algorithm 6 is the bounded by the sum of the number of samples for testing $\lambda$ from 1 to $U_*$, and then from $U_*$ to $L_*$. Combining (37) and (39), we have the total number of samples needed until Algorithm 6 terminates is at most

$$
\begin{aligned}
O&\left(\frac{C\ln(K/\delta)\log_5(\frac{5}{\lambda^*})}{(\lambda^*)^2} + \frac{C\ln(K/\delta)\log_5\left(\frac{U_*}{L_*}\right)}{L_*^2}\right) \\
&\le O\left(f(\lambda^*)\ln(K/\delta)\ln(1/\epsilon)\right),
\end{aligned}
$$

where the inequality is from $f(\lambda^*) = \frac{C}{(\lambda^*)^2} + \frac{g(\lambda^*)}{\epsilon^2} \ge \frac{C}{(\lambda^*)^2}$. □

## C.2. Proofs for Section 4.1

Let $\mathcal{B}(\theta, r) = \{\theta' \in \Theta : \|\theta - \theta'\|_2 \le r\}$ be a $\ell_2$-ball of radius $r$ centered at $\theta \in \Theta$. In this section, we prove Theorem 4.1 which specifies the sample complexity of $\texttt{SB-GDRO-A}$ (Algorithm 8) for the setting where no $\lambda$ is known beforehand. The most important component of $\texttt{SB-GDRO-A}$ is computing an estimate $\hat{\lambda}$ for the optimal $\lambda^*$ using the algorithm $\texttt{SolveOpt}$ (Algorithm 6). This computation uses the algorithm $\texttt{EstG}$ (Algorithm 7) to compute an estimate of $\beta_\lambda$ for $\lambda$ in the geometric sequence $(1, \frac{1}{5}, \frac{1}{25}, \dots)$ of common ratio $\frac{1}{5}$.

First, we prove the following lemma which bounds the number of $\lambda$ tested in Algorithm 6.

**Lemma C.7.** *For any* $\texttt{OPT}(C, g)$ *problem, in* $\texttt{SolveOpt}$ *(Algorithm 6), the number of values* $\lambda$ *whose* $g(\lambda)$ *need to be evaluated is at most*

$$
N = \ln\left(\frac{2}{\epsilon}\right).
\tag{40}
$$

---

**Algorithm 7** EstG: estimating $g(\lambda)$ for $\lambda$ in the geometric sequence of common ratio $\frac{1}{5}$

---

**Input:** $\lambda \in (0, 1)$

Compute a $\frac{0.1\lambda}{G}$−cover $\widehat{\Theta}$ of $\Theta$ with centers $\hat{\theta}^{(1)}, \hat{\theta}^{(2)}, \ldots, \hat{\theta}^{(|\widehat{\Theta}|)}$

Let $N = \ln\left(\frac{2}{\epsilon}\right)$

Draw $m_N = \frac{384n \ln\left(\frac{741GDKN}{\delta}\right)}{0.01\lambda^2}$ samples from each of the $K$ groups into a set $V_\lambda$

Compute $S^{(i)} = \text{DominantSet}(\hat{\theta}^{(i)}, V_\lambda, 0.7\lambda)$ for $i = 1, 2, \ldots, \left|\widehat{\Theta}\right|$ by Algorithm 3

**Return:** $\hat{\beta}_\lambda = \max_{i=1,2,\ldots,|\widehat{\Theta}|} \left|S^{(i)}\right|$

---

---

**Algorithm 8** SB-GRDRO-A: SB-GDRO without knowing any $\lambda$

---

**Input:** Constants $T, K, D, G > 0, \delta > 0, \epsilon > 0$

Compute $\hat{\lambda} = \text{SolveOpt}(\epsilon, K, \hat{C}, \hat{g})$ by Algorithm 6 where $\hat{g}$ is defined in Equation (45).

Let $\hat{\Theta} = \{\hat{\theta}^{(i)}\}_{i=1,\ldots,|\hat{\Theta}|}$ be the $\frac{0.1\hat{\lambda}}{G}$-cover of $\Theta$ constructed when querying $\hat{\lambda}$ in EstG

Initialize $\theta_1 = \arg\min_{\theta \in \Theta} \|\theta\|_2$

**for** each round $t = 1, \ldots, T$ **do**

    Let $c_t = \arg\min_{i \in |\hat{\Theta}|} \left\|\hat{\theta}^{(i)} - \theta_t\right\|$ be the index of the center in $\widehat{\Theta}$ closest to $\theta_t$

    Let $\hat{S}_{\theta_t}$ be the pre-computed $0.4\hat{\lambda}$-dominant set at $\hat{\theta}^{(c_t)}$

    Compute $q_t = \text{MaxP}(t, \hat{S}_{\theta_t})$ by Algorithm 2

    Draw a group $i_t \sim q_t$ and a sample $z_{i_t, t} \sim \mathcal{P}_{i_t}$

    Compute $\theta_{t+1} = \text{MinP}(\theta_t, z_{i_t, t})$ by Algorithm 4

**Return:** $\bar{\theta}$

---

*Proof.* In SolveOpt, the values of $\lambda$ belong to a geometric sequence of common ratio $\frac{1}{5}$ starting at $1$ and terminating at a value no smaller than $\sqrt{\frac{C}{C + \frac{g(1)-1}{\epsilon^2}}}$, where $C > K \geq g(1)$. Therefore, the number of values in this sequence is at most

$$\log_5\left(\frac{1}{\sqrt{\frac{C}{C + \frac{g(1)-1}{\epsilon^2}}}}\right) = \log_5\left(\sqrt{\frac{C + \frac{g(1)-1}{\epsilon^2}}{C}}\right)$$

$$= \log_5\left(\sqrt{1 - \frac{1}{C\epsilon^2} + \frac{g(1)}{C\epsilon^2}}\right)$$

$$< \log_5\left(\sqrt{1 + \frac{g(1)}{C\epsilon^2}}\right)$$

$$\leq \log_5\left(\sqrt{1 + \frac{1}{\epsilon^2}}\right) \quad \text{since } g(1) < C$$

$$\leq \frac{1}{2}\log_5\left(\frac{4}{\epsilon^2}\right) \quad \text{since } 1 + \frac{1}{\epsilon^2} \leq \frac{4}{\epsilon^2}$$

$$\leq \ln\left(\frac{2}{\epsilon}\right)$$

where the last inequality is due to $\frac{\log_5(x^2)}{2} = \frac{\ln(x)}{\ln(5)} \leq \ln(x)$ for any $x > 0$. $\qquad \square$

Next, we show that any $0.4\lambda$-dominant set $S_{0.4\lambda,\theta}$ at a $\theta \in \Theta$ is also a $0.2\lambda$-dominant set at any $\theta'$ within the Euclidean ball $\mathcal{B}\left(\theta, \frac{0.1\lambda}{G}\right)$.

**Lemma C.8.** *Let $\theta \in \Theta$ and $\lambda \in [0, 1]$. For any $\theta' \in \mathcal{B}\left(\theta, \frac{0.1\lambda}{G}\right)$, any $0.4\lambda$-dominant set $S_{0.4\lambda,\theta}$ at $\theta$ is also a $0.2\lambda$-dominant set at $\theta'$.*

*Proof.* The statement holds trivially if $S_{0.4\lambda,\theta} = [K]$. If $S_{0.4\lambda,\theta} \neq [K]$, for any $\theta' \in \mathcal{B}(\theta, \frac{0.1\lambda}{G})$ and any group $k \in [K]$, we have

$$
\begin{aligned}
|R_k(\theta') - R_k(\theta)| &\leq G\|\theta' - \theta\|_2 \\
&\leq 0.1\lambda,
\end{aligned}
$$

where the first inequality is due to the Lipschitzness of the loss function, and the second inequality is due to $\|\theta' - \theta\|_2 \leq \frac{0.1\lambda}{G}$. It follows that for any $k \in S_{0.4\lambda,\theta}$ and $k' \in [K] \setminus S_{0.4\lambda,\theta}$, we have

$$
\begin{aligned}
R_k(\theta') - R_{k'}(\theta') &\geq R_k(\theta) - 0.1\lambda - (R_{k'}(\theta) + 0.1\lambda) \\
&\geq 0.2\lambda,
\end{aligned}
$$

where the second inequality is due to $R_k(\theta) - R_{k'}(\theta) \geq 0.4\lambda$. This implies that $S_{0.4\lambda,\theta}$ is also a $0.2\lambda$-dominant set at $\theta'$. $\qquad\square$

Using Lemma C.8, we prove the following guarantee of $\mathtt{EstG}$, which is obtained directly from Lemma B.1 and Lemma 3.1 by re-scaling $\delta$ to $\delta/N$.

**Lemma C.9.** *For any input* $\lambda \in [0, 1]$, $\mathtt{EstG}$ *(Algorithm 7) outputs a* $\hat{\beta}_\lambda$ *such that with probability at least* $1 - \frac{\delta}{4N}$, *the following condition hold:*

$$
\beta_{0.2\lambda} \leq \hat{\beta}_\lambda \leq \beta_\lambda. \tag{41}
$$

*Moreover, the number of samples needed to compute* $\hat{\beta}_\lambda$ *is*

$$
\begin{aligned}
Km_N &= \frac{384Kn \ln\left(\frac{741GDK\ln\left(\frac{2}{\epsilon}\right)}{\delta}\right)}{0.01\lambda^2} \\
&= O\left(\frac{Kn\ln\left(\frac{GDK}{\delta}\right)}{\lambda^2}\right)
\end{aligned} \tag{42}
$$

*Proof.* In $\mathtt{EstG}$, for each $g(\lambda)$ being evaluated, the number of samples drawn from each of the $K$ groups is

$$
m_N = \frac{384n \ln\left(\frac{741GDK\ln\left(\frac{2}{\epsilon}\right)}{\delta}\right)}{0.01\lambda^2}. \tag{43}
$$

By Lemma B.1 and Lemma 3.1, this value of $m_N$ is sufficiently large so that with probability at least $1 - \frac{\delta}{4N}$, for all $i = 1, 2, \ldots, |\widehat{\Theta}|$, the set $S^{(i)}$ is a $0.4\lambda$-dominant set at $\hat{\theta}^{(i)}$. Since $|S^{(i)}| \leq \beta_\lambda$ by Lemma 3.1, we have

$$
\hat{\beta}_\lambda = \max_{i=1,2,\ldots,|\widehat{\Theta}|} |S^{(i)}| \leq \beta_\lambda.
$$

Moreover, by Lemma C.8, at any $\theta \in \mathcal{B}(\hat{\theta}^{(i)}, \frac{0.1\lambda}{G})$, $S^{(i)}$ is also a $0.2\lambda$-dominant set at $\theta$. It follows that

$$
|S^{(i)}| \geq \beta_{0.2\lambda,\theta} \tag{44}
$$

where we recall the definition of $\beta_{0.2\lambda,\theta}$ being the size of the smallest $0.2\lambda$-dominant set at $\theta$. Taking the maximum over $i$ on both sides, we obtain

$$
\begin{aligned}
\hat{\beta}_\lambda &= \max_{i\in\{1,2,\ldots,|\hat{\Theta}|\}} |S^{(i)}| \\
&\geq \max_{i\in\{1,2,\ldots,|\hat{\Theta}|\}} \max\left\{\beta_{0.2\lambda,\theta} : \theta \in \mathcal{B}\left(\hat{\theta}^{(i)}, \frac{0.1\lambda}{G}\right)\right\} \\
&= \max_{\theta\in\Theta} \beta_{0.2\lambda,\theta} \\
&= \beta_{0.2\lambda},
\end{aligned}
$$

where the second equality (third line) is due to $\hat{\Theta}$ being a cover of $\Theta$ and the last equality is due to the definition of $\beta_{0.2\lambda}$. We conclude that $\beta_{0.2\lambda} \leq \hat{\beta}_\lambda \leq \beta_\lambda$.

Finally, since $m_N$ samples are drawn from each of $K$ groups, the total number of samples needed to compute $\hat{\beta}_\lambda$ is $Km_N$. $\qquad\square$

Next, we define a function $\hat{g} : [0,1] \to [K]$ as follows.

$$\hat{g}(\lambda) = \begin{cases} 1 & \text{if } \lambda < \frac{1}{5^{\lfloor \log_5(\frac{2}{\epsilon}) \rfloor}} \\ \texttt{EstG}(\lambda) & \text{if } \lambda \in (1, \frac{1}{5}, \frac{1}{25}, \dots, \frac{1}{5^{\lfloor \log_5(\frac{2}{\epsilon}) \rfloor}}) \\ \hat{g}(x) \text{ for } x = \arg\max\{t < \lambda : t \in (1, \frac{1}{5}, \frac{1}{25}, \dots, \frac{1}{5^{\lfloor \log_5(\frac{2}{\epsilon}) \rfloor}})\} & \text{otherwise} \end{cases} \tag{45}$$

In other words, we define $\hat{g}(\lambda) = 1$ for any sufficiently small $\lambda$ that will never be called during $\texttt{SolveOpt}$, which consists of values smaller than $\frac{1}{5^{\lfloor \log_5(\frac{2}{\epsilon}) \rfloor}}$. For any $\lambda \geq \frac{1}{5^{\lfloor \log_5(\frac{2}{\epsilon}) \rfloor}}$, if $\lambda$ that belongs to the geometric sequence $(1, \frac{1}{5}, \frac{1}{25}, \dots, )$, then $\hat{g}(\lambda)$ is the output of $\texttt{EstG}$ with input $\lambda$. Otherwise, $\hat{g}(\lambda)$ is equal to the output $\hat{\beta}_x$ of $\texttt{EstG}$ with input $x = \frac{1}{5^{\lceil \log_5(\frac{1}{\lambda}) \rceil}}$, which is the first value in the geometric sequence that is smaller than $\lambda$. Let

$$\hat{C} = \frac{Kn\ln\left(\frac{GDK\ln\left(\frac{1}{\epsilon}\right)}{\delta}\right)}{\ln(K/\delta)}, \tag{46}$$

and

$$\hat{f}(\lambda) = \frac{\hat{C}}{\lambda^2} + \frac{\hat{g}(\lambda)}{\epsilon^2}, \tag{47}$$

We have $\hat{g}(\lambda) \in [1, K]$ due to the fact that $\texttt{DominantSet}$ always returns a non-empty subset of $[K]$. Moreover, $\hat{C} > K$. The following lemma shows that with high probability, this function $\hat{g}(.)$ is non-decreasing.

**Lemma C.10.** *With probability at least $1 - \frac{\delta}{4}$, the function $\hat{g}$ defined in (45) is non-decreasing.*

*Proof.* Since $\frac{\epsilon}{2} \leq \frac{1}{5^{\lfloor \log_5(\frac{2}{\epsilon}) \rfloor}}$, within the range $[0, \frac{\epsilon}{2}]$ we have $\hat{g}(\lambda) = 1$ which is never larger than any possible returned value by $\texttt{EstG}$. Therefore, we only need show that $\hat{g}(\lambda)$ is non-decreasing for $\lambda > \frac{\epsilon}{2}$. To this end, we will prove that $\hat{g}(\frac{\lambda}{5}) \leq \hat{g}(\lambda)$ for any value $\lambda$ in the truncated geometric sequence $(1, \frac{1}{5}, \frac{1}{25}, \dots, \frac{1}{5^{\lfloor \log_5(\frac{2}{\epsilon}) \rfloor}})$. This trivially holds for the last value $\lambda_{\text{last}} = \frac{1}{5^{\lfloor \log_5(\frac{2}{\epsilon}) \rfloor}}$ in this sequence, since by definition $\hat{g}(\frac{\lambda_{\text{last}}}{5}) = 1$ and the returned value of $\texttt{EstG}$ is always greater than or equal to 1. For other $\lambda$ in this sequence, let $\lambda' = \frac{\lambda}{5} = 0.2\lambda$. Observe that the number of values in this truncated geometric sequence is at most

$$\log_5\left(\frac{2}{\epsilon}\right) \leq \ln\left(\frac{2}{\epsilon}\right) = N,$$

hence we can apply Lemma C.9 and take a union bound (over at most $N$ values of the truncated geometric sequence) to obtain that with probability at least $1 - \frac{\delta}{4}$, we have $\hat{g}(\lambda') = \hat{\beta}_{\lambda'} \leq \beta_{\lambda'} = \beta_{0.2\lambda}$ and $\beta_{0.2\lambda} \leq \hat{\beta}_\lambda$ simultaneously for any $\lambda > \lambda_{\text{last}}$. We conclude that $\hat{g}(\lambda') \leq \beta_{0.2\lambda} \leq \hat{\beta}_\lambda = \hat{g}(\lambda)$ for any $\lambda' = \lambda/5$ and $\lambda \leq 1$ in the geometric sequence $(1, \frac{1}{5}, \frac{1}{25}, \dots)$. Furthermore, this implies that for any pair $(\lambda', \lambda)$ where $\lambda' \leq \lambda$ from this geometric sequence, we have $\hat{g}(\lambda') \leq \hat{g}(\lambda)$.

More generally, for any $0 \leq x < y \leq 1$, we have three possibilities:

- if $y \leq \frac{\epsilon}{2}$, then $g(x) = g(y) = 1$

- if $x \leq \frac{\epsilon}{2} < y$, then $g(x) = 1 \leq g(y)$

- if $\frac{\epsilon}{2} < x$, then

$$
\begin{aligned}
g(x) = g\left(\frac{1}{5^{\lceil \log_5(\frac{1}{x})\rceil}}\right) \\
\leq g\left(\frac{1}{5^{\lceil \log_5(\frac{1}{y})\rceil}}\right) \\
= g(y),
\end{aligned}
$$

In all cases, $g(x) \leq g(y)$. We conclude that the function $g$ is piecewise-constant and non-decreasing. $\qquad\square$

Lemma C.10 indicates that with high probability, the function $\hat{g}$ defined in (45) satisfies the conditions of the optimization problem (10), thus enabling the use of `SolveOpt` (Algorithm 6) and Theorem C.3. From Lemma C.9, taking a union bound over all queried $\lambda$ throughout `SolveOpt` and note that there are at most $N$ such $\lambda$ by Lemma C.7, we immediately obtain the following result.

**Corollary C.11.** *Running `SolveOpt` (Algorithm 6) with $\hat{g}(\lambda)$ defined in (45) guarantees that with probability at least $1 - \delta/2$, simultaneously for all $\lambda$ queried in `SolveOpt`, `EstG` (Algorithm 7) returns a value $\hat{\beta}_\lambda$ such that $\beta_{0.2\lambda} \leq \hat{\beta}_\lambda \leq \beta_\lambda$.*

We are now ready to prove Theorem 4.1.

*Proof (of Theorem 4.1).* Let

$$
\lambda^* = \underset{\lambda \in [0,1]}{\arg\min} \left( \frac{Kn \ln\left(\frac{GDK \ln(1/\epsilon)}{\delta}\right)}{\lambda^2} + \frac{(D^2 G^2 + \beta_\lambda)\ln(K/\delta)}{\epsilon^2} \right). \tag{48}
$$

We run `SolveOpt` for solving $\mathrm{OPT}(\hat{C}, \hat{g})$ and obtain $\hat{\lambda}$ as an estimate for $\lambda^*_{\hat{C}, \hat{g}}$. Theorem C.3 and Corollary C.11 implies that with probability at least $1 - \frac{\delta}{2}$, the returned value $\hat{\lambda}$ is an element of the geometric sequence $(1, \frac{1}{5}, \frac{1}{25}, \dots)$ and satisfies $\hat{f}(\hat{\lambda}) \leq 50 \hat{f}(\lambda^*_{\hat{C}, \hat{g}})$, which is equivalent to

$$
\begin{aligned}
\frac{\hat{C}}{\hat{\lambda}^2} + \frac{\hat{\beta}_{\hat{\lambda}}}{\epsilon^2} = \frac{\hat{C}}{\hat{\lambda}^2} + \frac{\hat{g}(\hat{\lambda})}{\epsilon^2} \\
\leq 50 \left( \frac{\hat{C}}{(\lambda^*_{\hat{C}, \hat{g}})^2} + \frac{\hat{g}(\lambda^*_{\hat{C}, \hat{g}})}{\epsilon^2} \right) \\
\leq 50 \left( \frac{\hat{C}}{(\lambda^*)^2} + \frac{\hat{g}(\lambda^*)}{\epsilon^2} \right) \\
\leq 50 \left( \frac{\hat{C}}{(\lambda^*)^2} + \frac{\beta_{\lambda^*}}{\epsilon^2} \right),
\end{aligned} \tag{49}
$$

where the second inequality is from the definition of $\lambda^*_{\hat{C}, \hat{g}}$, and the last inequality is due to $\hat{g}(\lambda^*) = \hat{g}(x_*) \leq \beta_{x_*} \leq \beta_{\lambda^*}$, where $x_* = \frac{1}{5^{\lceil \log_5(\frac{1}{\lambda^*})\rceil}} \leq \lambda^*$. Moreover, the number of samples needed for running `SolveOpt` is at most

$$
\begin{aligned}
O\left( \hat{f}(\lambda_{\hat{C}, \hat{g}, *}) \ln(K/\delta) \ln(1/\epsilon) \right) \leq O\left( \hat{f}(\lambda^*) \ln(K/\delta) \ln(1/\epsilon) \right) \\
= O\left( \left( \frac{\hat{C}}{(\lambda^*)^2} + \frac{\hat{g}(\lambda^*)}{\epsilon^2} \right) \ln(K/\delta) \ln(1/\epsilon) \right) \\
\leq O\left( \left( \frac{\hat{C}}{(\lambda^*)^2} + \frac{\beta_{\lambda^*}}{\epsilon^2} \right) \ln(K/\delta) \ln(1/\epsilon) \right).
\end{aligned} \tag{50}
$$

---

**Algorithm 9** `SB-GDRO-SA`: adaptive and computationally efficient approach without knowing any $\lambda$

---

**Input:** Constants $K \geq 2, D, G > 0, \delta > 0, \epsilon > 0$

Compute constant $C = \frac{Kn \ln\left(\frac{GDK}{\delta}\right)}{\ln(K/\delta)}$

Initialize $\lambda_1 = 1, L = \epsilon\sqrt{\frac{C}{\ln(K)}}$

Initialize $\theta_1 = \arg\min_{\theta \in \Theta}(\|\theta\|_2)$

Initialize $\delta_1 = \frac{\delta}{2}$, counter $c_1 = 1$

Draw a new set of samples $V_1$ of size $Km_1$, where $m_1 = \frac{384n \ln\left(\frac{741GDK}{\delta_t}\right)}{0.01\lambda_1^2}$.

**for** each round $t = 1, \ldots,$ **do**

    Min-player plays $\theta_t$

    Compute a $0.4\lambda_t$-dominant set $S_t = \text{DominantSet}(\theta_t, V_t, 0.7\lambda_t)$ at $\theta_t$ using Algorithm 3

    **if** $|S_t| > \ln(K)$ and $\lambda_t \geq L$ **then**

        Increase counter $c_{t+1} = c_t + 1$

        Reduce $\lambda_{t+1} \leftarrow \frac{\lambda_t}{2}$

        Reduce $\delta_{t+1} \leftarrow \frac{6\delta_t}{\pi^2 c_{t+1}^2}$

        Draw a new set of samples $V_{t+1}$ of size $Km_t$, where $m_t = \frac{384n \ln\left(\frac{741GDK}{\delta_{t+1}}\right)}{0.01\lambda_{t+1}^2}$

    **else**

        Set $\lambda_{t+1} \leftarrow \lambda_t, V_{t+1} \leftarrow V_t$ and $\delta_{t+1} \leftarrow \delta_t, c_{t+1} \leftarrow c_t$

    Compute $q_t = \text{MaxP}(t, S_t)$

    Draw $i_t \sim q_t$ and $z_{i_t,t} \sim \mathcal{P}_{i_t}$

    Compute $\theta_{t+1} = \text{MinP}(\theta_t, z_{i_t,t})$

**Return:** $\bar{\theta} = \frac{1}{T}\sum_{t=1}^{T} \theta_t$.

---

In each round $t$ of the two-player zero-sum game in `SB-GDRO-A`, the dominant set used by the max-player is taken to be the pre-computed $0.4\hat{\lambda}$-dominant set of the center $c_t$ closest to $\theta_t$, where $c_t \in \{1, 2, \ldots, |\widehat{\Theta}|\}$:

$$c_t = \arg\min_{c=1,2,\ldots,|\widehat{\Theta}|} \left\|\theta_t - \hat{\theta}^{(c)}\right\|.$$

As a result, the sizes of the dominant sets used by the max-player never exceeds $\hat{\beta}_{\hat{\lambda}}$. Together with Corollary B.5, this implies that with probability at least $1 - \delta/2$, the number of samples used by the two-player zero-sum game in `SB-GDRO-A` is

$$O\left(\frac{(D^2G^2 + \hat{\beta}_{\hat{\lambda}})\ln(2K/\delta)}{\epsilon^2}\right) \tag{51}$$

Finally, combining (50) and (51) and taking a union bound, we obtain that with probability at least $1 - \delta$, `SB-GDRO-A` returns an $\epsilon$-optimal hypothesis $\bar{\theta}$ with sample complexity

$$O\left(\left(\frac{Kn\ln\left(GDK\ln\left(\frac{1}{\epsilon}\right)/\delta\right)}{(\lambda^*)^2} + \frac{(D^2G^2 + \beta_{\lambda^*})\ln(K/\delta)}{\epsilon^2}\right)\ln(1/\epsilon)\right) = \tag{52}$$

$$O\left(\left(\frac{C}{(\lambda^*)^2} + \frac{D^2G^2 + \beta_{\lambda^*}}{\epsilon^2}\right)\ln(K/\delta)\ln(1/\epsilon)\right), \tag{53}$$

where $C = \frac{Kn\ln\left(\frac{GDK}{\delta}\right)}{\ln(K/\delta)}$ and we dropped the $\ln(\ln(1/\epsilon))$ term in the final bound for ease of exposition. $\qquad\square$

## C.3. Proofs for Section 4.2

The detailed procedure of the computationally efficient approach `SB-GDRO-SA` is given in Algorithm 9. Similar to `SB-GDRO` (Algorithm 1), `SB-GDRO-SA` uses the two-player zero-sum game framework. The main difference is

that `SB-GDRO-SA` does not assume any input $\lambda$. Instead, it uses $\lambda$ from the geometric sequence $\left(1, \frac{1}{2}, \frac{1}{4}, \dots\right)$. A new value of $\lambda_{t+1}$ in this sequence is used for computing the dominant set in round $t+1$ if *both* of the following conditions hold:

- The size $|S_t|$ of the dominant set in round $t$ is larger than $\ln(K)$

- The value of $\lambda_t$ used in round $t$ is not smaller than $L = \epsilon\sqrt{\frac{C}{\ln(K)}}$.

If at least one of the two conditions does not hold, we set $\lambda_{t+1} = \lambda_t$.

Whenever a new value of $\lambda_t$ is used, i.e., either $t = 1$ or $\lambda_t \neq \lambda_{t-1}$, a new set of samples of size $m$ is drawn from each of $K$ groups. The value of $m$ is set by Lemma B.1 and Lemma 3.1, that is $m_t = \frac{384n\ln\left(\frac{741GDK}{\delta_t}\right)}{0.01\lambda_t^2}$. Here, the failure probability $\delta_t$ is set by a geometric sequence of the form (recall that $\delta$ is the global failure probability of the algorithm)

$$\delta_t = \frac{3\delta}{\pi^2(\sum_{s=2}^t \mathbb{1}\{\lambda_s \neq \lambda_{s-1}\})^2}, \tag{54}$$

so that the total failure probability of computing the dominant sets is bounded by

$$\sum_{s=1}^\infty \delta_s \mathbb{1}\{\lambda_s \neq \lambda_{s-1}\} \leq \frac{3\delta}{\pi^2}\sum_{s=1}^\infty \frac{1}{s^2} \leq \frac{\delta}{2}. \tag{55}$$

Note that we define $\lambda_0 = -1$ by convention, so that $\lambda_s \neq \lambda_{s-1}$ holds for $s = 1$.

We will prove the following theorem, which is more general than Theorem 4.2

**Theorem C.12.** *Let $C = \frac{Kn\ln\left(\frac{GDK}{\delta}\right)}{\ln(K/\delta)}$ and $L = \epsilon\sqrt{\frac{C}{\ln(K)}}$. For any $\epsilon > 0, \delta \in (0,1)$, with probability at least $1 - \delta$,* `SB-GDRO-SA` *(Algorithm 9) returns an $\epsilon$-optimal hypothesis with sample complexity*

$$O\left(\frac{(D^2G^2 + \max(\ln(K), \beta_L))}{\epsilon^2}\ln(K/\delta)\ln\frac{1}{\epsilon}\right) \tag{56}$$

Obviously, Theorem C.12 immediately implies Theorem 4.2 since $\beta_L \leq \beta_{\lambda^*}$ for all $L < \lambda^*$.

Before proving Theorem C.12, we first prove a lemma showing that the sets $S_t$ in all $t = 1, 2, \dots, T$ rounds are indeed $0.4\lambda_t$-dominant sets with probability $1 - \delta/2$.

**Lemma C.13.** *With probability at least $1 - \frac{\delta}{2}$,* `SB-GDRO-SA` *(Algorithm 9) guarantees that for all $t \geq 1$, the set $S_t$ is a $0.4\lambda_t$-dominant set at $\theta_t$.*

*Proof.* Fix a $\lambda$ in the geometric sequence $\left(1, \frac{1}{2}, \frac{1}{4}, \dots\right)$. Let $t$ and $t'$ be the first and last rounds in which $\lambda$ is used for computing the dominant sets, respectively. By Lemma B.1 and Lemma 3.1, $m_t$ is sufficiently large so that with probability at least $1 - \frac{\delta_t}{2}$, for all the rounds from $t$ to $t'$, the set $S_h$ for $h = t, t+1, \dots, t'$ is a $0.4\lambda$-dominant set of $\theta_h$. By construction, $\delta_t = \frac{3\delta}{\pi^2(\sum_{s=2}^t \mathbb{1}\{\lambda_s \neq \lambda_{s-1}\})^2}$. Taking a union bound over all $\lambda$ in the geometric sequence $\left(1, \frac{1}{2}, \frac{1}{4}, \dots\right)$ and using

$$\sum_{s=1}^\infty \frac{1}{s^2} = \frac{\pi^2}{6},$$

we obtain with probability at least $1 - \sum_{s=1}^\infty \delta_s \mathbb{1}\{\lambda_s \neq \lambda_{s-1}\} \geq 1 - \frac{\delta}{2}$, the set $S_t$ is a $0.4\lambda_t$-dominant set at $\theta_t$ for all $t \geq 1$. $\square$

The next technical lemma helps bounding the sum $\sum_{s=1}^T m_s\mathbb{1}\{\lambda_s \neq \lambda_{s-1}\}$.

**Lemma C.14.** *Let $\delta > 0, G \geq 1, D \geq 1, K \geq 1$ and $C = \frac{Kn\ln\left(\frac{GDK}{\delta}\right)}{\ln(K/\delta)}$. For any $x \in (0,1)$, we have*

$$Kn\sum_{s=1}^{\lceil -\log_2(x)\rceil} \ln\left(\pi^2\frac{s^2GDK}{3\delta}\right) \leq 2C\ln(K/\delta)\ln\left(\frac{1}{x}\right) + O\left(Kn\ln\left(\frac{1}{x}\right)\ln\left(\ln\left(\frac{1}{x}\right)\right)\right).$$

*Proof.* Without loss of generality, assume $1/x$ is a power of $e$. We have

$$\sum_{s=1}^{\lceil -\log_2(x) \rceil} \ln\left(\frac{\pi^2 s^2 GDK}{3\delta}\right) \leq \sum_{s=1}^{-\ln(x)} \left(\ln\left(\frac{GDK}{\delta}\right) + \ln\left(\frac{\pi^2}{3}\right) + \ln\left(s^2\right)\right)$$

$$\leq 2\ln\left(\frac{GDK}{\delta}\right)\ln\left(\frac{1}{x}\right) + \ln\left(\prod_{s=1}^{-\ln(x)} s^2\right) \tag{57}$$

$$= 2C\ln(K/\delta)\left(\frac{1}{x}\right) + O\left(\ln\left(\frac{1}{x}\right)\ln\left(\ln\left(\frac{1}{x}\right)\right)\right),$$

where the inequalities are from $\log_2(1/x) \leq \ln(1/x)$ and $\ln(n!) = O(n\ln(n))$. Multiplying $Kn$ to both sides leads to the desired statement. $\qquad\square$

We are now ready to prove Theorem C.12.

*Proof (of Theorem C.12).* Let $\lambda_{\ln K}$ be the largest $\lambda$ such that $\beta_\lambda = \ln(K)$. If no such $\lambda$ exists, we set $\lambda_{\ln K} = 0$. Let $\bar{\lambda} = \max(L, \lambda_{\ln(K)})$. Note that $\bar{\lambda} \geq L = \epsilon\sqrt{\frac{C}{\ln(K)}} > \epsilon$ since $C > K > \ln(K)$.

In the worst case, Algorithm 9 draw a new set of samples until $\frac{\bar{\lambda}}{2} \leq \lambda_t \leq \bar{\lambda}$. Without loss of generality, we can assume $\bar{\lambda} < \frac{1}{4}$. Otherwise, Algorithm 9 draws only three sets of samples and stops doing so immediately after some $\lambda_t \geq \frac{1}{4}$, which trivially leads to a sample complexity of $O\left(\frac{G^2 D^2 + \ln(K)}{\epsilon^2}\right)$.

With $\bar{\lambda} < \frac{1}{4}$, the total number of samples of used for computing the dominant sets in Algorithm 9 are

$$\sum_{t=1}^{T} \mathbb{1}\{\lambda_t \neq \lambda_{t-1}\}\frac{384Kn\ln(741GDK/\delta_t)}{0.01\lambda_t^2} \leq O\left(\frac{Kn}{\bar{\lambda}^2}\sum_{s=1}^{-\log_2(\bar{\lambda})}\ln\left(\frac{2^s GDK}{\delta_1}\right)\right)$$

$$\leq O\left(\frac{C}{\bar{\lambda}^2}\ln(K/\delta)\ln(1/\bar{\lambda})\right) \tag{58}$$

$$\leq O\left(\frac{C}{\bar{\lambda}^2}\ln(K/\delta)\ln(1/\epsilon)\right),$$

where the second inequality is from Lemma C.14 and the last inequality is from $\bar{\lambda} > \epsilon$. Note that we dropped the $\ln\left(\ln\left(\frac{1}{\epsilon}\right)\right)$ for ease of exposition.

Next, we bound the regret bound of the max-player. Let $\hat{\beta} = \max(\ln(K), \beta_L)$. We show the average $\bar{\beta}_T = \frac{1}{T}\sum_{t=1}^{T}|S_t|$ is not much larger than $\hat{\beta}$.

$$\frac{1}{T}\sum_{t=1}^{T}|S_t| = \frac{1}{T}\sum_{t=1}^{T}\left(\mathbb{1}\{|S_t| > \hat{\beta}\} + \mathbb{1}\{|S_t| \leq \hat{\beta}\}\right)|S_t|$$

$$\leq \frac{1}{T}\sum_{t=1}^{T}\mathbb{1}\{|S_t| > \hat{\beta}\}K + \mathbb{1}\{|S_t| < \hat{\beta}\}\hat{\beta}$$

$$\leq \frac{1}{T}\left(K\log_2\left(\frac{1}{\bar{\lambda}}\right) + \hat{\beta}T\right) \tag{59}$$

$$\leq \frac{1}{T}\left(2\hat{\beta}T\right)\log_2(1/\bar{\lambda})$$

$$\leq 2\hat{\beta}\ln(1/\epsilon),$$

where the first inequality is from $|S_t| < K$ for all $t$, the second inequality is because there are at most $\log_2(\frac{1}{\bar{\lambda}})$ rounds where $|S_t| > \hat{\beta}$, the third inequality is from $K < \hat{\beta}T$ as $\hat{\beta} \geq 1$, and the last inequality is from $\log_2(1/\bar{\lambda}) \leq \ln(1/\bar{\lambda})$ as $\bar{\lambda} > \epsilon$.

---

**Algorithm 10** `SB-GDRO-DF`: Dimension-free `SB-GDRO` Algorithm with known $(\lambda, \beta)$

---

**Input:** Constants $K, D, G, \eta_w, \lambda, \beta, \epsilon > 0$, hypothesis set $\Theta \subset \mathbb{R}^n$

Compute $T = O(\frac{(D^2 G^2 + \beta) \ln(K/\delta)}{\epsilon^2})$

Compute the maximum length of each episode $\sigma = \left\lfloor \frac{0.1\lambda}{\eta_w G^2} \right\rfloor$

Initialize an episode counter $\rho = 1$

Compute $m' = \frac{24 \ln\left(\frac{4KT}{\sigma\delta}\right)}{\lambda^2}$

Draw $m'$ samples from each $K$ groups into set $V^1$

Initialize $\theta_1 = \arg\min_{\theta \in \Theta} \|\theta\|_2$

Compute a dominant set $S^1 = \mathtt{DominantSet}(\theta_1, V^1, \lambda)$ at $\theta_1$ by Algorithm 3

Let $\hat{S}_{\theta_1} = S^1$

Compute $q_1 = \mathtt{MaxP}(\theta_1, \hat{S}_{\theta_1})$ by Algorithm 2

**for** each round $t = 1, \ldots, T$ **do**

    Draw a group $i_t \sim q_t$ and a sample $z_{i_t,t} \sim \mathcal{P}_{i_t}$

    Compute $\theta_{t+1} = \mathtt{MinP}(\theta_t, z_{i_t,t})$ by Algorithm 4

    **if** $t$ is divisible by $\sigma$ **then**

        Increase episode counter $\rho \leftarrow \rho + 1$

        Draw new $m'$ samples from each of $K$ groups into $V^\rho$.

        Compute a dominant set $S^\rho = \mathtt{DominantSet}(\theta_{t+1}, V^\rho, \lambda)$ at $\theta_{t+1}$ by Algorithm 3

    Let $\hat{S}_{\theta_{t+1}} = S^\rho$

    Compute $q_{t+1} = \mathtt{MaxP}(\theta_{t+1}, \hat{S}_{\theta_{t+1}})$ by Algorithm 2 using the last computed $S^\rho$

**Return:** $\bar{\theta} = \frac{1}{T} \sum_{t=1}^{T} \theta_t$ and $\bar{q} = \frac{1}{T} \sum_{t=1}^{T} q_t$

---

Combining this with (35), the regret of the max-player is bounded by

$$
\begin{aligned}
R_{\mathcal{A}_q} &\leq O\left( \sqrt{T \bar{\beta}_T \ln(K/\delta)} \right) \\
&= O\left( \sqrt{T \hat{\beta} \ln(K/\delta) \ln(1/\epsilon)} \right).
\end{aligned}
\tag{60}
$$

Plugging (60) into (33) and combining with (58), we have the total amount of samples to get an $\epsilon$-optimal hypothesis is

$$
O\left( \frac{C}{(\bar{\lambda})} \ln(1/\epsilon)^2 \right) + O\left( \frac{(G^2 D^2 + \hat{\beta}) \ln(K/\delta)}{\epsilon^2} \ln(1/\epsilon) \right) \leq
\tag{61}
$$

$$
O\left( \left( \left( \min\left\{ \frac{\ln(K)}{\epsilon^2}, \frac{C}{\lambda_{\ln(K)}^2} \right\} + \frac{(D^2 G^2 + \max(\ln(K), \beta_L))}{\epsilon^2} \right) \ln(K/\delta) \left( \ln \frac{1}{\epsilon} \right) \right) \right)
\tag{62}
$$

where the inequality is from $\bar{\lambda} = \max(\lambda_{\ln(K)}, L)$ and $\frac{C}{L^2} = \frac{\ln(K)}{\epsilon^2}$, thus

$$
\frac{C}{(\bar{\lambda})^2} \leq \min\left\{ \frac{C}{L^2}, \frac{C}{\lambda_{\ln(K)}^2} \right\} = \min\left\{ \frac{\ln(K)}{\epsilon^2}, \frac{C}{\lambda_{\ln(K)}^2} \right\}.
$$

Since $\min\left\{ \frac{\ln(K)}{\epsilon^2}, \frac{C}{\lambda_{\ln(K)}^2} \right\} \leq \frac{\ln(K)}{\epsilon^2} \leq \frac{\max(\ln(K), \beta_L)}{\epsilon^2}$, the final bound can be simplified to $O\left( \frac{(D^2 G^2 + \max(\ln(K), \beta_L))}{\epsilon^2} \ln(K/\delta) \ln \frac{1}{\epsilon} \right)$. $\qquad \square$

## D. A Completely Dimension-Independent Approach

First, we give a detailed description of `SB-GDRO-DF` (Algorithm 10) and prove its sample complexity bound in Theorem 5.1. Essentially, `SB-GDRO-DF` also uses the two-player zero-sum game framework similar to `SB-GDRO`. Note that since $\beta$

is known, we can compute the number of rounds $T = O(\frac{(G^2D^2+\beta)\ln(K/\delta)}{\epsilon^2})$ before the game starts. Unlike the previous algorithms, knowing $T$ before the game starts allows us to use a fixed learning rate

$$\eta_{w,t} = \eta_t = \frac{2D}{G\sqrt{T}} \tag{63}$$

for the min-player in Algorithm 10. Another difference is that `SB-GDRO-DF` proceeds in episodes, each consists of multiple consecutive rounds, and the max-player uses the same dominant set for the rounds within each episode. More concretely, in `SB-GDRO-DF`, the $T$ rounds of the game are divided into $\lceil\frac{T}{\sigma}\rceil$ episodes, each is of length $\sigma$, except for the last episode which may have fewer than $\sigma$ rounds if $T$ is not divisible by $\sigma$. The value $\sigma$ is defined as follows:

$$\sigma = \left\lfloor \frac{0.1\lambda}{\eta_w G^2} \right\rfloor. \tag{64}$$

By this construction, the first episode contains rounds $(1, 2, \ldots, \sigma)$, the second episode contains rounds $(\sigma+1, \ldots, 2\sigma)$ and so on, until the last episode which contains rounds $(\lfloor\frac{T}{\sigma}\rfloor\sigma + 1, \ldots, T)$. Let $\rho = 1, 2, \ldots, \lceil\frac{T}{\sigma}\rceil$ be the running index of the episodes. Within an episode $\rho$,

- Before the first round of this episode, a set $V^\rho$ of $Km'$ samples are drawn from the $K$ groups, where $m'$ i.i.d samples are drawn from each group. The value of $m'$ is $\frac{24\ln\left(\frac{4KT}{\sigma\delta}\right)}{\lambda^2}$.

- Let $t^\rho$ be the index of the first round in episode $\rho$ and $\theta^\rho = \theta_{t^\rho}$ be either the initial hypothesis (if $\rho = 1$) or the hypothesis played by the min-player using the algorithm `MinP` (Algorithm 4) (if $\rho > 1$) in round $t^\rho$. A $0.4\lambda$-dominant set $S^\rho$ is computed using `DominantSet` (Algorithm 3) with input $\theta^\rho$ and $V^\rho$.

- In rounds $t \in (t^\rho, t^\rho + 1, \ldots, \min\{t^\rho + \sigma, T\})$ of this episode, the max-player plays $q_t$ using the algorithm `MaxP` (Algorithm 2) with the same input $S^\rho$. Then, a group $i_t \sim q_t$ is drawn and a sample $z_{i_t,t} \sim \mathcal{P}_{i_t}$ is drawn from group $i_t$. The min-player then follows the `MinP` strategy (Algorithm 4) with input $\theta_t$ and $z_{i_t}$ to compute $\theta_{t+1}$.

The algorithm returns $\bar\theta = \frac{1}{T}\sum_{t=1}^T \theta_t$ after $T$ rounds. The following lemma shows that for any episode $\rho$, with high probability, $S^\rho$ is a $0.4\lambda$-dominant set at $\theta^\rho$.

**Lemma D.1.** *At the beginning of episode $\rho$ in `SB-GDRO-DF`, with probability at least $1 - \frac{\sigma\delta}{2T}$, the set $S^\rho$ is a $0.4\lambda$-dominant set at $\theta^\rho$.*

*Proof.* In each episode $\rho$, we draw $m' = \frac{24\ln\left(\frac{4KT}{\sigma\delta}\right)}{\lambda^2}$ samples from each group. By Hoeffding's inequality, for each group $k$, we have

$$\Pr_{V_k^\rho}\left(\frac{1}{m}\left|\sum_{j=1}^{m'}\ell(\theta^\rho, V_{k,j}^\rho) - R_k(\theta^\rho)\right| \geq 0.15\lambda\right) \leq 2\exp\left(-0.045\lambda^2 m'\right)$$

$$= 2\exp\left(-1.08\ln\left(\frac{4KT}{\sigma\delta}\right)\right)$$

$$\leq 2\exp\left(-\ln\left(\frac{4KT}{\sigma\delta}\right)\right)$$

$$= \frac{\sigma\delta}{2KT}.$$

By taking a union bound over $K$ groups, we have

$$\left|\frac{1}{m}\sum_{j=1}^{m'}\ell(\theta^\rho, V_{k,j}^\rho) - R_k(\theta^\rho)\right| \leq 0.15\lambda \tag{65}$$

holds simultaneously for all $k \in [K]$ with probability at least $1 - \frac{\sigma\delta}{2T}$. The condition (65) of $V^\rho$ is the same as the event $\mathcal{E}_{k,\theta^\rho}$ in (19) of $V$ in Lemma B.1. Hence, we can apply Lemma 3.1 and conclude that with probability at least $1 - \frac{\sigma\delta}{2T}$, the set $S^\rho$ is a $0.4\lambda$-dominant set at $\theta^\rho$. □

The next lemma shows that the set $S^\rho$ is a dominant set not only at $\theta^\rho$ but also at the hypotheses within the episode $\rho$.

**Lemma D.2.** `SB-GDRO-DF` *guarantees that if $S$ is a $0.4\lambda$-dominant set at $\theta_t$ for some $t \in [T]$, then for any non-negative integer $\sigma' \leq \min\left\{\left\lfloor \frac{0.1\lambda}{\eta_w G^2} \right\rfloor, T-t\right\}$, $S$ is also a $0.2\lambda$-dominant set at $\theta_{t+\sigma'}$.*

*Proof.* `SB-GDRO-DF` uses the update rule (5) to compute $\theta_{t+1}$. This update rule can be written as follows:

$$\theta_{t+1} = \underset{\theta \in \Theta}{\arg\min}\{2\langle \eta_w \tilde{g}_t, \theta - \theta_t\rangle + \|\theta - \theta_t\|^2 + \eta_w^2\|\tilde{g}_t\|^2\}$$
$$= \underset{\theta \in \Theta}{\arg\min}\{\|\theta_t - \eta_w \tilde{g}_t - \theta\|^2\}$$

which is equivalent to projecting $\theta_t - \eta_w \tilde{g}_t$ onto the convex set $\Theta$. By properties of projection onto convex sets (see e.g. Orabona, 2019, Proposition 2.11), for any $1 \leq t < T$, we have

$$\begin{aligned}
\|\theta_{t+1} - \theta_t\| &\leq \|(\theta_t - \eta_w\tilde{g}_t) - \theta_t\| \\
&= \eta_w\|\tilde{g}_t\| \\
&\leq \eta_w G,
\end{aligned} \tag{66}$$

where the last inequality is $\|\tilde{g}_t\| \leq G$ by the Lipschitzness of the loss function $\ell$. Combining (66) and triangle inequality, we obtain

$$\begin{aligned}
\|\theta_{t+\sigma'} - \theta_t\| &\leq \|\theta_{t+\sigma'} - \theta_{t+\sigma'-1}\| + \|\theta_{t+\sigma'-1} - \theta_t\| \\
&\leq \underbrace{\|\theta_{t+\sigma'} - \theta_{t+\sigma'-1}\| + \|\theta_{t+\sigma'-1} - \theta_{t+\sigma'-2}\| + \cdots + \|\theta_{t+1} - \theta_t\|}_{\sigma' \text{ elements}} \\
&\leq \sigma'\eta_w G \\
&\leq \frac{0.1\lambda}{G},
\end{aligned}$$

where the last inequality is due to $\sigma' \leq \lfloor \frac{0.1\lambda}{\eta_w G^2}\rfloor \leq \frac{0.1\lambda}{\eta_w G^2}$. This implies that $\theta_{t+\sigma'} \in \mathcal{B}(\theta_t, \frac{0.1\lambda}{G})$. By Lemma C.8, it follows that if a set is $0.4\lambda$-dominant at $\theta_t$, then it is also a $0.2\lambda$-dominant set at the hypotheses $\theta_{t+1}, \theta_{t+2}, \ldots, \theta_{t+\sigma}$ played in $\sigma$ subsequent rounds of the game. $\square$

Finally, we show the proof of Theorem 5.1.

*Proof (of Theorem 5.1).* Since the maximum number of rounds in each episode is $\sigma \leq \frac{0.1\lambda}{\eta_w G^2}$, there are at most $\frac{T}{\sigma}$ episodes. Combining Lemma D.1, Lemma D.2 and taking a union bound over $\frac{T}{\sigma}$ episodes, in total we draw

$$O\left(\frac{\eta_w KTG^2 \ln\left(\frac{KT}{\sigma\delta}\right)}{\lambda^3}\right) \tag{67}$$

samples over $\frac{T}{\sigma}$ episodes to guarantee that with probability at least $1 - \delta/2$, all the computed sets over $\frac{T}{\sigma}$ episodes are dominant sets at $(\theta_t)_{t=1,2,\ldots,T}$ with sizes no larger than $\beta_{0.4\lambda}$. Plugging $\eta_w = \frac{2D}{G\sqrt{T}}$ and $\sigma = \frac{0.1\lambda}{\eta_w G^2} = \frac{0.1\lambda\sqrt{T}}{2DG}$ into (67), we obtain a sample complexity of order

$$O\left(\frac{\eta_w KTG^2 \ln\left(\frac{KT}{\sigma\delta}\right)}{\lambda^3}\right) = O\left(\frac{DKG\sqrt{T}\ln\left(\frac{KDG\sqrt{T}}{\lambda\delta}\right)}{\lambda^3}\right). \tag{68}$$

From Corollary B.5, we have $T = O(\frac{(D^2G^2+\beta)\ln(K/\delta)}{\epsilon^2})$ is sufficient for obtaining an $\epsilon$-optimal hypothesis with probability at least $1 - \frac{\delta}{2}$. By plugging $T = O(\frac{(D^2G^2+\beta)\ln(K/\delta)}{\epsilon^2})$ into (68), we obtain the number of samples collected for computing the dominant sets over $\frac{T}{\sigma}$ episodes is

$$O\left(\frac{DKG\sqrt{(D^2G^2+\beta)\ln(K/\delta)}\ln\left(\frac{KDG}{\epsilon\lambda\delta}\right)}{\lambda^3\epsilon}\right). \tag{69}$$

**Algorithm 11** `FTARL`: Follow the regularized and active leader with $\alpha$-Tsallis entropy regularizer and time-varying learning rates for sleeping bandits

---

**Input:** $K \geq 2$, $\alpha$-Tsallis entropy function $\psi(x) = \frac{1 - \sum_{i=1}^{K} x_i^\alpha}{1-\alpha}$

Initialize $\tilde{L}_{i,0} = 0$ for all arms $i \in [K]$.

**for** each round $t = 1, \ldots,$ **do**

    The non-oblivious adversary selects and reveals $\mathbb{A}_t$

    Compute $q_t = \arg\min_{q \in \Delta_K} \psi_t(q) + \langle q, \tilde{L}_{t-1} \rangle$

    Compute $p_{i,t} = \frac{I_{i,t} q_{i,t}}{\sum_{j=1}^{K} I_{j,t} q_{j,t}}$ by Equation (26)

    Draw arm $i_t \sim p_t$ and observe $\hat{\ell}_t = \ell_{i_t,t}$

    **for** each arm $i \in [K]$ **do**

        If $I_{i,t} = 1$, compute $\tilde{\ell}_{i,t} = \frac{\mathbb{1}\{i_t = i\}\hat{\ell}_t}{p_{i,t} + \gamma_t}$ by Equation (27)

        If $I_{i,t} = 0$, compute $\tilde{\ell}_{i,t} = \hat{\ell}_t - \gamma_t \sum_{j \in \mathbb{A}_t} \tilde{\ell}_{j,t}$ by Equation (28)

        Update $\tilde{L}_{i,t} = \tilde{L}_{i,t-1} + \tilde{\ell}_{i,t}$

---

In addition, each of the $T$ rounds uses exactly one sample to compute the outputs of the two players in the next round. Hence, with probability at least $1 - \delta$, the total sample complexity of the two-player zero-sum game needed to return an $\epsilon$-optimal hypothesis is of order

$$O\left( \frac{DKG\sqrt{(D^2G^2 + \beta)\ln(K/\delta)}\ln\left(\frac{KDG}{\epsilon\lambda\delta}\right)}{\lambda^3\epsilon} + \frac{(D^2G^2 + \beta)\ln(K/\delta)}{\epsilon^2} \right).$$

$\square$

## E. FTARL with Time-Varying Learning Rates

We consider a variant of the FTARL algorithm in (Nguyen & Mehta, 2024) with time-varying learning rates. The procedure is given in Algorithm 11. The only difference between this algorithm and the `FTARLShannon` algorithm (Algorithm 5) is that Algorithm 11 uses the $\alpha$-Tsallis entropy regularizer to compute the weight $q_t$ as follows:

$$q_t = \arg\min_{q \in \Delta_K} \psi_t(q) + \langle q, \tilde{L}_{t-1} \rangle, \tag{70}$$

where $\psi_t(q) = \frac{\psi(q) - \min_{v \in \Delta_K} \psi(v)}{\eta_t}$ for $\psi(q) = \frac{1 - \sum_{i=1}^{K} q_i^\alpha}{1-\alpha}$ and $\alpha \in (0,1)$ is a constant. The computation of the sampling probability $p_t$ and the loss estimates of active and non-active arms $\tilde{\ell}_{i,t}$ are identical to that of `FTARLShannon`. Since the $\alpha$-Tsallis entropy tends to Shannon entropy when $\alpha \to 1$ (see e.g. Nielsen & Nock, 2011), we will prove the following high-probability per-action regret bound of Algorithm 11 and then take the limit $\alpha \to 1$ to obtain Theorem B.3.

**Theorem E.1.** *Let $(\eta_t)_{t=1,\ldots}$ and $(\gamma_t)_{t=1,\ldots}$ be two sequences of non-increasing learning rates and exploration factors such that $\eta_t \leq 2\gamma_t$. With probability at least $1 - \delta$, FTARL (Algorithm 11) guarantees that*

$$\max_{a \in [K]} \mathrm{Regret}(a) \leq \frac{K^{1-\alpha} - 1}{\eta_T(1-\alpha)} + \frac{\ln(3K/\delta)}{2\gamma_T} + \left(\frac{1}{2\alpha} + \frac{1}{2}\right)\ln(3/\delta) + \sum_{t=1}^{T} \left(\frac{\eta_t}{2\alpha} + \gamma_t\right) A_t$$

Before proving Theorem E.1, similar to (Nguyen & Mehta, 2024), we state the following results on the concentration bound of the IX-loss estimator. These results are adapted from Neu (2015, Lemma 1 and Corollary 1) in the non-sleeping bandits setting to the sleeping bandits setting with nearly identical proofs.

**Lemma E.2** (Lemma 1 of (Neu, 2015))**.** *Let $(\nu_{i,t})$ be non-negative random variables satisfying $\nu_{i,t} \leq 2\gamma_t$ for all $i \in [K]$ and $t \geq 1$. With probability at least $1 - \delta'$,*

$$\sum_{t=1}^{T} \sum_{i=1}^{K} \nu_{i,t} \mathbb{1}\{I_{i,t} > 0\}(\tilde{\ell}_{i,t} - \ell_{i,t}) \leq \ln(1/\delta').$$

Since the sequence $(\gamma_t)_{t=1,\ldots}$ is non-increasing, we have $\gamma_T \leq \gamma_t$ for all $t \leq T$. Hence, for any fixed arm $a \in [K]$, we can apply Lemma E.2 with $\nu_{i,t} = 2\gamma_T \mathbb{1}\{i = a\} \leq 2\gamma_t$ and take a union bound over $K$ arms to obtain the following corollary.

**Corollary E.3.** *With probability at least $1 - \delta'$, simultaneously for all $a \in [K]$,*

$$\sum_{t=1}^{T} I_{a,t}(\tilde{\ell}_{a,t} - \ell_{a,t}) \leq \frac{\ln(K/\delta')}{2\gamma_T}$$

We turn to the proof of Theorem E.1.

*Proof (of Theorem E.1).* Fix an arm $a \in [K]$ and let $e_a$ be the $a$-th standard basis vector of $\mathbb{R}^K$. Let $\tilde{\ell}_t = \begin{bmatrix} \tilde{\ell}_{1,t} \\ \tilde{\ell}_{2,t} \\ \cdots \\ \tilde{\ell}_{K,t} \end{bmatrix}$ be the

vector of estimated losses of $K$ arms in round $t$. Since the sequence of learning rates is non-increasing and positive, we have $\psi_t(x) \geq 0$ and $\psi_{t+1}(x) \geq \psi_t(x)$ for all $x \in \Delta_K$. Hence, we can invoke the standard local-norm analysis of FTRL with Tsallis entropy regularizer (e.g. Orabona, 2019, Lemma 7.14) on non-negative loss estimates $(\tilde{L}_t)_{t=1,\ldots}$, to obtain

$$\sum_{t=1}^{T} \langle \tilde{\ell}_t, q_t - e_a \rangle \leq \psi_{T+1}(e_a) - \min_{x \in \Delta_K} \psi_1(x) + \frac{1}{2\alpha} \sum_{t=1}^{T} \eta_t \sum_{i=1}^{K} \tilde{\ell}_{i,t}^2 q_{i,t}^{2-\alpha} \tag{71}$$

Following the proof of Nguyen & Mehta (2024, Lemma 26) and by definition of $\tilde{\ell}_t$, we obtain

$$
\begin{aligned}
\langle \tilde{\ell}_t, q_t \rangle &= \sum_{i \in \mathbb{A}_t} \tilde{\ell}_{i,t} q_{i,t} + \sum_{i \notin \mathbb{A}_t} \tilde{\ell}_{i,t} q_{i,t} \\
&= \tilde{\ell}_{i_t,t} q_{i_t,t} + \sum_{i \notin \mathbb{A}_t} \tilde{\ell}_{i,t} q_{i,t} \\
&= \tilde{\ell}_{i_t,t} q_{i_t,t} + \left( \hat{\ell}_t - \gamma_t \sum_{j \in \mathbb{A}_t} \tilde{\ell}_{j,t} \right) \sum_{i \notin \mathbb{A}_t} q_{i,t} \\
&= \tilde{\ell}_{i_t,t} p_{i_t,t} \sum_{i \in \mathbb{A}_t} q_{i,t} + \left( \hat{\ell}_t - \gamma_t \sum_{j \in \mathbb{A}_t} \tilde{\ell}_{j,t} \right) \sum_{i \notin \mathbb{A}_t} q_{i,t} \\
&= \left( \hat{\ell}_t - \gamma_t \sum_{j \in \mathbb{A}_t} \tilde{\ell}_{j,t} \right) \sum_{i=1}^{K} q_{i,t} \\
&= \hat{\ell}_t - \gamma_t \sum_{j \in \mathbb{A}_t} \tilde{\ell}_{j,t},
\end{aligned}
$$

where the second equality is due to $\tilde{\ell}_{i,t} = 0$ for $i \in \mathbb{A}_t, i \neq i_t$, the third equality is due to $\tilde{\ell}_{i,t} = \hat{\ell}_t - \gamma_t \sum_{j \in \mathbb{A}_t} \tilde{\ell}_{j,t}$ for all non-active arms $i \notin \mathbb{A}_t$, the second-to-last equality is due to

$$\tilde{\ell}_{i_t,t} p_{i_t,t} = \frac{p_{i_t,t} \hat{\ell}_t}{p_{i_t,t} + \gamma_t} = \hat{\ell}_t - \frac{\gamma_t \hat{\ell}_t}{p_{i_t,t} + \gamma_t} = \hat{\ell}_t - \gamma_t \tilde{\ell}_{i_t,t} = \hat{\ell}_t - \gamma_t \sum_{j \in \mathbb{A}_t} \tilde{\ell}_{j,t},$$

and the last equality is due to $q \in \Delta_K$. Plugging this into (71) and using $\langle \tilde{\ell}_t, e_a \rangle = \tilde{\ell}_{a,t}$ implies that

$$\sum_{t=1}^{T} \left( \hat{\ell}_t - \gamma_t \sum_{j \in \mathbb{A}_t} \tilde{\ell}_{j,t} - \tilde{\ell}_{a,t} \right) \leq \psi_{T+1}(e_a) - \min_{x \in \Delta_K} \psi_1(x) + \frac{1}{2\alpha} \sum_{t=1}^{T} \eta_t \sum_{i=1}^{K} \tilde{\ell}_{i,t}^2 q_{i,t}^{2-\alpha}.$$

By the definition of the loss estimate for non-active arms in (28), in the rounds where $I_{a,t} = 0$, we have $\hat{\ell}_t - \gamma_t \sum_{j\in\mathbb{A}_t} \tilde{\ell}_{j,t} - \tilde{\ell}_{a,t} = 0$. It follows that

$$\sum_{t=1}^T I_{a,t}\left(\hat{\ell}_t - \gamma_t \sum_{j\in\mathbb{A}_t}\tilde{\ell}_{j,t} - \tilde{\ell}_{a,t}\right) = \sum_{t=1}^T\left(\hat{\ell}_t - \gamma_t \sum_{j\in\mathbb{A}_t}\tilde{\ell}_{j,t} - \tilde{\ell}_{a,t}\right)$$

$$\leq \psi_{T+1}(e_a) - \min_{x\in\Delta_K}\psi_1(x) + \frac{1}{2\alpha}\sum_{t=1}^T\eta_t\sum_{i=1}^K\tilde{\ell}_{i,t}^2 q_{i,t}^{2-\alpha}. \tag{72}$$

By the non-negativity of the regularizer function, we have

$$\psi_{T+1}(e_a) - \min_{x\in\Delta_K}\psi_1(x) = \psi_{T+1}(e_a)$$
$$= \frac{1}{\eta_{T+1}}\left(\psi(e_a) - \min_{v\in\Delta_K}\psi(v)\right)$$
$$= \frac{K^{1-\alpha}-1}{\eta_{T+1}(1-\alpha)},$$

where the third equality is from $\psi(e_a) = 0$ and $\min_{v\in\Delta_K}\psi(v) = \frac{1-K^{1-\alpha}}{1-\alpha}$ by properties of Tsallis entropy function (Abernethy et al., 2015). Since the round $T+1$ does not contribute to the total regret, we can set $\eta_{T+1} = \eta_T$ and obtain $\psi_{T+1}(e_a) - \min_{x\in\Delta_K}\psi_1(x) \leq \frac{K^{1-\alpha}-1}{\eta_T(1-\alpha)}$. Furthermore, by Lemma 10 in (Nguyen & Mehta, 2024), for all $t\geq 1$,

$$\sum_{i=1}^K \tilde{\ell}_{i,t}^2 q_{i,t}^{2-\alpha} \leq \sum_{j\in\mathbb{A}_t}\tilde{\ell}_{j,t}^2 p_{j,t}^{2-\alpha}.$$

It follows that the right-hand side in (72) can be further bounded by

$$\sum_{t=1}^T I_{a,t}\left(\hat{\ell}_t - \gamma_t \sum_{j\in\mathbb{A}_t}\tilde{\ell}_{j,t} - \tilde{\ell}_{a,t}\right) \leq \frac{K^{1-\alpha}-1}{\eta_T(1-\alpha)} + \frac{1}{2\alpha}\sum_{t=1}^T\eta_t\sum_{j\in\mathbb{A}_t}\tilde{\ell}_{j,t}^2 p_{j,t}^{2-\alpha}$$
$$\leq \frac{K^{1-\alpha}-1}{\eta_T(1-\alpha)} + \frac{1}{2\alpha}\sum_{t=1}^T\eta_t\sum_{j\in\mathbb{A}_t}\tilde{\ell}_{j,t} p_{j,t}^{1-\alpha}$$
$$\leq \frac{K^{1-\alpha}-1}{\eta_T(1-\alpha)} + \frac{1}{2\alpha}\sum_{t=1}^T\eta_t\sum_{j\in\mathbb{A}_t}\tilde{\ell}_{j,t}$$

where the second inequality is due to $\tilde{\ell}_{j,t}p_{j,t} = \frac{\hat{\ell}_t \mathbb{1}\{i_t=j\}p_{j,t}}{p_{j,t}+\gamma_t} \leq 1$ for all $j\in\mathbb{A}_t$ and the last inequality is due to $p_{j,t}^{1-\alpha}\leq 1$ for $p_{j,t}\in[0,1]$ and $\alpha\in(0,1)$. Moving $\sum_{t=1}^T I_{a,t}\gamma_t\sum_{j\in\mathbb{A}_t}\tilde{\ell}_{j,t}$ to the right-hand side and using $I_{a,t}\leq 1$, we obtain

$$\sum_{t=1}^T I_{a,t}(\hat{\ell}_t - \tilde{\ell}_{a,t}) \leq \frac{K^{1-\alpha}-1}{\eta_T(1-\alpha)} + \sum_{t=1}^T\left(\frac{\eta_t}{2\alpha}+\gamma_t\right)\sum_{j\in\mathbb{A}_t}\tilde{\ell}_{j,t}. \tag{73}$$

We then apply Lemma E.2 twice and Corollary E.3 once, each of them with $\delta' = \frac{\delta}{3}$. The first application of Lemma E.2 uses $\nu_{i,t} = \eta_t \leq 2\gamma_t$ and obtains with probability at least $1-\delta/3$,

$$\sum_{t=1}^T\eta_t\sum_{j\in\mathbb{A}_t}\tilde{\ell}_{j,t} \leq \ln\left(\frac{3}{\delta}\right) + \sum_{t=1}^T\eta_t\sum_{j\in\mathbb{A}_t}\ell_{j,t}. \tag{74}$$

The second application of Lemma E.2 uses $\nu_{i,t} = 2\gamma_t$ and obtains with probability at least $1-\delta/3$,

$$\sum_{t=1}^T 2\gamma_t\sum_{j\in\mathbb{A}_t}\tilde{\ell}_{j,t} \leq \ln\left(\frac{3}{\delta}\right) + \sum_{t=1}^T 2\gamma_t\sum_{j\in\mathbb{A}_t}\ell_{j,t}. \tag{75}$$

An application of Corollary E.3 leads to

$$\sum_{t=1}^{T} I_a \tilde{\ell}_{a,t} \leq \frac{\ln(3K/\delta)}{2\gamma_T} + \sum_{t=1}^{T} I_a \ell_{a,t} \tag{76}$$

with probability at least $1 - \delta/3$. Plugging (74), (75) and (76) into (73) and taking a union bound, we obtain that with probability at least $1 - \delta$,

$$\sum_{t=1}^{T} I_{a,t}(\hat{\ell}_t - \ell_{a,t}) \leq \frac{K^{1-\alpha} - 1}{\eta_T(1-\alpha)} + \frac{\ln(3K/\delta)}{2\gamma_T} + \left(\frac{1}{2\alpha} + \frac{1}{2}\right) \ln(3/\delta) + \sum_{t=1}^{T} \left(\frac{\eta_t}{2\alpha} + \gamma_t\right) \sum_{j \in \mathbb{A}_t} \ell_{j,t}$$

$$\leq \frac{K^{1-\alpha} - 1}{\eta_T(1-\alpha)} + \frac{\ln(3K/\delta)}{2\gamma_T} + \left(\frac{1}{2\alpha} + \frac{1}{2}\right) \ln(3/\delta) + \sum_{t=1}^{T} \left(\frac{\eta_t}{2\alpha} + \gamma_t\right) A_t,$$

holds for simultaneously for all $a \in [K]$, where the last inequality is $\sum_{j \in \mathbb{A}_t} \ell_{j,t} \leq \sum_{j \in \mathbb{A}_t} 1 = A_t$. $\qquad\square$

Finally, we prove Theorem B.3.

*Proof (of Theorem B.3).* Since Theorem E.1 holds for any $\alpha$ arbitrarily close to 1, we can take the limit of $\alpha$ to 1 on the right-hand side of its bound and obtain the desired bound in Theorem B.3:

$$\max_{a \in [K]} \text{Regret}(a) \leq \frac{\ln(K)}{\eta_T} + \frac{\ln(3K/\delta)}{2\gamma_T} + \ln(3/\delta) + \sum_{t=1}^{T} \left(\frac{\eta_t}{2} + \gamma_t\right) A_t. \tag{77}$$

$\qquad\square$

## F. Stochastic OMD with non-increasing, time-varying learning rate

(Zhang et al., 2023) lamented that they could not find an analysis of stochastic mirror descent for non-oblivious online convex optimization with stochastic gradients, and they therefore proved their own high probability result. Their result uses a fixed learning rate, whereas we would like to avoid needing knowledge of the time horizon $T$ and therefore will describe how one can trivially (in light of known results) extend their derivation to the case of a non-increasing learning rate. All that is needed is to extend their upper bounds in equations (40) and (44) in their work to the case of a non-increasing learning rate $\eta_{w,t}$ (so that $\eta_{w,t+1} \leq \eta_{w,t}$ for $t \in [T]$). Such an extension is for free using, e.g., Theorem 6.10 of (Orabona, 2019). All other steps of the proof of Theorem 2 of (Zhang et al., 2023) can proceed without any important modifications, including the application of the Hoeffding-Azuma inequality. Here, we just highlight a few keyframes of the proof.

Using our notation and with non-increasing learning rate sequence $(\eta_{w,t})_{t \geq 1}$ and applying Theorem 6.10 of (Orabona, 2019), the bound in equation (40) of (Zhang et al., 2023) becomes, for any $\theta \in \Theta$,

$$\sum_{t=1}^{T} \langle \tilde{g}_t, \theta_t - \theta \rangle \leq \frac{D^2}{\eta_{w,T}} + \frac{G^2}{2} \sum_{t=1}^{T} \eta_{w,t}.$$

Fastforwarding to our analogue of equation (42) of (Zhang et al., 2023), we now get

$$\max_{\theta \in \Theta} \sum_{t=1}^{T} \left[\phi(\theta_t, q_t) - \phi(\theta, q_t)\right] \leq \frac{D^2}{\eta_{w,T}} + \frac{G^2}{2} \sum_{t=1}^{T} \eta_{w,t} + \max_{\theta \in \Theta} \left\{\sum_{t=1}^{T} \langle \nabla_\theta \phi(\theta_t, q_t) - \tilde{g}_t, \theta_t - \theta \rangle\right\}.$$

Setting

$$\tilde{\theta}_{t+1} = \underset{\theta \in \Theta}{\arg\min} \left\{\eta_{w,t}\langle \nabla_\theta \phi(\theta_t, q_t) - \tilde{g}_t, \theta - \tilde{\theta}_t \rangle + \frac{1}{2}\left\|\theta - \tilde{\theta}_t\right\|^2\right\},$$

we now again apply Theorem 6.10 of (Orabona, 2019) to get the following analogue of equation (44) of (Zhang et al., 2023):

$$\sum_{t=1}^{T} \langle \nabla_\theta \phi(\theta_t, q_t) - \tilde{g}_t, \tilde{\theta}_t - \theta \rangle \leq \frac{D^2}{\eta_{w,T}} + 2G^2 \sum_{t=1}^{T} \eta_{w,t}.$$

All of the remaining steps of the analysis of (Zhang et al., 2023) go through without any interesting modification, giving the result that with probability at least $1 - \delta$,

$$\sum_{t=1}^{T} \phi(\theta_t, q_t) - \min_{\theta \in \Theta} \sum_{t-1}^{T} \phi(\theta, q_t) \leq \frac{D^2}{\eta_{w,T}} + \frac{G^2}{2} \sum_{t=1}^{T} \eta_{w,t} + \frac{D^2}{\eta_{w,T}} + 2G^2 \sum_{t=1}^{T} \eta_{w,t} + 8DG\sqrt{T \ln \frac{1}{\delta}},$$

where the last term is from applying the Hoeffding-Azuma inequality in precisely the same way as in (Zhang et al., 2023). Using a learning rate schedule of $\eta_{w,t} = \eta_0 \cdot \frac{D}{G\sqrt{t}}$ gives the upper bound

$$DG\sqrt{T} \left( \frac{1}{\eta_0} + \eta_0 + \frac{1}{\eta_0} + 4\eta_0 + \sqrt{\ln \frac{1}{\delta}} \right) = DG\sqrt{T} \left( \frac{2}{\eta_0} + 5\eta_0 + \sqrt{\ln \frac{1}{\delta}} \right),$$

which, letting $\eta_0 = 1$, gives

$$DG\sqrt{T} \left( 7 + \sqrt{\ln \frac{1}{\delta}} \right) = O\left( DG\sqrt{T \ln \frac{1}{\delta}} \right),$$

as desired.

# G. Details of the Experiments

In this section, we provide the full setup details of the experiments presented in Section 6.

### G.1. The Lower Bound Environment

For the GDRO problem instance constructed based on the lower bound construction in the proof of Theorem 3.5, we scale the loss by $\frac{1}{2}$ to ensure that the losses are in $[0, 1]$. This implies that for a hypothesis $\theta \in [0, 1]$, its maximum risk over $K$ groups is

$$\mathcal{L}(\theta) = \frac{1}{2} \max \left( \Delta\theta + \frac{1}{2}, \Delta(1 - \theta) + \frac{1}{2} \right).$$

We set $\Delta = 0.1$. The optimal hypothesis is $\theta^* = 0.5$ with $\mathcal{L}(\theta^*) = \frac{1.1}{4} = 0.275$. The optimality gap of a hypothesis $\theta$ is

$$\text{err}(\theta) = \mathcal{L}(\theta) - \mathcal{L}(\theta^*) = \frac{1}{2}\Delta \left| \frac{1}{2} - \theta \right| = 0.05 \left| \frac{1}{2} - \theta \right|.$$

With the desired optimality gap of $\epsilon = 0.005$, the acceptable range of the risk of $\bar{\theta}_T$ is $[0.27, 0.28]$. The set of $\epsilon$-optimal hypotheses is obtained by solving $0.05 \left| \frac{1}{2} - \theta \right| \leq 0.005$, which implies that $\theta \in [0.4, 0.6]$ is the set of $\epsilon$-optimal hypotheses.

### G.2. The Adult Dataset

**Loss function and data normalization**. Similar to (Soma et al., 2022), we train a linear classifier with hinge loss

$$\ell(\theta, z, y) = \max(0, 1 - y\langle \theta, z \rangle),$$

where $z \in \mathbb{R}^5$ is a feature vector of a sample and $y \in \{-1, 1\}$ is the label.

On the Adult dataset, the default value of the features could be much larger than 1, leading to loss values larger than 1. To avoid exceedingly large losses, we compute the maximum norm of all feature vectors in the dataset and then divide all features by this maximum norm. Note that the same maximum norm value is used for all 10 groups.

**UCI Adult Dataset** As mentioned in the main text, we construction $K = 10$ groups from five races `White`, `Black`, `Asian-Pac-Islander`, `Amer-Indian-Eskimo`, `Other` and two genders `male`, `female`. The dataset of $48\,842$ samples is heavily imbalanced. The largest group is (`White`, `male`) having $28\,736$ samples while the smallest group is (`Other`, `female`) having $156$ samples.

**No batch processing**. Our results in Section 6 are generated by the exact algorithms described in Sections 3 and 4 without adding any batch processing. This is different from (Soma et al., 2022), who used a batch of 10 samples to stabilize the gradients. We find that as the dominant sets quickly converge to just one or two groups, especially the groups with small amount of samples such as (`Amer-Indian-Eskimo`, `female`), the gradients computed from just one random sample are sufficiently stable with the long horizon of $T = 10^6$.

**Computing (an estimate of)** $\theta^*$. In order to obtain the optimality gap of `SB-GDRO-SA` and `SMD-GDRO`, we compute an estimate of $\mathcal{L}(\theta^*)$ using the following algorithm: we run a deterministic two-player zero-sum game in which both players have full knowledge of $(\mathcal{P}_i)_{i=1,2,\ldots,K}$. In each round $t$, the max-player is able to compute a dominant set consisting of just one group – that is, the group with maximum risk on $\theta_t$. Similarly, the min-player is given the expected value of the gradient $\mathbb{E}[\tilde{g}_t]$ instead of the stochastic gradients. We run the game for $T = 10^7$ rounds and record the final maximum risk of $\bar{\theta}_T$ to be $\mathcal{L}(\theta^*) \approx 0.49945$. This final maximum risk of is observed to be on group 8 (i.e., female Amer-Indian-Eskimo).

## H. Discussion of the Competing Approach in Stochastically Constrained Adversarial Regime

Our approach to going beyond minimax bounds in GDRO is based on the $(\lambda, \beta)$-sparsity condition and, algorithmically, based on the sleeping bandits framework. The expert bandit reader may wonder about the viability of the following competing approach: suppose that after some unknown time horizon $\tau$, all $\theta_t$'s fall within a radius-$\rho$ ball of $\theta^*$, and within such a ball, further suppose for simplicity that a unique group obtains the maximum risk in all subsequent rounds by a margin of at least $\lambda$. This setup generalizes the previously studied stochastically constrained adversarial (SCA) regime (Wei & Luo, 2018; Zimmert & Seldin, 2021) wherein the best arm's mean is separated with a gap from the other arms' means *for all rounds*. In this generalized SCA regime, one might hope for better regret bounds for the max player than we achieve using our sleeping bandits-based approach. However, there are at least three major challenges: first, to our knowledge, it is not known how to get high probability regret bounds in the SCA regime even when $\tau = 1$; second, we are not aware of results that provide last iterate convergence so that, eventually, all iterates $\theta_t$ are within distance $\rho$ of $\theta^*$ (SCA requires such convergence); third, there could well be multiple best arms or multiple nearly best arms, which recently has been addressed in some different regimes but adds another layer of complexity for the generalized SCA regime.

As mentioned in Section 7, if we had last iterate convergence, then our $(\lambda, \beta)$-sparsity condition could be relaxed to hold only within some proximity of $\theta^*$. However, our condition is more flexible as compared to SCA since it is not known how the latter can be analyzed when the best arm (or set of best arms) changes throughout the game, whereas such a changing set of approximate (within gap $\lambda$) maximizers fits naturally with sleeping bandits.

