# OpenReview forum: "Beyond Minimax Rates in Group Distributionally Robust Optimization via a Novel Notion of Sparsity"
_ICML.cc/2025/Conference — ICML 2025 poster_

### Official Review · Reviewer_S1YD · 2025-03-05

**Overall Recommendation:** 3

**Summary:**

The paper discusses the group distributionally robust optimization framework. They propose a new sparsity measurement of the distributions called $(\lambda,\beta)$-sparsity, and show that the dependence on K (the number of distributions) can be reduced to log K.

**Claims And Evidence:**

Yes

**Essential References Not Discussed:**

None.

**Experimental Designs Or Analyses:**

The setting of the experimental is reasonable. But, the author shows that $(\lambda,\beta)$-sparsity condition holds for parameters around the optimal parameter in Adult dataset. Is it possible to show the sparsity condition holds for all parameters in this case? Or, in the other hand, if the sparsity condition only holds for the parameters around the optimal parameter $\theta^*$, does the theoretical result still hold?

**Methods And Evaluation Criteria:**

Yes

**Other Comments Or Suggestions:**

None

**Other Strengths And Weaknesses:**

Strengths
1.	The technique shows that there exists interesting relationships between the problem and the sleeping bandits framework.

Weaknesses
1.	Is there any motivation to show that $(\lambda,\beta)$-sparsity is reasonable in the practical problems? Especially, $(\lambda,\beta)$-sparsity requires that for ''arbitrary'' parameter, there exists a small set of distributions whose risk are much larger than the risks of the other distributions. This seems a very strong assumption.

**Questions For Authors:**

The main question I have is to what extent, the $(\lambda,\beta)$-sparsity assumption holds for the practical case. See "Experimental Designs Or Analyses" and "Other weaknesses" for the detailed questions.

**Relation To Broader Scientific Literature:**

The results theoretically prove that when there is significant ''gap'' between the distributions, the sample complexity to find a good parameter can be largely reduced w.r.t K. This is an interesting finding.

**Theoretical Claims:**

I don't check the details of the proof in the appendix, and I don't find any issues in the general framework of the proof .

---

> ### Author Rebuttal · Authors · 2025-03-29
>
> We thank the reviewer for raising great points regarding the practicality of our $(\lambda, \beta)$-sparsity definition.
>
> While we acknowledge that our Definition 2.3 requires a strong condition that a non-trivial $(\lambda, \beta)$-sparsity holds globally for all $\theta \in \Theta$, we would like to explain why our work is an important step towards studying the more practical setting in which the $(\lambda, \beta)$-sparsity condition may hold locally in a neighborhood around the optimal $\theta^*$.
>
> + From a practical perspective, our experimental results show that our algorithms, which are developed for the global $(\lambda, \beta)$-sparsity condition, works really well on the real-life Adult dataset where the $(\lambda, \beta)$-sparsity condition holds only locally.
>     This is partially explained in our Equation 27, where we bound the regret of the max-player by the *average size of dominant sets* $\bar{\beta}\_T = \frac{1}{T}\sum\_{t=1}^T \beta\_t$.
>     This indicates if the sequence of $(\theta\_t)\_t$ in an algorithm converges sufficiently quickly to a neighborhood of $\theta^*$, then the majority of the dominant sets will have small sizes. This turns out to be true for our algorithms as empirically verified in our Figure 2 (left).
> + From a theoretical perspective, in order to prove mathematical statements about the performance of an algorithm on the local version of $(\lambda, \beta)$, there must be some robust mechanism to control how quickly the sequence of $(\theta_t)_t$ converges to a neighborhood of $\theta^*$. This requires developing a high-probability *last-iterate* guarantee for stochastic saddle-point optimization, which is a fundamental problem in optimization. To the best of our knowledge, there are no existing works on this problem that are applicable to our GDRO setting without further (strong) assumptions (e.g. strong convexity).
>     Even with strong convexity, the best known rates of convergence are poor. Without strong convexity, to our knowledge nothing is known. We are excited about future progress on this question, but we emphasize that it would be a major undertaking in its own right.
> + In our response to Reviewer K8N8, we proved that $(\lambda, \beta)$-sparsity condition holds *globally* at every $\theta \in \Theta$ for the problem of linear regression with a linear Gaussian model.
>     Together with the examples in our paper, this additional theoretical result further strongly motivate the global version of the $(\lambda, \beta)$-sparsity condition in our work.

---

### Official Review · Reviewer_6PQC · 2025-03-13

**Overall Recommendation:** 3

**Summary:**

This paper studies the GDRO problem where a special structural assumption called $(\lambda,\beta)$-sparsity is satisfied. This assumption requires that for any hypothesis $\theta\in \Theta$, only $\le \beta$ distributions have a "large" risk, which is quantified by the parameter $\lambda$.

1. Given any $\lambda$, let $\beta_\lambda$ be the smallest $\beta$ s.t. $(\lambda,\beta)$-sparsity is true. Then an algorithm with SC $\tilde{\mathcal O}(\frac{Kd}{\lambda^2}+\frac{\beta_\lambda}{\epsilon^2})$ (omitting a bunch of factors) is prosed.
2. Without knowing $\lambda$, a fully adaptive algorithm that automatically finds the "optimal" $\lambda^\ast$ is derived. Resolving the computational efficiency issue, a semi-adaptive algorithm that attains asymptotically best SC is derived.
3. Finally, an $\Omega(\frac{\beta}{\epsilon^2})$ LB is given.

**Claims And Evidence:**

Yes, the proofs are sketched clearly

**Essential References Not Discussed:**

No

**Experimental Designs Or Analyses:**

The two experiments look good from a theoretical perspective.

**Methods And Evaluation Criteria:**

I'm not sure whether the $(\lambda,\beta)$-sparsity notion is well-justified. But other than that, every comparison with previous work is fair.

**Other Comments Or Suggestions:**

No

**Other Strengths And Weaknesses:**

Strength:

1. The organization of this paper is pretty clear. Motivations and overviews are well-written.
2. The adaptivity to $\lambda^\ast$ is pretty good.

Weakness:

1. The techniques in Algo 1 looks relatively straightforward given the $(\lambda,\beta)$-sparsity definition (basically what people do for sparse linear bandits). The two-player max-min game framework is standard in GDRO, and the sleeping bandit part is a direct application of a recent paper.
2. The sparsity definition lacks justification. I understand that the bound is always tighter than $\mathcal O(\frac{K}{\epsilon^2})$ as $[K]$ is $(1,K)$-sparse, but it doesn't really infer that $(\lambda,\beta)$-sparsity is the "correct" notion.
3. This paper only focuses on the convex GDRO setup.

**Questions For Authors:**

See Weaknesses. My overall concern is that, while $(\lambda,\beta)$-sparsity is a valid notion, it is unknown whether it is the "right" one.

---

Score updated based on authors' clarification.

**Relation To Broader Scientific Literature:**

It could be of interest to the more general VC dimension case

**Theoretical Claims:**

I only went through the sketches.

---

> ### Author Rebuttal · Authors · 2025-03-29
>
> We thank the reviewer for their critical feedback. Please find our clarifications to your concerns below.
>
> # 1. The techniques in Algo 1 looks relatively straightforward
>
> We understand that the algorithm may look straightforward because it takes in a given $\lambda$ and thus can focus solely on computing the dominant sets. Even so, we emphasize that the analysis of Algorithm 1 is highly non-trivial as it requires developing two new techniques: one is a high-probability anytime bound for sleeping bandits, and one is a high-probability anytime correctness for the computation of the dominant sets. These two new techniques are then combined by our Lemma 3.3, which is a new result showing that the high-probability sleeping bandits bound based on stochastically sampled losses also implies a regret bound based on the (hidden) expected losses. These new results and their proof techniques are highly nontrivial.
>
> We also point out that Algorithm 1 and its result are only the first of our *five* main results. In particular, the adaptive algorithms (and their analyses) were major theoretical undertakings. For example, the theory behind Theorem 4.1 required us to put forth new ideas (for space, in the appendix, although there is a brief sketch in Section 4.1), and this result itself was highly surprising to us as we originally were trying to prove a lower bound that shows such adaptivity was impossible.
>
> # 2. The sparsity definition lacks justification
> While we acknowledge that our Definition 2.3 requires a strong condition that a non-trivial $(\lambda, \beta)$-sparsity holds globally for all $\theta \in \Theta$, we would like to explain why our work is an important step towards studying the more practical setting in which the $(\lambda, \beta)$-sparsity condition may hold locally in a neighborhood around the optimal $\theta^*$.
>
> + From a practical perspective, our experimental results show that our algorithms, which are developed for the global $(\lambda, \beta)$-sparsity condition, works really well on a real-world dataset where the $(\lambda, \beta)$-sparsity condition holds only locally.
>     This is partially explained in our Equation 27, where we bound the regret of the max-player by the *average size of dominant sets* $\bar{\beta}\_T = \frac{1}{T}\sum\_{t=1}^T \beta\_t$.
>     This indicates if the sequence of $(\theta\_t)\_t$ in an algorithm converges sufficiently quickly to a neighborhood of $\theta^*$, then the majority of the dominant sets will have small sizes. This turns out to be true for our algorithms as empirically verified in our Figure 2 (left).
> + From a theoretical perspective, in order to prove mathematical statements about the performance of an algorithm on the local version of $(\lambda, \beta)$, there must be some robust mechanism to control how quickly the sequence of $(\theta_t)_t$ converges to a neighborhood of $\theta^*$. This requires developing a high-probability *last-iterate* guarantee for stochastic saddle-point optimization, which is a fundamental problem in optimization. To the best of our knowledge, there are no existing works on this problem that are applicable to our GDRO setting without further (strong) assumptions (e.g. strong convexity).
>     Even with strong convexity, the best known rates of convergence are poor. Without strong convexity, to our knowledge nothing is known. We are excited about future progress on this question, but we emphasize that it would be a major undertaking in its own right.
> + In our response to Reviewer K8N8, we proved that $(\lambda, \beta)$-sparsity condition holds *globally* at every $\theta \in \Theta$ for the problem of linear regression with a linear Gaussian model.
>     Together with the examples in our paper, this additional theoretical result further strongly motivate the global version of the $(\lambda, \beta)$-sparsity condition in our work.
>
> # 3. This paper only focuses on the convex GDRO setup.
> Without any further assumptions, due to the NP-Hardness of non-convex optimization, a general non-convex GDRO setup would require highly problem-specific assumptions and likely a new definition of robustness as well, both of which are not the focus of our work.
> In addition, the convex setup still has a number of interesting open problems (see e.g. Zhang et. al. 2023). Therefore, while we agree that ultimately the non-convex GDRO problems (with problem-specific constraints) is of great practical interest, works on convex settings such as ours are important in understanding the optimality and sample efficiency of the GDRO framework.
>
> **References**
>
> Zhang et. al. 2023. Stochastic approximation approaches to group distributionally robust optimization. NeurIPS 2023.

---

> > ### Comment · Reviewer_6PQC · 2025-04-01
> >
> > Thank you for your clarification regarding technical contributions in Alg 1. **Can you elaborate further on the anytime bound part?** From Theorem 3.2 it looks like the modified version of SB-EXP3 still only gives a bound for T-round regret. It looks like from the proof of Lemma 3.3, it uses the anytime property of Lemma 3.1; but it isn't related to the sleeping bandit part.
> >
> > Thanks for mentioning Section 4.1. Yes, I agree that the adaptivity to $\lambda^\ast$ is very interesting. I acknowledged this as a strength in my review.
> >
> > I get your explanation about the local sparsity, which is much weaker than the Definition 2.3 that requires global sparsity. It is also natural to hope that around the optimum, there are only few near-optimal choices. Given this, I think the $(\lambda,\beta)$-sparsity notion is more valid. Nevertheless, I didn't really catch the "around $\theta^\ast$" when I read Section 5.1, and I recommend the authors to elaborate further in their revision.
> >
> > I am now leaning more towards an accept, mainly due to authors' satisfactory response to my W2.

---

> > > ### Author Response · Authors · 2025-04-02
> > >
> > > We thank the reviewer for the positive evaluation of our answer, we will update the discussion on the local version of $(\lambda, \beta)$-sparsity in future version as suggested.
> > >
> > > **Regarding the question on the anytime bound part of our paper:** this is a great question. Here, we will focus on Algorithm 1 and the results in Section 3 (everything extends to the rest of our algorithms). First, we highlight that the *anytime* bound for the max-player is significantly more important in our work than previous works in GDRO. This is because in our work, the number of rounds $T$ is not fixed before the game begins, but the game (as a stochastic process) will stop once there is enough evidence that the average hypothesis $\bar{\theta}_T$ is $\epsilon$-optimal (with high probability).
> > > This evidence is based on the average size of the (fully observable) dominant sets.
> > > Therefore, it is important that our bounds hold for a *stopping time* $T$. As presented in line 1011 and 1012, the stopping condition in our work makes $T$ a valid stopping time since it is bounded by a constant and is measureable with respect to all randomness observable up to time $T$ (no future information is used).
> > >
> > > Next, we elaborate how Theorem 3.2 and Lemma 3.1 hold for any stopping time $T$.
> > > - For Theorem 3.2, the anytime property comes from two components. The first is the adaptivity of the learning rates $\eta_{q,t}$ and $\gamma_t$, which depend only on the dominant sets observed upto time $t$. The second is the fact that for any valid stopping time $T$, the estimated cumulative loss by the IX-loss estimator in (Neu, 2015) is close to the true cumulative loss (our Lemma D.2).
> > > - For Lemma 3.1, the anytime property comes from the fact that a dominant set can be computed correctly for all $\theta \in \Theta$ with high probability (our Lemma A.1). Since all $\theta_t$ are in $\Theta$ by the update rule of the min-player, the anytime correctness of the dominant sets is guaranteed.
> > >
> > > We hope our answer clear things up. Please let us know if you have any further questions.

---

### Official Review · Reviewer_4vrz · 2025-03-14

**Overall Recommendation:** 3

**Summary:**

This paper considers a practical setting of the group DRO problem called (lambda,beta)-sparisty, which means for any parameter there is a set of at most beta groups whose risks are all at least lambda larger than the risks of other group. By taking this condition into account, the authors can derive sharper complexity bounds for this problem.

**Claims And Evidence:**

Yes. The authors provide rigouosu theoretical justification and provide intuition by drawing the connection with sleeping bandits problem.

**Essential References Not Discussed:**

N/A

**Experimental Designs Or Analyses:**

Yes. The experiment designs follow existing literature and make sense.

**Methods And Evaluation Criteria:**

Experiments are conducted on a synthetic dataset and the real-world Adult dataset with fairness applications.

**Other Comments Or Suggestions:**

N/A

**Other Strengths And Weaknesses:**

N/A

**Questions For Authors:**

N/A

**Relation To Broader Scientific Literature:**

N/A

**Theoretical Claims:**

I do not check the technical proofs in detail, but it seems that the theoretical claims do not contain mistakes.

---

> ### Author Rebuttal · Authors · 2025-03-29
>
> We thank the reviewer for their time on our paper. Please let us know if you have any questions.

---

> > ### Comment · Reviewer_4vrz · 2025-04-09
> >
> > I thank the authors for their efforts in addressing reviewers' concerns. I have read those comments and confirm that I am, in general, satisfied with the contributions of this paper and do not have any other concerns.

---

### Official Review · Reviewer_K8N8 · 2025-03-14

**Overall Recommendation:** 3

**Summary:**

In this paper, the authors revisit the problem of an optimization framework where a single hypothesis is chosen to handle a group of K risks associated with K data distributions - this framework is known as GDRO. While minimax rates have been established for this problem already, the authors provide a finer-grained analysis of this problem in terms of a characteristic feature of the K groups at hand known as $(\lambda,\beta)$ sparsity. The authors provide novel algorithms based on sleeping bandits to provide improved problem dependent rates.

**Claims And Evidence:**

Yes

**Essential References Not Discussed:**

None to best of my knowledge

**Experimental Designs Or Analyses:**

Yes

**Methods And Evaluation Criteria:**

Yes

**Other Comments Or Suggestions:**

See below

**Other Strengths And Weaknesses:**

The paper has strength but there are certain weaknesses outlined in the questions section

**Questions For Authors:**

While the paper was very interesting to read I have some questions and most likely, better explanations can help the flow of the paper.

1. My first question is how can we characterize the non-triviality of a group (Definition 2.3) given a set of K distributions and a set of parameters $\Theta$. Can we design an efficient algorithm for the same? For instance, I can think of the linear regression problem in n dimensions where the prior distribution of covariates has K possibilities N(\mu_1, I), N(\mu_2, I), N(\mu_K, I). For a feature vector x, the output y is simply N( < \mu, x>, 1). How can we know the $\lambda,\beta$ for this setting given \{\mu_i\}? I am worried that in most situations, $\lambda,\beta$ is trivial. Please convince me otherwise.

2. Why is it that the algorithm designer can choose the distribution from which the data can be generated at each round? When is this applicable? I understand that this allows usage of bandit techniques but I am failing to see a real world situation when this is applicable.

3. In L103, what does two dimension-dependent bounds mean?

4. Can the authors provide intuition on how the individual regret bounds (eqs 3 and 4) relate to the final sample complexity? it is not immediately clear.

5. For unknown $\lambda$, is it possible to use some kind of meta-algorithm? For instance, perhaps we can use corralling algorithms outlined in

Alekh Agarwal, Haipeng Luo, Behnam Neyshabur, and Robert E Schapire. Corralling a band of bandit algorithms. In Conference on Learning Theory, pp. 12–38. PMLR, 2017.

**Relation To Broader Scientific Literature:**

The literature has already been covered well in the paper

**Theoretical Claims:**

No, I did not through the proof details carefully

---

> ### Author Rebuttal · Authors · 2025-03-29
>
> We thank the reviewer for their feedback and interesting questions, especially the one on $(\lambda, \beta)$-sparsity for linear regression. Please find our answers to your questions below.
> # 1. $(\lambda, \beta)$-sparsity for linear regression with linear Gaussian model.
> For the linear Gaussian model mentioned by the reviewer, we will show that a non-trivial $(\lambda, \beta)$-sparsity holds with large $\lambda = 0.5$ and small $\beta = 2$, even for an arbitrarily large number of groups $K$. Moreover, we are able to show this for dimension $n = 1$ (everything can extend to higher dimensions). Before continuing with the construction, let us first mention that our algorithms are adaptive to problems both with and without non-trivial $(\lambda, \beta)$-sparsity; therefore, if a problem does not have any non-trivial sparsity, our algorithms will automatically recover the best known worst-case bounds.
>
> We now show our construction. For any group $j$:
> + we assume $X \sim \mathcal{N}(\mu_j, I)$, as suggested by the reviewer;
> + for $Y$, we assume $Y \sim \mathcal{N}(\langle \theta^*_j, X \rangle, 1)$.
>
> We quickly clarify an important point about the conditional distribution of $Y$ given $X$. While the reviewer may have suggested $Y \sim \mathcal{N}(\langle \mu, X \rangle, 1)$ for all groups $j$, we note that this would imply all groups share a common minimizer, in which case the problem instance is not suited to GDRO (because there is no robustness issue, i.e., minimizing the maximum risk over groups becomes the same as minimizing the risk of any individual group).
>
> Next, taking $K = 3$ and $n  = 1$, we show a choice of the $\mu_j$'s and $\theta^*_j$'s for which $(\lambda = 0.5, \beta = 2)$-sparsity holds.
> We set the parameter space $\Theta = [-1, 1]$. Next, we take $\mu_j = 0$ for all $j$ and set $\theta^*_1 = -1$, $\theta^*_2 = 1$, and $\theta^*_3 = 0$. The model is well-specified since all $\theta_j^*$ are in $\Theta$. Using *squared loss*, straightforward calculations show that for any $\theta \in \Theta$, its risks on groups 1, 2, and 3 are $R_1(\theta) = (\theta+1)^2 + 1$, $R_2(\theta) = (\theta - 1)^2 + 1$, and $R_3(\theta) = \theta^2 + 1$, respectively. One can easily show that $\max\\{R_1(\theta), R_2(\theta) \\} - R_3(\theta) \geq 1$ holds for all $\theta \in \Theta$. This immediately implies that this GDRO instance is $(0.5, 2)$-sparse, and the $0.5$-dominant set is either {1}, {2} or {1, 2}.
>
> To extend to $K \geq 4$, for $j \geq 4$, we take $\mu_j = 0$ and let $\theta^*_j$ be slightly perturbed from $\theta^*_3$ so that all these groups' risks are always dominated by (the maximum of) groups 1 and 2 with a margin of 1. As a result, it still is the case that $(0.5, 2)$-sparsity holds.
>
> # 2. Why can the algorithm choose the distribution to sample from?
> Our approach, as well as other recent approaches to the GDRO problem that also employ adaptive sampling, is a form of active learning where in each round, the algorithm adaptively decides from which group it will obtain a sample. Theoretically, we show that one can have massive gains in sample complexity via an adaptive approach. For the question on *when* and *why* the algorithm designer would be able to actively sample, we point to a popular example from the well-cited paper of Sagawa et al. (2020): when training deep neural networks on unbalanced datasets, it is beneficial to feed a network samples from classes on which the network is struggling the most. Another example would be when a company has a fixed budget for data collection and wants to spend this budget the most efficiently. If groups are visible (for example, collecting samples in different geographic regions, each of which corresponds to a different group), then adaptive sampling as we do here is viable and sensible from a cost standpoint.
>
> # 3. Intuition on the players' regret bounds and sample complexity.
> Intuitively, small regret bounds of the two players imply a small bound on the optimality gap of the final output $\bar{\theta}_T$.
> The regret bounds of the two players depend on the number of rounds $T$ of the game.
> Since our algorithm collects one sample in each round of the two-player zero-sum game, the number of rounds $T$ of the game is equal to the number of samples collected for the game.
>
> # 4. Is it possible to use some kind of meta-algorithm?
> Without prior knowledge of $\lambda$, we believe it is not possible to use some meta-algorithm to achieve the same near-optimal sample complexity as our results. By running multiple instances of the same algorithm with different $\lambda$, the total sample complexity will always be dominated by the instance with the smallest $\lambda$. Without a procedure to lower bound the smallest value of $\lambda$ like our Algorithms 8 and 9, any meta-algorithm would end up using a $\lambda$ far below $\lambda^*$, thus incurring a much higher sample complexity.
>
> **References**
>
> Sagawa et. al., 2020. Distributionally robust neural networks. ICLR 2020

---

### Decision · Program_Chairs · 2025-05-01

**Decision:**

Accept (poster)

**Comment:**

This paper studies Group DRO in the form of minimizing the worst group loss. It considers a new notation of $\lambda, \beta$ sparsity structure. The authors have established several upper bounds, including when $\lambda$ is know and when the optimal $\lambda$ is unknown. While the paper has made some theoretical contributions, there are some minor concerns about the assumption, especially it requires the  $\lambda, \beta$ sparsity  holds for any model parameter. It would be great to only assume this assumption at the optimal model parameter. In addition, the considered formulation does not cover the top-k average group loss, which is more general. The authors are encouraged to cite and discuss some related work for optimizing the top-k average group DRO loss, e.g., Hu et al., Wang &Yang.

Given the overall contributions of this paper, I will recommend acceptance.

Hu et al. Non-Smooth Weakly-Convex Finite-sum Coupled Compositional Optimization.
Wang & Yang. ALEXR: Optimal Single-Loop Algorithms for Convex Finite-Sum Coupled Compositional Stochastic Optimization.